# Coral reef carbonate budgets and ecological drivers in the central Red Sea - a naturally high temperature and high total alkalinity environment

Anna Roik[1,*], Till Röthig[1,+], Claudia Pogoreutz[1], Vincent Saderne[1], Christian R. Voolstra[1]

[1]Red Sea Research Center, King Abdullah University of Science and Technology, 23955 Thuwal, Saudi Arabia

[*]Current address: Marine Microbiology, GEOMAR Helmholtz Centre for Ocean Research, 24105 Kiel, Germany

[+]Current address: Aquatic Research Facility, Environmental Sustainability Research Centre, University of Derby, Kedleston Road, Derby, DE22 1GB, UK

*Correspondence to*: Christian R. Voolstra (christian.voolstra@kaust.edu.sa), Anna Roik (aroik@geomar.de)

**Abstract.** The structural framework provided by corals is crucial for reef ecosystem function and services, but high seawater temperatures can be detrimental to the calcification capacity of reef-building organisms. The Red Sea is very warm, but total alkalinity (TA) is naturally high and beneficial for reef accretion. To date, we know little about how such beneficial and detrimental abiotic factors affect each other and the balance between calcification and erosion on Red Sea coral reefs, that is overall reef growth, in this unique ocean basin. To provide estimates of present-day reef growth dynamics in the central Red Sea, we measured two metrics of reef growth, i.e., *in situ* net-accretion/-erosion rates ($G_{net}$) determined by deployment of limestone blocks and ecosystem scale carbonate budgets ($G_{budget}$) along a cross-shelf gradient (25 km, encompassing near-, mid-, and offshore). Along this gradient, we assessed multiple abiotic (i.e., temperature, salinity, diurnal pH fluctuation, inorganic nutrients, and TA) and biotic (i.e., calcifier and epilithic bioeroder communities) variables. Both reef growth metrics revealed similar patterns from nearshore to offshore: net-erosive, neutral, and net-accretion states. The average cross-shelf $G_{budget}$ was 0.66 kg $CaCO_3$ $m^{-2}$ $y^{-1}$, with the highest budget of 2.44 kg $CaCO_3$ $m^{-2}$ $y^{-1}$ measured in the offshore reef. These data are comparable to the contemporary $G_{budgets}$ from the western Atlantic and Indian Ocean, but lie well below "optimal reef production" (5 - 10 kg $CaCO_3$ $m^{-2}$ $y^{-1}$) and below maxima recently recorded in remote high coral cover reef sites. Yet, the erosive forces observed in the Red Sea nearshore reef contributed less as observed elsewhere. A higher TA accompanied reef growth across the shelf gradient, whereas stronger diurnal pH fluctuations were associated with negative budgets. Noteworthy for this oligotrophic region was the positive effect of phosphate, which is a central micronutrient for reef building corals. While parrotfish contributed substantially to bioerosion, our dataset also highlights coralline algae as important local reef-builders. Altogether, our study establishes a baseline for reef growth in the central Red Sea that should be useful in assessing trajectories of reef growth capacity under current and future ocean scenarios.

## 1 Introduction

Coral reef growth is mostly limited to warm, aragonite-saturated, and oligotrophic tropical oceans and is pivotal for reef ecosystem functioning (Buddemeier, 1997; Kleypas et al., 1999). The coral reef framework not only maintains a remarkable biodiversity, but also provides highly valuable ecosystem services that include food supply and coastal protection, among others (Moberg and Folke, 1999; Reaka-Kudla, 1997). Biogenic calcification, erosion, and dissolution contribute to the formation of the reef framework constructed of calcium carbonate ($CaCO_3$, mainly aragonite). The balance of carbonate loss and accretion is influenced by biotic and abiotic factors. On a reef scale, the main antagonists are calcifying benthic communities on the one hand, such as scleractinian corals and coralline algal crusts, and grazing and endolithic bioeroders on the other hand, such as parrotfish, sea urchins, microbioeroding chlorophytes, boring sponges, and other macroborers (Glynn, 1997; Hutchings, 1986; Perry et al., 2008; Tribollet and Golubic, 2011). The export or loss of carbonate as sediments is considered an essential part, in particular in the wider geomorphic perspective of reef carbonate production states (Cyronak et al., 2013; Perry et al., 2008, 2017). Temperature and carbonate chemistry parameters (e.g., pH, total alkalinity: TA, and aragonite saturation state: $\Omega_a$, $p$CO$_2$) have been identified as important players in regulating these carbonate accretion and erosion processes (Albright et al., 2018; Schönberg et al., 2017). Furthermore, different light regimes across depths, water flow, and wave exposure can alter the rates of reef-formation processes (Dullo et al., 1995; Glynn and Manzello, 2015; Kleypas et al., 2001).

Reef growth is maintained when reef calcification produces more $CaCO_3$ than is being removed, and depends largely on the ability of benthic calcifiers to precipitate calcium carbonate from seawater (e.g., Langdon et al., 2000; Tambutté et al., 2011). TA and $\Omega_a$, positively correlate with calcification rates (Marubini et al., 2008; Schneider and Erez, 2006), and while calcification rates of corals and coralline algae increase with higher temperature, they have upper thermal limits (Jokiel and Coles, 1990; Marshall and Clode, 2004; Vásquez-Elizondo and Enríquez, 2016). Today's Oceans are warming and high temperatures begin to exceed the thermal optima of calcifying organisms and thereby slowing down or interrupting calcification (e.g., Carricart-Ganivet et al., 2012; Death et al., 2009). At the same time, ocean acidification decreases the Oceans' pH and $\Omega_a$ (Orr et al., 2005). Arguably, calcification under these conditions becomes energetically costlier (Cai et al., 2016; Cohen and Holcomb, 2009; Strahl et al., 2015; Waldbusser et al., 2016). In addition, ocean acidification stimulates destructive processes, for instance the proliferation of bioeroding endolithic organisms (e.g., Enochs, 2015; Fang et al., 2013; Tribollet et al., 2009). Apart from that, locally impaired reef growth due to an increased intensity or frequency of extreme climate events (Eakin, 2001; Schuhmacher et al., 2005), human impacts including pollution and eutrophication (Chazottes et al., 2002; Edinger et al., 2000), and other ecological events such as population outbreaks of grazing sea urchins or crown-of-thorn starfish that feed on coral can induce reef framework degradation (Bak, 1994; Pisapia et al., 2016; Uthicke et al., 2015).

A number of studies have employed experimental limestone blocks cut from coral skeletons to study reef growth processes (Chazottes et al., 1995; Kiene and Hutchings, 1994; Silbiger et al., 2014; Tribollet and Golubic, 2005). Deployment of such blocks in a reef captures the endolithic and epilithic accretion and erosion agents and forces,

simultaneously allowing for the measurement of net-accretion and net-erosion rates. In particular, these studies have
provided insight into the colonization progression and activity of endolithic micro- and macroorganisms. To
comparatively assess the persistence of reef framework at the ecosystem scale, a census-based reef carbonate budget
(*ReefBudget*) approach that integrates reef site-specific ecological data into the calculation of the erosion-accretion
balance was introduced recently (Kennedy et al., 2013; Perry et al., 2012, 2015). Using the *ReefBudget* approach, a
recent study determined that 37 % of all current reefs that were investigated are in a net-erosive state (Perry et al.,
2013). For the Caribbean, it revealed a 50 % decrease of reef growth compared to historical mid- to late-Holocene
reef growth (Perry et al., 2013). Indeed, the use of carbonate budgets provided valuable insight into the reef growth
trajectories in the Seychelles, where surveys conducted since the 1990s provide important ecological baseline data
that were employed in reef growth calculations (Januchowski-Hartley et al., 2017). Most recently, carbonate budget
data were used to explore the relation of vertical reef growth potential and trends in sea level rise suggesting that reef
submergence poses a threat as long as climate-driven and human-made perturbations persist (Perry et al., 2018). Other
studies highlight the susceptibility of marginal coral reefs to ocean warming and acidification (Couce et al., 2012).
Such marginal reefs are found in the Eastern Pacific or in the Middle East in the Persian/Arabian Gulf, where reefs
exist at their environmental limits, e.g., at low pH or high temperatures, respectively (Bates et al., 2010; Manzello,
2010; Riegl, 2003; Sheppard and Loughland, 2002).
Although the Red Sea features high sea surface temperatures that exceed thermal thresholds of tropical corals
elsewhere (Kleypas et al., 1999; Osman et al., 2018), it supports a remarkable coral reef framework along its entire
coastline (Riegl et al., 2012). Yet, coral skeleton core samples indicate that calcification rates have been declining
over the past decades, which has been widely attributed to ocean warming (Cantin et al., 2010).  In this regard Red
Sea coral reefs are on a similar trajectory as other coral reefs under global ocean warming (Bak et al., 2009; Cooper
et al., 2008). In the central and southern Red Sea, present-day data show reduced calcification rates of corals and
calcifying crusts when temperatures peak during summer (Roik et al., 2015; Sawall et al., 2015). While increasing
temperatures are seemingly stressful and energetically demanding for reef calcifiers, high TA values, as found in the
Red Sea ($\sim$ 2400 $\mu$mol kg$^{-1}$, Metzl et al. 1989), are indicative of a putatively beneficial environment for calcification
(Albright et al., 2016; Langdon et al., 2000; Tambutté et al., 2011). At present, little is known about the reef-scale
carbonate budgets of Red Sea coral reefs (Jones et al., 2015). Apart from one early assessment of reef growth capacity
for a high-latitude reef in the Gulf of Aqaba (GoA, northern Red Sea) that considered both calcification and
bioerosion/dissolution rates (Dullo et al., 1996), studies only report calcification rates (e.g., Cantin et al., 2010; Heiss,
1995; Roik et al., 2015; Sawall and Al-Sofyani, 2015) or focus on bioerosion generally caused by one group of
bioeroders (Alwany et al., 2009; Kleemann, 2001; Mokady et al., 1996). Therefore, we set out to determine reef growth
in central Red Sea coral reefs and evaluate the biotic and abiotic drivers. We show and compare two reef growth
metrics: $G_{net}$ and $G_{budget}$. We present net-accretion/-erosion rates ($G_{net}$) measured *in situ* using limestone blocks
deployed in the reefs, which simultaneously capture the rates of epilithic accretion and epilithic and endolithic
bioerosion. We also apply a census-based approach adapted from the *ReefBudget* protocol (Perry et al., 2012) to
estimate reef growth on an ecosystem scale, as the net carbonate production state or carbonate budget ($G_{budget}$). Our
study provides a broad and first insight into reef growth dynamics and a comparative baseline to further assess the
effects of environmental change on reef growth in the central Red Sea.

## 2 Material and Methods

### 2.1 Study sites and environmental monitoring

Study sites are located in the Saudi Arabian central Red Sea along an environmental cross-shelf gradient, described in detail in Roik et al. (2015) and Roik et al. (2016). Data for this study were collected at three sites: an offshore forereef at ~25 km distance from the coastline (22° 20.456 N, 38° 51.127 E, "Shi'b Nazar"), midshore forereef at ~10 km distance (22° 15.100 N, 38° 57.386 E, "Al Fahal"), and a nearshore forereef (22° 13.974 N, 39° 01.760 E, "Inner Fsar") at ~3 km distance to shore. All sampling stations were located between 7.5 and 9 m depth. In the following, reef sites are referred to as "offshore", "midshore", and "nearshore", respectively. Abiotic variables were measured during "winter" and "summer" 2014. CTD data was collected continuously during "winter" (9th February - 7th April 2014) and "summer" (19th June - 23th October 2014). At each station, seawater samples were collected on SCUBA for 5 - 6 consecutive weeks during each of the seasons to determine inorganic nutrients, i.e., nitrate and nitrite ($NO_3^-$ & $NO_2^-$), ammonia ($NH_4^+$), phosphate ($PO_4^{3-}$), and total alkalinity (TA) (Table S1).

### 2.2 Net-accretion/-erosion rates of limestone blocks

Net-accretion/-erosion rates ($G_{net}$) were assessed using a "limestone block assay". Blocks cut from "coral stone" limestone were purchased from a local building material supplier in Jeddah, KSA. Each block was fixed with one stainless steel bolt to aluminum racks permanently deployed at the monitoring station of each reef site (a total of 36 blocks, n = 4, Fig. S1). The blocks were oriented in parallel to the reef slope with one side facing up while the other side was facing down towards the reef. Block dimensions were 100 x 100 x 21 mm with an average density of $\rho$ = 2.3 kg $L^{-1}$. Blocks were dry-weighed before and after deployment on the reefs (Mettler Toledo XS2002S, readability = 10 mg). Before weighing, the blocks were autoclaved and dried in a climate chamber (BINDER, Tuttlingen, Germany) at 40 °C for a week. Four replicate blocks were deployed at the reef sites for three different exposure periods each (Fig. 1 a) to measure natural processes of calcification and erosion. Exposure periods were 6 months (September 2012 - March 2013), 12 months (June 2013 - June 2014), and 30 months each (January 2013-June 2015). We measured a total of 12 blocks and all blocks were measured only once. Upon recovery, the blocks were treated with 10 % bleach for 24 - 36 h and rinsed with deionized water to remove organic material and any residual salts. $G_{net}$ were expressed as normalized differences of pre-deployment and post-deployment weights [kg $CaCO_3$ $m^{-2}$ $y^{-1}$] (Table 1).

### 2.3 Biotic parameters

To assess coral reef benthic calcifier and epilithic bioeroder communities (as input data for the reef carbonate budgets), we conducted *in situ* surveys on SCUBA along the cross-shelf gradient at each of our study sites.

### 2.3.1 Benthic community composition

Community composition and coverage of coral reef calcifying groups were assessed in six replicate transects per site using the belt-transect rugosity method (Perry et al., 2012) as detailed in Roik et al. (2015). From these surveys we extracted data on benthic calcifiers (% cover total hard coral, % hard coral morphs (branching, encrusting, massive, and platy/foliose), % major reef-building coral families (Acroporidae, Pocilloporidae, and Poritidae), % cover calcareous crusts, % recently dead coral, and % rock surface area for carbonate budget calculations (Table S2). In addition, benthic rugosity was assessed in the same transects following the *Chain and Tape Method* (n = 6, Perry et al., 2012).

### 2.3.2 Epilithic bioeroder/grazer populations along the cross-shelf gradient

For each reef site, we surveyed abundances and size classes of the two main groups of coral reef framework epilithic bioerorders, parrotfishes (Scaridae) (Bellwood, 1995; Bruggemann et al., 1996) and sea urchins (Echinoidea) (Bak, 1994). Surveys were conducted on SCUBA using stationary plots (adapted from Bannerot and Bohnsack, 1986, Text S1) and line transects (n = 6 per site), respectively. Briefly, abundances of parrotfishes and sea urchins were assessed for different size classes. Abundances for all prevalent parrotfish species were assessed in six size classes, based on estimated fork length (FL; FL size classes: 1 = 5 - 14 cm, 2 = 15 - 24 cm, 3 = 25 - 34 cm, 4 = 35 - 44 cm, 5 = 45 - 70, and 6 > 70 cm). We focused on the most abundant bioeroding parrotfish species in the Red Sea (Table S4), which encompassed two herbivorous functional groups: excavators and scrapers (Green and Bellwood 2009). Most abundant across study sites were the excavators *Chlorurus gibbus*, *Scarus ghobban*, and *Cetoscarus bicolor*, and the scrapers *Scarus frenatus*, *Chlorurus sordidus*, *Scarus niger* and *Scarus ferrugenius*, following Alwany et al., 2009. Additionally, we counted *Hipposcarus harid*, which occurred frequently at the study sites, along with members of the genus *Scarus* that could not be identified to species level and were therefore pooled in the category 'Other *Scarus*'. Both *H. harid* and *Scarus* spp. were broadly categorized as scrapers (Green and Bellwood, 2009). The sea urchin census targeted five size classes of the four most common bioerosive genera *Diadema*, *Echinometra*, *Echinostrephus*, and *Eucidaris*, based on urchin diameter (size classes 1 = 0 - 20 mm, 2 = 21 - 40 mm, 3 = 41 - 60 mm, 4 = 61 - 80 mm, 5 = 81 - 100 mm, Table S7). For details on the field surveys and data treatment for biomass conversion, refer to the supplementary materials (Text S1 and references therein).

### 2.4 Reef carbonate budgets

Ecosystem scale reef carbonate budgets, $G_{budget}$ [kg $CaCO_3$ m$^{-1}$ y$^{-1}$], were determined following the census-based *ReefBudget* approach by Perry et al. (2012) (Table 1). $G_{budget}$ incorporates local census data, site-specific net-accretion/-erosion data ($G_{net}$ over 30 months) and calcification data (buoyant weight measurements) collected for this and a previous study (Roik et al., 2015). Importantly, the approach incorporates epilithic bioerosion, which is based on abundance rather than bite or erosion rates; therefore, parrotfish and sea urchin census data collected in this study were employed in the *ReefBudget* calculations using bite and erosion rates from the literature (Alwany et al., 2009;

Perry et al., 2012). In summary, site-specific benthic calcification rates ($G_{benthos}$, kg $CaCO_3$ $m^{-1}$ $y^{-1}$), net-accretion/-
erosion rates of reef "rock" surface area ($G_{netbenthos}$, kg $CaCO_3$ $m^{-1}$ $y^{-1}$), and epilithic erosion rates by sea urchins ($E_{echino}$,
kg $CaCO_3$ $m^{-1}$ $y^{-1}$) and parrotfishes ($E_{parrot}$, kg $CaCO_3$ $m^{-1}$ $y^{-1}$), were determined for the $G_{budget}$ calculations (Fig. 1 (b)
and Fig. 3 (a)). A detailed account of Red Sea specific calculations and modifications of the *ReefBudget* approach
employed in this study are outlined in the supplementary materials (Text S1, Equation box S1-3, and Tables S2-S8).

**2.5 Abiotic parameters**


**2.5.1 Continuous data: temperature, salinity, and diurnal pH variation**


Factory-calibrated conductivity-temperature-depth loggers (CTDs, SBE 16plusV2 SEACAT, RS-232, Sea-Bird
Electronics, Bellevue, WA, USA) were deployed at the monitoring stations on tripods at ~0.5 m above the reef to
collect time series data of temperature, salinity, and $pH_{NBS}$ at hourly intervals. The pH probes (SBE 18/27, Sea-bird
Electronics) were factory calibrated before the winter deployment (9th February - 7th April 2014). Calibrations were
verified using NBS scale standard buffers (pH 7 and 10, Fixanal, Fluka Analytics, Sigma Aldrich, Germany) before
the winter and the summer deployment (19th June - 23th October 2014).

**2.5.2. Seawater samples: Inorganic nutrients and total alkalinity**


Seawater samples were collected on SCUBA at each of the stations using 4 L collection containers (Table S1).
Simultaneously, 60 mL seawater samples were taken through a 0.45 µm syringe filter for TA measurements. Seawater
samples for inorganic nutrient analyses and TA measurements were transported on ice in the dark and were processed
on the same day. Samples were filtered over GF/F filters (0.7 µm, Whatman, UK) and filtrates were frozen at -20 °C
until analysis. The inorganic nutrient content ($NO_3^-$ & $NO_2^-$, $NH_4^+$, and $PO_4^{3-}$) was determined using standard
colorimetric tests and a Quick-Chem 8000 AutoAnalyzer (Zellweger Analysis, Inc.). TA samples were analyzed
within 2 - 4 h after collection using an automated acidimetric titration system (Titrando 888, Metrohm AG,
Switzerland). Gran-type titrations were performed with a 0.01 M HCl (prepared from 0.1 HCl Standard, Fluka
Analytics) at an average accuracy of $\pm$ 9 µmol $kg^{-1}$ (standard deviation of triplicate measurements).

**2.6 Statistical analyses**


**2.6.1 Net-accretion/-erosion rates and carbonate budgets**


$G_{net}$ data (Table 2) were tested for effects of the factors "reef" (fixed factor: nearshore, midshore, and offshore) and
"deployment time" (random factor: 6, 12, and 30 months). A univariate 2-factorial PERMANOVA was performed on
$log_n(x)$ transformed data (i.e., $log_n(x+1-min(x_{1-n}))$ as data contained negative and near-zero values). A Euclidian
distance matrix and 9999 permutations of residuals under a reduced model and type III partial sum of squares were
employed. Pair-wise tests followed where applicable (PRIMER-E V6, Table S9).
$G_{budget}$ data (Table 3) were tested for statistical differences between the reef sites (fixed factor: nearshore, midshore,
and offshore) using a 1-factorial ANOVA. In parallel, $G_{benthos}$ was tested using a 1-factorial ANOVA with $log_{10}$
transformed data, while non-parametric Kruskal-Wallis tests were employed for non-transformed $G_{netbenthos}$, $E_{echino}$,
and $E_{parrot}$ data. Tukey's HSD post-hoc tests or Dunn's multiple comparisons followed where applicable (Table S10).
Assumptions about parametric distribution of data were evaluated using the Shapiro-Wilk normality test. Statistical
tests were performed as implemented in R (R Core Team, 2013).

### 2.6.2 Abiotic parameters

All abiotic data were summarized as means and standard deviations per reef and season and over each season (Table
4) and boxplots were generated (Fig. 4). Diurnal pH variation was extracted from the continuous data as the $pH_{NBS}$
standard deviation per day. Outliers were detected and removed from the TA data. All outliers (data points beyond
the upper boxplot 1.5 IQR) clustered to one sampling day (23 June 2014), which we considered an artifact of the
chemical analysis and the outliers from this day were removed. All continuous abiotic variables and inorganic nutrients
($PO_4^{3-}$ after square-root transformation) fulfilled parametric assumptions and were evaluated using univariate 2-
factorial ANOVAs testing the factors "reef" (nearshore, midshore, and offshore) and "season" (winter and summer).
TA data was square-root transformed, which improved symmetry of data (Anderson et al., 2008), and tested under the
same 2-factorial design, as outlined above, using a PERMANOVA (Euclidian resemblance matrix and 9999
permutations of residuals under a reduced model and type II partial sums of squares). Within each significant factor,
Tukey's HSD post-hoc tests or PERMANOVA integrated pair-wise tests followed (Table S11 and S12). Assumptions
were evaluated by histograms and the Shapiro-Wilk normality test. Statistical tests and outlier detection were
performed in R or PRIMER-E V6.

### 2.6.3 Abiotic-biotic correlations

To evaluate the relationship of abiotic and biotic predictors of $G_{net}$ and $G_{budget}$, Spearman rank correlation coefficients
were obtained for the predictor variables (at a confidence level of 95%) using *cor.test* in R (R Core Team, 2013;
Wickham and Chang, 2015). *P*-values were adjusted using *p.adjust* in R employing the Benjamini-Hochberg method.
Correlations were performed using $G_{net}$ data obtained in the 30-months measurements from the reef sites (nearshore,
midshore, and offshore) (Table 5 and Table S13). Predictor variables were the site-specific means of CTD measured
variables (temperature, salinity, and diurnal pH variation), means of inorganic nutrients ($NO_3^-$&$NO_2^-$, $NH_4^+$, and $PO_4^{3-}$
), and TA (Table 4). Biotic predictors were variables that likely impacted the limestone blocks, i.e. parrotfish
abundances, sea urchin abundances, calcareous crusts cover, and algal and sponge cover. Since we did not observe
any coral recruits of substantial size on the blocks, we did not include % coral cover and related variables in the
correlations.
$G_{budget}$ correlations included all the above-mentioned abiotic variables and 13 biotic transect variables (i.e., parrot fish
abundances, sea urchin abundances, % branching coral, % encrusting coral, % massive coral, % platy/foliose coral,
% of Acroporidae, % Pocilloporidae, % Poritidae, % total hard coral cover, calcareous crusts cover, algal and sponge
cover, and rugosity). Prior to analysis, some of the predictors (i.e., % platy/foliose corals and % Poritidae) were
$\log_{10}(x+1)$ transformed to improve the symmetry in their distributions (Table 5 and Table S14).

**3 Results**

**3.1 Net-accretion/-erosion rates of limestone blocks**

Net-accretion/-erosion rates $G_{net}$ were measured in assays over periods of 6, 12, and 30 months in the reef sites along the cross-shelf gradient. These measurements represent the result of calcification and bioerosion processes impacting the deployed limestone blocks. Visible traces of boring endolithic fauna were only found on the surface of blocks recovered after 12 and 30 months as presented in Fig. 2 (c)-(f). A brief visual inspection of the block surfaces after retrieval showed colonization by coralline algae, bryozoans, boring sponges, small size boring worms and clams, as well as parrotfish bite-marks. No coral recruits were noticed by the unaided eye. Further analyses of the established presence of calcifying and bioeroding communities were not within the scope of this study. $G_{net}$ based on the 30-months deployment of blocks ranged between -0.96 and 0.37 kg $CaCO_3$ m$^{-2}$ y$^{-1}$ (Table 2). $G_{net}$ for 12 and 30-months blocks were negative on the nearshore reef (between -0.96 and -0.6 kg $CaCO_3$ m$^{-2}$ y$^{-1}$, i.e., net erosion is apparent), slightly positive on the midshore reef (0.01 - 0.06 kg $CaCO_3$ m$^{-2}$ y$^{-1}$, i.e., almost neutral carbonate production state), and positive on the offshore reef (up to 0.37 kg $CaCO_3$ m$^{-2}$ y$^{-1}$, i.e., net accretion of reef framework). Deployment times had a significant effect on the variability of $G_{net}$ (Pseudo-$F$ = 5.9, $p_{PERMANOVA} < 0.01$, Table S9). As expected, accretion/erosion was overall higher when measured over the longer deployment period (Fig. 2 (g)) in comparison to the shorter deployment times, reflecting the continuous and exponential nature of bioerosion due to the colonization progress of fouling organisms over time. The significant interaction of reef site and deployment time (Pseudo-$F$ = 7.3, $p_{PERMANOVA} < 0.001$) shows that only blocks deployed over 12 and 30 months revealed significant site variability, specifically the differences between nearshore vs. offshore and midshore vs. offshore sites became evident ($p_{pair-wise} <$ 0.05, Table S9). The within-group variability was highest for the nearshore reef, where standard deviations were up to 7-times higher compared to the midshore and the offshore reefs.

**3.2 Biotic parameters**

**3.2.1 Benthic community composition**

A detailed account of benthic community structure of the study sites is provided in Roik et al. (2015). In brief, a low percentage of live substrate (20 %) and calcifier community cover (hard corals = 11 % and calcifying crusts = 1 %) were characteristic for the nearshore site, while rock (23 %) and rubble (4 %) were more abundant compared to the other sites. The midshore and offshore reefs provided live benthos cover of around 70 % and a large proportion of calcifiers (48 and 59 %). The proportion of coral and calcifying crusts, which were dominated by coralline algae, were 38 % and 10 % in the midshore reef compared to 35 % and 23 % in the offshore reef, respectively. Major reef-building coral families were Acroporidae, Pocilloporidae, and Poritidae forming 32 - 56 % of the total hard coral cover. A soft coral community (of around 25 %) occupied large areas in the midshore reef. This community was minor in the nearshore and offshore reefs with 4 % and 8.5 %, respectively. Specific benthic accretion rates $G_{benthos}$ [kg $CaCO_3$ m$^{-2}$ y$^{-1}$], which were used as input data for the $G_{budget}$ calculation, were determined using these benthic data in addition to site and calcifier specific calcification rates (Tables S2 and S3).


### 3.2.2 Epilithic bioeroder/grazer populations along the cross-shelf gradient

A total of 718 parrotfishes and 110 sea urchins were observed and included in subsequent *ReefBudget* analyses.
Parrotfish mean abundances and biomass estimates ranged between $0.08 \pm 0.01$ and $0.17 \pm 0.60$ individuals m$^{-2}$, and
$24.69 \pm 6.04$ and $82.18 \pm 46.67$ g m$^{-2}$, respectively (Table S4). The largest parrotfish (category 5 parrotfish, i.e., $> 45$
- 70 cm fork length) were observed at the midshore site. With the exception of the midshore reef, category 1 (5 - 14
cm) parrotfish were commonly observed at all sites. Large parrotfish (category 6 with $> 70$ cm fork length) were not
observed during the surveys. For sea urchins, mean abundances of $0.002 \pm 0.004$ - $0.014 \pm 0.006$ individuals m$^{-2}$ per
site were observed and mean biomasses $0.05 \pm 0.04$ - $1.43 \pm 0.98$ g m$^{-2}$ estimated per site, respectively (Table S7).
The midshore site exhibited the largest range of sea urchin size classes (from categories 1 or 2 to the largest size class
5), while at the other two exposed sites, only the two smallest size classes of sea urchins were recorded.

### 3.3 Reef carbonate budgets

The carbonate budget, $G_{budget}$, averaged over all sites was $0.66 \pm 2.01$ kg CaCO$_3$ m$^{-2}$ y$^{-1}$ encompassing values ranging
from a negative nearshore budget ($-1.48 \pm 1.75$ kg CaCO$_3$ m$^{-2}$ y$^{-1}$) to a positive offshore budget ($2.44 \pm 1.03$ kg CaCO$_3$
m$^{-2}$ y$^{-1}$) (Figure 3 and Table 3). $G_{budget}$ significantly differed between reef sites ($F = 16.7$, $p_{ANOVA} < 0.001$, Table S10),
where nearshore vs. offshore site and midshore vs. offshore site showed significant differences ($p_{Tukey\ HSD} < 0.01$).
Further, biotic variables that contribute to the final $G_{budget}$ were diverse: $G_{benthos}$ significantly varied between midshore
vs. nearshore site and offshore vs. nearshore site ($p_{Tukey\ HSD} < 0.01$), $G_{netbenthos}$ varied between all site combinations
($p_{Tukey\ HSD} < 0.001$), $E_{echino}$ significantly differed between midshore and nearshore, and $E_{parrot}$ variability was similar at
all sites. The within-group variation for the nearshore reef was 5-times higher compared to the midshore reef and the
offshore reef. Overall, the proportional loss of accreted carbonate to bioerosion was 15 % in the offshore reef, 42 %
in the midshore reef, and the loss even exceeded the accretion by four-fold in the nearshore reef, i.e., ~ 440 %
proportional loss when considering accreted carbonate to bioerosion.

### 3.4 Abiotic parameters

### 3.4.1 Temperature, salinity, and diurnal pH variation

We used abiotic monitoring data to characterize environmental conditions at each reef site throughout the year (Table
1, Table S11 and S12). Temperature and salinity comprised ~4400 data points per reef site in the nearshore and
offshore reef, and ~2700 in the midshore reef; diurnal pH standard deviations comprised 185 data points for the
midshore and offshore site, and 87 for the nearshore site. The seasonal mean temperature varied between $26.1 \pm 0.5$
°C in winter and $30.9 \pm 0.7$ °C in summer across all reefs. The cross-shelf difference was largest in summer (~0.6 °C),
and significant during both seasons ($F = 1042.6$, $p_{ANOVA} < 0.001$). From all sites, the nearshore site experienced the
lowest mean temperature (26.1 °C) in winter and the highest (31.3 °C) in summer. In comparison, the midshore and
offshore reefs were slightly cooler with means around 30.6 °C during summer. Overall salinity was high ranging
between 39.18 – 39.44 over the year. In summer nearshore salinity was significantly increased by 0.36 compared to
winter and by 0.18 compared to the other reefs ($F = 945.3$, $p_{ANOVA} < 0.001$). Salinity in the midshore and offshore reef
was not significantly different between the two sites. Mean diurnal standard deviations of pH ranged between 0.04 –
0.07 of pH units in the midshore and offshore reefs. The nearshore reef experienced the largest diurnal variations as
indicated by mean diurnal standard deviations of 0.29 pH units during winter and 0.6 pH units during summer. The
diurnal pH fluctuation differed significantly between all reef sites ($F = 1241$ $p_{ANOVA} < 0.001$).

**3.4.2 Seawater samples: Inorganic nutrients and total alkalinity**

Concentrations of all measured inorganic nutrients were below 1 µmol kg$^{-1}$ (Table 1). $NO_3^-$&$NO_2^-$ was on average
between $0.63 \pm 0.26$ and $0.28 \pm 0.22$ µmol kg$^{-1}$, $NH_4^+$ between $0.51 \pm 0.17$ and $0.35 \pm 0.19$ µmol kg$^{-1}$, and $PO_4^{3-}$ as low
as $0.02 \pm 0.01$ and $0.09 \pm 0.02$ µmol kg$^{-1}$ (the highest and lowest site-season averages are reported here). By trend,
mean $NO_3^-$&$NO_2^-$ and $NH_4^+$ levels were higher in winter compared to summer with a difference of 0.29 and 0.16 µmol
kg$^{-1}$, respectively (Fig. 4, Table S11 and S12). In contrast, $PO_4^{3-}$ was significantly higher in winter than in summer
with means differing on average by 0.04 µmol kg$^{-1}$ ($F = 16$ , $p_{ANOVA} < 0.001$, Table S11). Mean differences across the
shelf were 0.1 µmol kg$^{-1}$ in $NO_3^-$&$NO_2^-$ during winter, 0.1 µmol kg$^{-1}$ in $NH_4^+$ during summer, and 0.02 µmol kg$^{-1}$ in
$PO_4^{3-}$ throughout both seasons. TA ranged between $2391 \pm 15$ and $2494 \pm 16$ µmol kg$^{-1}$. TA was significantly different
between seasons and reef sites (Pseudo-$F_{season}$ = 297.6, Pseudo-$F_{reefsite}$ = 22.5, $p_{PERMANOVA} < 0.001$, Table S11 and
S12). During both seasons, TA was decreasing from the offshore to the nearshore reef. During winter, TA was slightly
higher with $2487 \pm 20$ µmol kg$^{-1}$ compared to $2417 \pm 27$ µmol kg$^{-1}$ during summer. The increase from nearshore to
offshore was on average between 20 and 50 µmol kg$^{-1}$ (Fig. 4).

**3.5 Abiotic-biotic correlations**

To explore the relationship between environmental variables and reef growth, we performed correlation analyses. For
$G_{net}$, strong, positive, and significant correlates were calcareous crust cover, $NO_3^-$&$NO_2^-$, $PO_4^{3-}$, and TA. Negative
correlates were salinity, diurnal pH variation, and parrotfish abundance (strong correlates: $\rho > |0.75|$, $p < 0.001$). For
$G_{budget}$, abiotic correlates were $NO_3^-$&$NO_2^-$, $PO_4^{3-}$, and TA, the same correlates as for $G_{net}$. Looking at significant biotic
correlates of $G_{budget}$, we only found positive relationships, including calcareous crusts, hard corals, and rugosity.
Conversely, parrotfish and sea urchin abundances had a negative effect on $G_{budget}$, but the correlation was weak and
not significant ($\rho \sim -0.5$). The non-calcifying benthos, which represents the coverage by algae, soft corals, and sponges,
was not correlated with the dynamics of $G_{budget}$ and was correlated only weakly and not significantly with $G_{net}$ ($\rho \sim$
0.5) (Table 5, Tables S13 and S14).

## 4 Discussion

Central Red Sea reefs are characterized by unique environmental conditions of high temperature, salinity, TA, and oligotrophy (Fahmy, 2003; Kleypas et al., 1999; Steiner et al., 2014). On a global scale they support remarkable reef growth, supporting well established fringing reefs along most of the coastline. To date, processes affecting reef growth in various regions of the Red Sea have mostly been investigated individually. For instance, some studies focused on bioerosion by one specific group of bioeroders only (Alwany et al., 2009; Kleemann, 2001; Mokady et al., 1996), while other studies assessed calcification of reef-building corals (e.g., Cantin et al., 2010; Heiss, 1995; Roik et al., 2015; Sawall et al., 2015). To provide a more comprehensive picture, the present study integrated assessment of the antagonistic processes of calcification and bioerosion. We achieved this in a two-step approach assessing two central metrics of reef growth along a cross-shelf gradient. First, we assessed net-accretion/-erosion rates ($G_{net}$) from three reef sites along the cross-shelf gradient *in situ* using a limestone block assay. Second, we constructed ecosystem-scale estimates of reef carbonate budgets for Red Sea reef sites ($G_{budget}$) adapting the census-based *ReefBudget* approach by Perry et al. (2012). In the following, we highlight the complex dynamics and interactions of reef growth processes and discuss the importance of carbonate budgets as a powerful tool to explore the trajectories of reef growth in a global and historical context.

### 4.1 Net-accretion/-erosion rates ($G_{net}$) in the central Red Sea

### 4.1.1 Cross-shelf dynamics in a global context

The limestone block assay revealed three reef production states in the central Red Sea: 1) net erosion (nearshore), 2) near-neutrality (midshore), and 3) net accretion (offshore). This is in contrast to the pattern observed on the Great barrier reef (GBR), where total bioerosion rates were higher in offshore reefs than inshore reefs as assessed from limestone blocks (Tribollet et al., 2002; Tribollet and Golubic, 2005). Generally, most block assay studies conducted in various reef habitats and regions found net-erosive rates. For instance, studies from reefs in the Thai Andaman Sea and Indonesian Java Sea note that the accretion by calcifying crusts, such as coralline algae, were negligible compared to the high degree of bioerosion measured in the limestone blocks (Edinger et al., 2000; Schmidt and Richter, 2013). In contrast, our limestone block assays captured a substantial net accretion rate, in particular for the offshore reef site in the central Red Sea (0.37 kg $CaCO_3$ $m^{-2}$ $y^{-1}$ net accretion), indicating that accretion was substantial, while erosion was negligible. The midshore reef was characterized by a near-neutral or minor net accretion (0.06 kg $CaCO_3$ $m^{-2}$ $y^{-1}$) on the order of net accretion rates recorded in French Polynesia in reef sites of uninhabited, oceanic atolls (0.08 and 0.62 kg $CaCO_3$ $m^{-2}$ $y^{-1}$; Pari et al., 1998). Notably, our study recorded a net-erosive state only in the Red Sea nearshore site (-0.96 kg $CaCO_3$ $m^{-2}$ $y^{-1}$, 30 months deployment). This is a moderate rate compared to the larger net erosion observed in the GBR, French Polynesia, and Thailand (-4 or -8 kg $CaCO_3$ $m^{-2}$ $y^{-1}$) (Osorno et al., 2005; Pari et al., 1998; Schmidt and Richter, 2013; Tribollet and Golubic, 2005).

**4.1.2 Limestone block deployment duration and biotic drivers**

Our data show that $G_{net}$ values were overall higher with longer deployment times, reflecting the succession and early establishment of calcifying crust and bioeroding communities on the limestone blocks. Due to our sampling design (weight-based block assay), accretion and erosion processes however are simultaneously captured and cannot be disentangled. Overall, the block assay data are indicative of a calcifier-beneficial offshore environment and a nearshore reef habitat that is supporting endolithic bioeroders.

Following other work, carbonate loss in the 12-month blocks from the nearshore site was supposedly due to a young microbioeroder community, which is typically most active during this early phase. For instance, during the early stages of colonization by endolithic microorganisms, the chlorophyte *Ostreobium* sp. predominantly contributes to microbioerosion, while the erosion rate steadily increases with deployment time (Grange et al., 2015; Tribollet and Golubic, 2011). Microbioerosion rates have been reported to be -0.93 kg $CaCO_3$ $m^{-2}$ $y^{-1}$ after 12 months of block exposure, which represents the average rate at the early colonization stage when the steadily increasing microbioerosion rate has leveled off (Grange et al., 2015). This rate is slightly higher compared to our measurements of net erosion in the nearshore site after the same deployment time (i.e. -0.61 kg $CaCO_3$ $m^{-2}$ $y^{-1}$), and the difference may reflect measurements encompassing both, bioerosion and accretion.

Studies have shown that site differences in total bioerosion are typically becoming visible after 1 year of deployment and are significantly enhanced after 3 years (Tribollet and Golubic, 2005). In line with this, the deployment time of 12 months in our study was sufficient to reveal differences between the nearshore and offshore reef sites. Further, calcifying crusts, specifically coralline algae, observed on all blocks from the offshore reef contributed to the respective net accretion. This is corroborated by the positive correlation of their abundances with $G_{net}$ across all reef sites. Given that we could not identify coral recruits on any limestone block, we assume that contribution of corals to the measured accretion was minor. However, we acknowledge that we might have missed some that could be detected by more sophisticated methods (e.g. such as microscopic examination).

Significant differences in accretion/erosion between all three sites of the cross-shelf gradient became apparent after 30 months deployment, and macroborer traces were observed in blocks for the first time (Fig. 2). Over the course of 2 - 3 years, macrobioeroders such as polychaetes, sipunculids, bivalves, and boring sponges can establish communities in limestone blocks (Hutchings, 1986). Between the first two years, macrobioeroder contribution to the total bioerosion can quadruple (0.02 - 0.09 kg $CaCO_3$ $m^{-2}$ $y^{-1}$), before levelling off around 3 - 4 years post-deployment (Chazottes et al., 1995).

In our study, the increase of $G_{net}$ between the 12- and 30-month deployment (~0.30 kg $CaCO_3$ $m^{-2}$ $y^{-1}$ on average in the nearshore and offshore site) indicates that calcifying and eroding communities were still in a state of succession. As such, we cannot unequivocally rule out that the blocks deployed for 30 months still represented an immature community, and hence, underestimated maximal calcification and erosion rates.

Correlation analyses indicate a significant contribution of parrotfish to the net erosion rates in the nearshore reef. This observation is in line with previous work demonstrating a significant contribution of parrotfish activity to bioerosion (Alwany et al., 2009; Bellwood, 1995; Bellwood et al., 2003). By comparison, sea urchin size and abundance do not appear to be significant for bioerosion on the central Red Sea reefs. On other reefs, sea urchin bioerosion can be

substantial, equaling or even exceeding reef carbonate production (Bak, 1994). The low contribution of sea urchins to
bioerosion on central Red Sea reefs may be a result of potentially low abundances of highly erosive sea urchins
(McClanahan and Shafir, 1990). This is in line with the observed parrotfish bite-marks and a lack of sea urchins on
and in the direct vicinity of the recovered blocks. Taken together, our data confirms that endolithic micro- and
macrobioerosion, as well as parrotfish feeding, likely provides a substantial contribution to calcium carbonate loss.

**4.2 Carbonate budgets ($G_{budget}$) in the central Red Sea**
**4.2.1 Cross-shelf dynamics, regional and global context**
On an ecosystem scale, the $G_{budget}$ data suggest that the offshore reef site in the central Red Sea loses about 15 %
accreted carbonates to bioerosion per year. On the mid- and nearshore reef this loss increases to to 42 % and to over
100 %, respectively. By comparison, on the scale of a single coral colony, the boring clam *Lithophaga lessepsiana*
alone can erode up to 40 % of the carbonate deposited by the coral *Stylophora pistillata* (Lazar and Loya, 1991). In
our study sites, the spatial dynamics of the two metrics $G_{net}$ and the census-based $G_{budget}$, were consistent and suggest
net erosion in nearshore reef sites and net accretion in offshore reef sites in the central Red Sea. Reef growth at the
central Red Sea cross-shelf gradient averaged $0.66 \pm 2.01$ kg $CaCO_3$ m$^{-2}$ y$^{-1}$, which was driven by the substantial
budget of the offshore reef, reflecting the location and habitat dependence for reef growth potential. That the offshore
reef budget is essential to maintain the entire shelf budget has also has been observed on a reef platform in the
Maldives. In the respective study, reef accretion was minor and highly heterogeneous at most sites and only few reef
sites at the platform margin promoted substantial net accretion and thereby greatly contributed to the positive average
budget of the entire platform (Perry et al., 2017).
The here presented central Red Sea $G_{budget}$ data are within the range of contemporary reef carbonate budgets from the
Atlantic ($2.55 \pm 3.83$ kg $CaCO_3$ m$^{-2}$ y$^{-1}$) and Indian Ocean ($1.41 \pm 3.02$ kg $CaCO_3$ m$^{-2}$ y$^{-1}$) (Perry et al., 2018). Notably,
these data are below the suggested "optimal reef budget" of 5 - 10 kg $CaCO_3$ m$^{-2}$ y$^{-1}$ observed in "healthy", high coral
cover fore-reefs (see data in Perry et al., 2018 and comparisons therein; Vecsei, 2001, 2004). The decline in coral
cover is likely central to the reduced carbonate budgets in contemporary reefs. For instance, the reefs investigated in
the present study do not exceed a coral cover of 40 % (as observed in the offshore study site). In comparison, the
dataset compiled by Vecsei 2001 encompasses hard coral cover of up to 80% for the Indo-Pacific and up to 95 % for
Pacific Islands. Further, the reduced contemporary carbonate budgets coincide with the observed decrease in
calcification rates of Red Sea corals at large (Cantin et al., 2010; Steiner et al., 2018). As such, the effect of climate
change and the corresponding increase in seawater temperature may have severe consequences via overall decrease
in coral reef cover as well as via reduced calcification of the resident corals. Hence, although the present $G_{budget}$ data
still suggest effective barrier reef formation in the central Red Sea (substantial accretion on the offshore reef),
carbonate accretion rates and therefore reef formation in the central Red Sea may be hampered in the long run by the
ongoing warming.

**4.2.2 Biotic drivers**
*Regional differences*
Cross-shelf patterns of $G_{budget}$ drivers from the central Red Sea are distinct from other reef systems. The central Red
Sea system is characterized by a nearshore site with a negative $G_{budget}$, impacted by high parrotfish abundances and
erosion rates, low coral cover, and putatively considerable endolithic bioerosion rates (see discussion of $G_{net}$ data).
Conversely, the offshore reef is characterized by high calcification rates, driven by high coral and coralline algae
abundances. In the GBR an opposing trend with high net accretion in the nearshore reefs (Browne et al., 2013)
coincided with high coral cover, low bioerosion rates, and lowest rates of parrotfish bioerosion (Hoey and Bellwood,
2007; Tribollet et al., 2002). On Caribbean reefs, parrotfish erosion rates were higher on leeward reefs (which may be
similar to protected nearshore habitats), but in contrast to the central Red Sea, these sites were typically characterized
by overall high coral cover driving a positive $G_{budget}$ (Perry et al., 2012, 2014). This inter-regional comparison strongly
suggests that reef accretion/erosion dynamics encountered in any given reef system cannot be readily extrapolated to
other reef systems. Hence, *in situ* assessments of individual reef systems are required to unravel local dynamics and
responses to environmental change, and are therefore imperative for the development of effective management
measures.

*The role of coral and coralline crusts*
Benthic calcifiers, in particular reef-building corals, are major contributors to carbonate production and are considered
the most influential drivers of $G_{budgets}$ globally (Franco et al., 2016). Corals in particular can contribute as much as 90
% to the gross carbonate production across different reef zones, which also includes low coral cover lagoonal and
rubble habitats (Perry et al., 2017). Hence, loss of coral cover rapidly gives way to increased bioerosion and thereby
critically contributes to reef framework degradation (Perry and Morgan, 2017). Indeed, on Caribbean reefs, $G_{budget}$
data were reported to shift into erosional states once live hard coral cover was below 10 % (Perry et al., 2013). A live
coral cover threshold remains to be determined for the central Red Sea and will require evaluation of a larger dataset.
Yet, we find that the nearshore reef featuring a negative $G_{budget}$ is characterized by a coral cover of 11 %, while the
midshore and offshore reefs, characterized by near-neutral vs. positive carbonate budgets, both feature similar average
coral covers (at 35 and 40 %, respectively). In this respect, our data show that a 2-fold higher abundance of coralline
algae and other encrusting calcifiers in the offshore reef (compared to the midshore reef) significantly added to a
higher $G_{budget}$. The positive contribution of coralline algae for central Red Sea reef accretion is corroborated by their
strong and significant correlation to $G_{budget}$. Coralline algae in particular are considered an important contributor to
reef growth, as they stabilize the reef framework through "cementation" (Perry et al., 2008) and by habitat priming
for successful coral recruitment (Heyward and Negri, 1999).

*Epilithic grazers*
Epilithic grazers such as parrotfish and sea urchin are considered important drivers of bioerosion on many reefs (Hoey
and Bellwood, 2007; Mokady et al., 1996; Pari et al., 1998; Reaka-Kudla et al., 1996). Sea urchins were identified as
significant bioeroders in some reefs of Réunion Island, French Polynesia, and in the GoA, northern Red Sea (Chazottes
et al., 1995, 2002; Mokady et al., 1996). For the northern Red Sea, sea urchins were abundant, and their removal of
reef carbonates was estimated to range around 13 - 22 % of total reef slope calcification (Mokady et al., 1996). In
contrast, sea urchins were rare in our study sites contributing to only 2 - 3 % of the total bioerosion resulting in low
contributions to $G_{budget}$. Only on the net-erosive nearshore reef were sea urchins more abundant causing 12 % of total
bioerosion.
Compared to sea urchins, parrotfish played a more important role for $G_{budgets}$ throughout the entire reef system,
contributing 70 - 96 % of the total bioerosion. In the correlation analyses, both grazers, i.e., sea urchins and parrotfish,
negatively correlated with $G_{budget}$, however these correlations were not very strong ($\rho \sim$ -0.5) and non-significant. The
weak correlation may be influenced by a considerable variability in the reef census dataset, specifically regarding
parrotfish abundances. Observer bias (parrotfish keep minimum distance from surveyors during dives and may
therefore not enter survey plots; pers. obs.), natural (e.g., species distribution, habitat preferences, reef rugosity, and
mobility or large roving excavating species, such as *Bolbometopon muricatum*), and/or anthropogenically-driven
factors (e.g., differential fishing pressure) may also contribute to the observed data heterogeneity (McClanahan, 1994;
McClanahan et al., 1994). Indeed, the Saudi Arabian central Red Sea has been subject to decade-long fishing pressure,
which has significantly altered reef fish community structures and reduced overall fish biomass compared to less
impacted Red Sea regions (Kattan et al., 2017). Unregulated fishing could at least in part explain the differences of
fish abundance dynamics between the present study and reefs on the GBR and the Caribbean. The heterogeneity of
grazer populations further propagates into $G_{budgets}$ estimates, resulting in a considerable within-site variability that
reduces power of statistical tests and correlations.
**4.3 Abiotic factors and reef growth dynamics**
Reef habitats in the central Red Sea are characterized by abiotic factors that differ from the majority of tropical reef
environments (Couce et al., 2012; Kleypas et al., 1999). Our sites were exposed to high summer temperatures (30 -
33 °C) and a high salinity throughout the year (39 - 40). Inorganic nutrients were mostly far below 1 µmol kg$^{-1}$,
whereas TA was comparably high, 2400 – 2500 µmol kg$^{-1}$, values typical for much of the Red Sea basin (Acker et al.,
2008; Steiner et al., 2014). As such, the Red Sea is considered a natural model system or "laboratory", which can
advance our understanding of ecosystem functioning under extreme or marginal conditions of which some are
projected under ocean change scenarios (Camp et al., 2018). The study of such natural systems is a challenge and the
documentation of governing factors both abiotic and biotic will contribute to a better understanding of the dynamics
and interactions, which can significantly improve ecosystem scale predictions (Boyd and Hutchins, 2012; Boyd and
Brown, 2015; Camp et al., 2018). In the present study, reef framework decline (i.e., net erosion) was associated with
the reef habitat of slightly increased salinity and stronger diel pH fluctuations, which are characteristic for shallow
water, limited flow systems and semi-enclosed reefs (Camp et al., 2017; Shamberger et al., 2017), such as the here
investigated nearshore study site (Roik et al., 2016). On the other hand, positive reef growth was associated with reef
habitats characterized by higher TA levels, but also with slightly increased inorganic nutrient species, namely $NO_3^-$
$\&NO_2^-$ and $PO_4^{3-}$.

*The nearshore site*
The nearshore reef is located on the shelf, surrounded by shallow waters of extended residency time and has a lower
water exchange rate compared to the other two reef sites (Roik et al., 2016). Evaporation and limited flow, particularly
during summer, may increase salinity, which was overall higher at this reef site. However, the difference to the other
sites was minuscule and unlikely to have affected calcifying (Röthig et al., 2016) and bioeroding biota. The variability
of diurnal pH on the other hand presumably has stronger impacts on the performance of calcifiers and bioeroders.
Previously, pH variability across a reef flat and slope were demonstrated to correlate with net accretion dynamics by
showing higher net accretion prevailing in sites of less variable pH conditions (Price et al., 2012; Silbiger et al., 2014),
which reflects the pattern observe here.
The fluctuation in pH may (in part) represent a biotic feedback signature in reef habitats, which entails changes in sea
water chemistry caused by dominant biotic processes, i.e., calcification, carbonate dissolution, and
respiration/photosynthesis (Bates et al., 2010; Silverman et al., 2007a; Zundelevich et al., 2007). Commonly, such pH
fluctuations are influenced by changes in carbonate system variables, e.g. DIC and TA (Shaw et al., 2012; Silbiger et
al., 2014), which can modify the antagonistic processes of calcification and bioerosion/dissolution (e.g., Andersson,
2015; Langdon et al., 2000; Tribollet et al., 2009). In particular, in our nearshore study site, where benthic macro
community abundance was low, biological activity in the sandy bottom (e.g., permeable carbonate sands) might be a
crucial factor contributing to the biotic feedback (Andersson, 2015; Cyronak et al., 2013; Eyre et al., 2018).

*Total alkalinity and nutrients*
The increase in TA is often associated with increased carbonate ion concentration and $\Omega_a$, which facilitate the
precipitation of carbonates supporting the performance of reef-builders (Albright et al., 2016, 2018; Langdon et al.,
2000; Schneider and Erez, 2006; Silbiger et al., 2014). We identified a positive correlation of TA with reef growth in
our dataset. The difference in TA across our study sites was small, but in the range of natural cross-shelf differences
reported from other reefs (e.g. reefs in Bermuda, 20 - 40 $\mu mol\ kg^{-1}$, Bates et al., 2010), and as high as 50 $\mu mol\ kg^{-1}$,
the TA enrichment that enhanced community net calcification in a reef-enclosed lagoon (Albright et al., 2016). On
the other hand, high calcification rates can deplete TA, whereas dissolution of carbonates can enrich TA measurably,
specifically in (semi) enclosed systems (Bates et al., 2010), which we do not observe along the cross shelf gradient. It
remains to be further investigated how TA dynamics across the shelf relate to reef growth processes.
Although increased nutrients are commonly linked to reef degradation initiated through phase shifts, increased
bioerosion rates, and/or the decline of calcifiers (Fabricius, 2011; Grand and Fabricius, 2010; Holmes, 2000), our
dataset suggests that a highly oligotrophic system such as the central Red Sea reefs may benefit from slight increases
of certain nutrient species. Specifically, natural minor increases of N and P might have a positive effect on ecosystem
productivity and functioning including carbonate budgets. A moderate natural source of nutrients, e.g., from sea bird
populations, can indeed have a positive effect on ecosystem functioning, in contrast to anthropogenic run-off (Graham
et al., 2018). Interestingly, our study also identified $PO_4^{3-}$ concentration as an abiotic correlate of reef growth. In the
Red Sea, high N:P ratios indicate that P is a limiting micronutrient, e.g. for phytoplankton (Fahmy, 2003). $PO_4^{3-}$ is not
only essential for pelagic primary producers, but also for reef calcifiers and their photosymbionts, such as the stony
corals and their micro-algal Symbiodiniaceae endosymbionts (Ferrier-Pagès et al., 2016; LaJeunesse et al., 2018).
Experimental studies have demonstrated that $PO_4^{3-}$ provision can maintain the coral-algae symbiosis in reef-building
corals under heat stress (Ezzat et al., 2016). Conversely, P limitation can increase the stress susceptibility of this
symbiosis (Pogoreutz et al., 2017; Rädecker et al., 2015; Wiedenmann et al., 2013). In light of our results, it will be
of interest to link spatio-temporal variation of inorganic nutrient ratios with patterns of reef resilience in the central
Red Sea to understand their effects on long-term trends of reef growth.

**4.4 Reef growth trajectories in the Red Sea**
Carbonate budgets provide an insight into ecosystem functioning and can be used as a powerful tool to track reef
trajectories through time. This includes the exploration of past and current reef trends, which may be critical for
prediction of future reef development (Januchowski-Hartley et al., 2017). Indeed, the absence of comparative baseline
data limits a historical perspective on the central Red Sea $G_{budget}$ presented here. Previously reported Red Sea data
include pelagic and reefal carbonate accretion rates from 1998, estimated using basin-scale historical measurements
of TA (Steiner et al., 2014). Another dataset employed the census-based budget approach for a highly seasonal high-
latitude fringing reef in the GoA from 1994 - 1996 (Dullo et al., 1996), which is methodologically similar to the
*ReefBudget* approach. Both reef growth estimates provide similar rates: The TA-based reef accretion estimate from
1998 was 0.9 kg $CaCO_3\,m^{-2}\,y^{-1}$ and the GoA fringing reef budget from 1994 - 1996 ranged between 0.7 and 0.9 kg
$CaCO_3\,m^{-2}\,y^{-1}$. Additionally, the gross calcification rate of the offshore benthic communities ($G_{benthos}$) compares well
with the maxima measured in the GoA reefs in 1994 (i.e., 2.7 kg $CaCO_3\,m^{-2}\,y^{-1}$) (Heiss, 1995). The $G_{budgets}$ assessed
in the present study are in accordance with these data, indicating stable reef growth rates in the Red Sea basin in the
recent 20 years, despite the ongoing warming trend and observed impairment in coral calcification in a coral species
(Cantin et al., 2010; Raitsos et al., 2011). Yet, data are limited and comparisons between the central Red Sea and the
GoA should be interpreted with great caution. Due to the strong latitudinal gradient of temperature and salinity along
with differences in seasonality between the central Red Sea and the GoA, reef growth dynamics from the two regions
may fundamentally differ. Hence, far larger (and ideally cross-latitude) datasets will be needed to determine more
accurately whether a declining calcification capacity of Red Sea corals has already become a basin-scale phenomenon
and whether there are species-specific differences. In this study we have demonstrated that offshore reefs in the central
Red Sea still maintain a positive carbonate budget, yet can be considered 'underperforming' below "optimal reefal
production" (Vecsei, 2004). In the context of reef growth trajectories, the data presented in this study should serve as
a valuable contemporary baseline for comparative future studies in the central Red Sea. Importantly, these data were
collected before the Third Global Bleaching Event, which impacted the region during summers 2015 and 2016
(Monroe et al., 2018). The present effort therefore will be of great value when assessing potential (long-term) changes
of Red Sea $G_{budgets}$ following this substantial disturbance.

## 5 Conclusions

The Red Sea is a geographic region where coral reefs exist in a naturally high temperature and high salinity environment. Baseline data for reef growth from this region are scarce and particularly valuable as they provide insight into reef functioning under environmental conditions that deviate from the global average for coral reefs. As such, they can provide a potential outlook to future ocean scenarios. Overall, we found net erosion in a near shore reef site, about neutral growth in a midshore reef site, and net accretion in an offshore reef site. A comparison of central Red Sea reef growth dynamics to other major reef systems revealed important differences and argue for *in situ* studies in underexplored major reef regions. For instance, our study highlights the importance of coralline algae as a reef-building agent and shows that the erosive forces in the Red Sea are not as pronounced (yet) as observed elsewhere. Reef growth on Red Sea offshore reefs is comparable to the majority of reef growth estimates from other geographic regions, which today perform well below what has been considered a 'healthy reef' carbonate budget. A first comparison with data from recent years suggests that reef growth rates in the central Red Sea have not decreased substantially over the last two decades, despite potential negative effects of the ongoing warming trend. The absence of comparative long-term data from the region hampers long-term predictions. We therefore advocate additional research to better inform past and future trajectories of reef growth dynamics under consideration of the challenging and unique environmental settings of the Red Sea.

**Acknowledgements**

We thank the Coastal and Marine Resources Lab (CMOR) at King Abdullah University of Science and Technology (KAUST) for logistics and operations at sea (E. Al-Jahdali, A. Al-Jahdali, G. Al-Jahdali, R. Al-Jahdali, H. Al-Jahdali, F. Mallon, P. Müller, and D. Pallett), as well as for the assistance with the deployment of oceanographic instruments (L. Smith, M.D. Pantalita, and S. Mahmoud). We would like to acknowledge field assistance by C. Roder and C. Walcher in setting up the monitoring sites. We thank M. Khalil for providing a map of the study sites. We thank the comments and suggestion from two anonymous reviewers and S. Comeau (Laboratoire d'Océanographie de Villefranche) for his critical comments and helpful suggestions to improve the manuscript. Research reported in this publication was supported by funding to CRV from KAUST.

**Data availability.**

All data are provided in the manuscript and supplement. In addition, physicochemical datasets, reef census, and limestone block assay raw data are available from the Dryad Digital Repository doi:10.5061/dryad.19kd421.

The Supplementary material related to this article is available online *(will be included by Copernicus)*

**Author contribution**

Resources: CRV

Project administration: CRV

Conceptualization: AR

Investigation: AR TR CP

Methodology: AR TR CP
Formal analysis: AR CP VS
Validation: AR CP TR VS CRV
Visualization: AR
Funding acquisition: CRV
Writing - original draft: AR
Writing – review & editing: CRV TR CP VS AR
Data curation: AR
**Competing interests**
The authors declare that they have no conflict of interest.

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

 **Figures**

**Figure 1. Design of studies and reef sites in the central Red Sea.** Maps (a) and (b) indicate geographic location and
the study sites along a cross-shelf gradient. Schemes in (c) – (e) summarize the study designs for the assessment of
the two reef growth metrics, $G_{net}$ and $G_{budget}$, and the characterization of the abiotic environments in the central Red
Sea. Maps have been adapted from Roik et al. (2015).

**Figure 2. Net-accretion/-erosion rates ($G_{net}$) in the central Red Sea.** $G_{net}$ were measured *in situ* using limestone
blocks (100 x 100 mm) that were deployed along the cross-shelf gradient, three sets of blocks were deployed for 6,
12, or 30 months, respectively. Photos (a), (b), and (c), show freshly collected limestone blocks that were recovered
after 30 months deployment. The photos (d), (e), and (f) show the same blocks after bleaching and drying. Boring
holes of endolithic sponges are clearly visible in blocks from the nearshore and midshore reef sites. Blocks from the
midshore and offshore reefs are covered with crusts of biogenic carbonate mostly accreted by coralline algae
assemblages (scales in the photos show cm). $G_{net}$ data obtained from the limestone block assay are plotted in (g). All
data are presented as mean ± standard deviation.

**Figure 3. Census-based carbonate budgets in the central Red Sea.** A schematic overview of the census-based
carbonate budget approach that was adapted from the *ReefBudget* methodology byPerry et al., (2012) is displayed in
(a). Details on input data and equations, employed in the calculations, are available as Supplementary Materials (Text
S1 and respective Supplemental Tables). In (b) reef carbonate budgets are plotted in dark grey ($G_{budget}$) and related
biotic variables in white. The biotic variables, i.e., site-specific calcification rates of benthic communities ($G_{benthos}$),
net-accretion/-erosion rates of reef "rock" surface area ($G_{netbenthos}$), and the epilithic erosion rates of echinoids and
parrotfishes ($E_{echino}$, $E_{parrot}$) contribute to the total reef carbonate budget ($G_{budget}$) at each reef site. All data are presented
as mean ± standard deviation. Images from www.ian.umces.edu*;* photos by A.Roik.

**Figure 4. Abiotic conditions in the reef sites.** Temperature, salinity, and diurnal $pH_{NBS}$ variation (= diurnal standard
deviations) were measured continuously over the respective seasons by CTDs (conductivity-temperature-depth
loggers including an auxiliary pH probe). Furthermore, inorganic nutrients and total alkalinity (TA) were measured in
discrete samples across reef sites and seasons. Boxplots illustrate the differences of seawater parameters between the
reefs within each season (box: 1st and 3rd quartiles, whiskers: 1.5-fold inter-quartile range, points: raw data scatter).

**Tables**
**Table 1. Glossary of reef growth metrics.**

| Metric | Description | Input data for calculation of the metric |
|---|---|---|
| $G_{net}$ | Site-specific net-accretion/-erosion rates (internal and epilithic) measured *in situ* using limestone blocks | - |
| *$G_{budget}$ | Ecosystem-scale census-based carbonate budget of a reef site | $G_{benthos}$, $G_{netbenthos}$, $G_{netbenthos}$, $E_{echino}$, $E_{parrot}$ |
| $G_{benthos}$ | Census-based calcification rate of benthic calcifier community (corals and coralline algae) per reef site | Site-specific benthic calcification rates (collated from this study and from Roik et al. 2015) |
| $G_{netbenthos}$ | Census-based net-accretion/-erosion rates of reef "rock" surface area per reef site | Site-specific net-accretion/-erosion rates measured in this study using limestone blocks ($G_{net}$) |
| $E_{echino}$ | Census-based echinoid (sea urchin) erosion rates per reef site | Genus and size specific erosion rates for sea urchins from literature |
| $E_{parrot}$ | Census-based parrotfish erosion rate per reef site | Genus and size specific erosion rates for parrotfishes from literature |

*The method of $G_{budget}$ calculation is described in the supplements (please refer to Text S1).

**Table 2. Net-accretion/-erosion rates $G_{net}$ [kg CaCO$_3$ m$^{-2}$ y$^{-1}$] in coral reefs along a cross-shelf gradient in the central Red**
**Sea.** $G_{net}$ was calculated using weight gain/loss of limestone blocks that were deployed in the reefs. For each deployment duration,
6, 12, and 30 months, a set of 4 replicate blocks was used. Each block was measured once. Provided are means per reef site and
standard deviations (in brackets).

| $G_{net}$ | Deployment time [months] | | |
|---|---|---|---|
| Reef site | 6 | 12 | 30 |
| Offshore | 0.14(0.11) | 0.08(0.09) | 0.37(0.08) |
| Midshore | 0.11(0.16) | 0.01(0.07) | 0.06(0.12) |
| Nearshore | 0.11(0.07) | -0.61(0.49) | -0.96(0.75) |



**Table 3. Reef carbonate budgets and contributing biotic variables [kg CaCO$_3$ m$^{-2}$ y$^{-1}$] along a cross-shelf gradient in the**
**central Red Sea.** Calcification rates of benthic calcifiers (G$_{benthos}$), net-accretion/-erosion rates of the reef "rock" surface area
(G$_{netbenthos}$), and the erosion rates of echinoids and parrotfishes (E$_{echino}$, E$_{parrot}$) contribute to the total carbonate budget (G$_{budget}$) at a
reef site. Shown are means per site are shown and standard deviations (in brackets).

| Reef | G$_{budget}$ | G$_{benthos}$ | G$_{netbenthos}$ | E$_{echino}$ | E$_{parrot}$ |
|---|---|---|---|---|---|
| **Offshore** | 2.44(1.03) | 2.81(0.65) | 0.09(0.02) | -0.02(0) | -0.44(0.7) |
| **Midshore** | 1.02(0.35) | 1.76(0.24) | 0.01(0) | -0.02(0.04) | -0.73(0.31) |
| **Nearshore** | -1.48(1.75) | 0.43(0.15) | -0.31(0.13) | -0.23(0.19) | -1.36(1.89) |


**Table 4. Abiotic parameters relevant for reef growth at the study sites along a cross-shelf gradient in the central Red Sea.**
Temperature (Temp), salinity (Sal), and diurnal pH variation (diurnal SDs of $pH_{NBS}$ measurements) were continuously measured
using *in situ* probes (CTDs). Weekly collected seawater samples were used for the determination of inorganic nutrient
concentrations, i.e. nitrate and nitrite ($NO_3^-$ & $NO_2^-$), ammonia ($NH_4^+$), phosphate ($PO_4^{3-}$), and total alkalinity (TA). Provided are
means and standard deviations (in brackets).

| Site / Season | Temp | Sal | Diurnal pH variation | $NO_3^-$ & $NO_2^-$ | $NH_4^+$ | $PO_4^{3-}$ | TA |
|---|---|---|---|---|---|---|---|
| | [°C] | | | [µmol kg⁻¹] | [µmol kg⁻¹] | [µmol kg⁻¹] | [µmol kg⁻¹] |
| Avg. winter | 26.07(0.54) | 39.18(0.18) | 0.11(0.12) | 0.32(0.19) | 0.38(0.29) | 0.08(0.02) | 2487(20) |
| Avg. summer | 30.85(0.69) | 39.44(0.18) | 0.05(0.05) | 0.61(0.25) | 0.54(0.34) | 0.04(0.05) | 2417(27) |
| Offshore / winter | 25.97(0.36) | 39.18(0.16) | 0.04(0.02) | 0.4(0.23) | 0.38(0.41) | 0.09(0.02) | 2492(21) |
| Offshore / summer | 30.68(0.63) | 39.38(0.17) | 0.04(0.04) | 0.59(0.24) | 0.51(0.17) | 0.04(0.03) | 2439(15) |
| Midshore / winter | 26.1(0.49) | 39.17(0.2) | 0.07(0.04) | 0.28(0.22) | 0.35(0.19) | 0.07(0.02) | 2494(16) |
| Midshore / summer | 30.56(0.61) | 39.39(0.14) | 0.05(0.05) | 0.63(0.26) | 0.7(0.53) | 0.06(0.08) | 2422(26) |
| Nearshore / winter | 26.13(0.69) | 39.2(0.17) | 0.23(0.14) | 0.29(0.12) | 0.4(0.29) | 0.07(0.01) | 2476(19) |
| Nearshore / summer | 31.32(0.59) | 39.56(0.15) | 0.09(0.06) | 0.6(0.28) | 0.42(0.16) | 0.02(0.01) | 2391(15) |



**Table 5. Coefficients from Spearman rank order correlations for abiotic and biotic predictor variables vs. $G_{net}$ and $G_{budget}$.**
The means of abiotic and biotic variables per reef site were correlated with $G_{net}$ (= net-accretion/-erosion rates of limestone blocks)
and $G_{budget}$ (= census-based carbonate budgets). Strong and significant correlations ($\rho$ values > |0.75|) are marked in **bold**. *P*-values
were adjusted by the Benjamini-Hochberg method. CCA = crustose coralline algae; CC = calcifying crusts

| | $G_{net}$ | | $G_{budget}$ | |
|---|---|---|---|---|
| **Abiotic variables** | $\rho$ | *p*(adj.) | $\rho$ | *p*(adj.) |
| Temperature | -0.47 | *n.s.* | -0.52 | *n.s* |
| Salinity | **-0.82** | *< 0.01* | **-0.82** | *0.001* |
| Diurnal pH variation | **-0.95** | *< 0.001* | **-0.89** | *< 0.001* |
| $NO_3^-$&$NO_2^-$ | **0.95** | *< 0.001* | **0.89** | *< 0.001* |
| $NH_4^+$ | 0.47 | *n.s.* | 0.52 | *n.s.* |
| $PO_4^{3-}$ | **0.82** | *< 0.01* | **0.82** | *0.001* |
| TA | **0.95** | *< 0.001* | **0.89** | *< 0.001* |
| **Biotic variables** | $\rho$ | *p*(adj.) | $\rho$ | *p*(adj.) |
| % cover CCA/CC | **0.95** | *< 0.001* | **0.78** | *< 0.01* |
| % cover Algae/Soft coral/Sponge | 0.47 | *n.s.* | 0.26 | *n.s.* |
| Parrot fish abundance | **-0.95** | *< 0.001* | -0.49 | *n.s.* |
| Echinoid abundance | 0.47 | n.s. | -0.54 | *n.s.* |
| % cover branching hard corals | | | -0.25 | *n.s.* |
| % cover encrusting hard corals | | | 0.26 | *n.s.* |
| % cover massive hard corals | | | 0.34 | *n.s.* |
| % cover foliose hard corals | | | 0.50 | *n.s.* |
| % cover Acroporidae | | | 0.27 | *n.s.* |
| % cover Pocilloporidae | | | 0.51 | *n.s.* |
| % cover Poritidae | | | 0.45 | *n.s.* |
| % cover hard coral | | | 0.63 | *n.s.* |
| Rugosity | | | **0.75** | *< 0.01* |
