# Peer review of "Coral reef carbonate budgets and ecological drivers in the central Red Sea - a naturally high temperature and high total alkalinity environment"

_Biogeosciences, 2018_

## Referee Comment (RC1) · Anonymous Referee #1 · 13 Apr 2018

This paper presents an interesting study that aimed to characterize the effects of biotic and abiotic factors on the carbonate budget of different sites of a reef in the Red Sea. This study is based on a pretty extensive dataset that combines benthic community monitoring, determinations of organisms' calcification, monitoring of chemical and physical parameters, etc. The paper could be pleasant to read but the discussion drags into discussing obvious facts and do not really discussed the results and the significance of this study. Aside of the specific comments listed below I have 2 major comments:

- First, the determination of pH and then the carbonate chemistry was incorrect. pH

[Figure]

on the NBS scale is not the appropriate scale to determine seawater pH. Furthermore, the authors used the R package seacarb and the pH free-scale (while measurements were done on the NBS scale) for all the calculations of the carbonate chemistry, which is incorrect. A quick test on CO2sys shows that for a pH of 8.1 (NBS scale), a TA of 2300 and a temperature of 28, the corresponding pCO2 is 507 and the saturation state 3.19. Using the same pH value on the free –scale yields to a pCO2 of 472 and a saturation state of 3.34.

- Second, one of the main conclusion of the authors is that TA is the main driver of reef calcification. While I do not question the mathematical approach of the authors, I do not believe that the relative minor differences in TA between sites can explain the large differences in Gnet and Gbudget between sites. From a biological and biogeochemical point of view, TA differences of $\sim$ 15-50 umol are not highly significant and are not sufficient to explain the large differences in precipitation and erosion between sites. A positive effect of TA on coral calcification has indeed been shown in previous studies but the TA enhancement was of several hundreds of $\mu$mol.

Specific comments

L-25-26: "Beneficial and detrimental factors" is not clear, what does that refer to? L-49-50: Sediment export is also very important in a reef budget. L-52-53: What about flow and light? Those two parameters are critical but not discussed at all in the manuscript. L-57: It would be good for the reader to start by clearly defining what Gbudget and Gnet are referring to in this manuscript. L-59: Not clear, calcification rate of the reef or corals? L61-63: This is not entirely true. Calcification rates of some organisms are actually enhanced by warming. Also, the temperature threshold between decrease in calcification and death is very thin for corals. L-64-65: This is still an hypothesis that is highly debated. L-67-69: What about a crown-of –thorn starfish outbreaks? L-86-87: Tambutté et al. 2011 is not the best reference here.

L120-121: The pH probes from CTD are unreliable and not recommended to characterize pH in seawater. L127-144: See my major comment#1 L-146-156: It is not clear how many blocks were deployed and retrieved at each sites/time points. L-182-188: I would recommend adding the supplementary material text here.

Results: It is a bit strange to separate the pH from the other carbonate chemistry parameters.

L270: Aside of the problem with the scales, there are some discrepancy between pH continuous and pH discrete. As a result, if the parameter of the carbo chem were calculated from the pH continuous the results for the saturation state etc would be completely different (especially for the inshore sites where continuous pH was much higher than the discrete one). Were any cross-calibrations made between the pHcontinuous and the pH discrete?

L-304: n= ?

L323: Why is pH continuous used in the model while the carbonate chemistry parameters are calculated from pHdiscrete?

L-326-327: As explained before it is hard to explain biologically how such small changes in At can explain 65% of Gnet.

L-347-348: O2 could indeed be an important driver, why was O2 not determined?

L-351-354: . . .and the wrong scale was used.

L-358-359: The high temperature of the Red Sea also explains the high omega.

L-365-366: This is already well known. . .

L-376: pH goes down to $\sim$ 7.3 in Camp et al. 2017 (Sci Rep)

L-381: pH variations are not an indication of fluctuation in AT.

L-405: PO43- is not "a source of energy".

L-446-452: Isn't that the case in a lot of reefs?

L489-490: Was this decrease in calcification linked to bleaching events, etc?

---

## Referee Comment (RC2) · Anonymous Referee #2 · 14 Apr 2018

Review for Roik et al. "Coral reef carbonate budgets. . ."

OVERALL COMMENTS:

This is a mostly well-written MS, and the language and style is above average, the figures and tables are of high quality. The study concerns a highly relevant topic from an interesting marine area that may be less well studied in this context than other coral reefs (Berumen et al. 2013; Schonberg et al. 2017). The MS is based on a large work effort and produced a large amount of valuable data. I compliment the authors for tackling such a timely and complex task.

However, I think the study has a number of shortcomings that need to be addressed

before the MS can be published. In my opinion the data anayses were not correctly performed and will need to be redone. This will likely lead to the need to re-write a few parts of the MS. The Methods section may require some more detail for clarity, and maybe the terminology could be simplified or streamlined to make it easier for the reader. In particular the assessmet of the bioerosion needs to be clearer. I have further listed references that may be useful in the context. These cannot all be included and are subject to the choice of the authors. The title seems to be a bit misleading.

I recommend publication after re-analysing the data and MS revision. It may thus be necessary to re-review the MS.

DETAILED COMMENTS:

DATA ANALYSIS: In my opinion the data evaluation is faulty, and the statistical models and means were not built respecting the existing data hierarchy. This matters even more as the sample size of 4 replicate blocks and 6 transects per subsite is very small for data that can be expected to be highly patchy and variable. To my understanding, there are two independent "between" factors, "distance from shore" (3 levels) and "time of exposure" (3 levels). Then there is the within factor "reef area" or "hydrodynamic exposure level" (backreef, forereef, lagoon or exposed, sheltered). The latter "within" levels are not independent within a given reef and would need to be nested in "distance from shore". Ignoring this data hierarchy resulted in pseudoreplication and a higher test power than actually justified (S12). This is the case for the 6 and 12 mo analyses, and the 30 mo analysis seems to include only 4 subsites, all at the same data level (near-fore, mid-fore, mid-lagoon and off-fore). I assume that the means are therefore also not correctly calculated (not stepwise). Sadly, this does not only apply to the block data. All figures and tables will need to be restructured accordingly, means will have to be calculated stepwise, following the same data hierarchy as the statistical models. It would be good to have some sort of schematic figure that visualises the data design and hierarchy. Maybe "lagoon" could be left out of the 30 mo analysis to make things clearer and possibly more powerful to pick up effects/trends? I think it

would be acceptable to define "lagoon" and "backreef" as "sheltered" and give them the same data status, but it is not acceptable to include near-fore, mid-fore and off-fore at the same level as mid-lagoon. Unfortunately the situation will cause significantly more work effort and potentially the need for rewriting parts of the MS.

THE CHOICE OF THE BIOERODER TAXA AND THE ASSESSMENT IS NOT EN-TIRELY CLEAR. The bioeroders were assessed in two groups – dominant epilithic bioeroders via biomass estimates, and endoliths in a block assay. It is not quite clear whether the counted scarids only represented "excavators", i.e. fishes that bite and break calcium carbonate or whether the "others" contained fishes that mainly eat fleshy algae and thus dilute the overall value? Please clarify in the Methods. The blocks were weighed but not otherwise assessed? 30 months exposure to settlement is not that long, and unless you consistently found larger borers in the blocks you probably have to assume that you are still capturing earlier successional communities. Can provide more details in the results what borers were present in the 6, 12 and 30 mo blocks?

THE BIOERODER DATA MAY PERHAPS NOT BE REPRESENTATIVE, AND THERE MAY BE A RISK OF COMPARING APPLES WITH PEARS. E.g. William Kiene has assessed temporal successions of coral reef biota in a comparatively pristine environment settling onto experimental blocks. According to his data, the present 30 mo blocks may still be reflecting a developing phase of that settlement. The block assay was not controlled by implementing non-erodable blocks so that the net value could be corrected with data that matched the experimental situation. Calcification and epilithic bioerosion was assessed in situ and thus in all likelihood concerned a mature community, the blocks assessed borers and may not be as representative. In any case, it would be good if the authors could provide more detail on how the blocks were deployed (fixed to the reef and allowing lateral invasion and grazer access, or on a rack only allowing larval settlement and probably excluding urchins) and what were their observations after retrieval. This would enable a better understanding what the data represent. The comparison of the present data with past data from the Gulf of Aqaba

may also be a case of apples and pears (see below). Overall the interpretation needs to be cautious. It is a pity that dominant borer distributions were not assessed in situ, which is a significant data lack for the Red Sea and could have been done while assessing the benthos. The epilithic bioeroders have always received more interest. In any case, this needs to be reflected in the wording, you did not assess bioeroders on the reef, but only the dominant epilithic bioeroder-grazers (fish, urchins).

THE DATA NEED TO BE USED AND INTERPRETED WITH CAUTION. The data are still valuable and good to have. However, in view of some of the above comments, and the extremely large error values, the interpretation cannot be generalised too much and needs to make allowances – in a larger dataset with smaller error the data may have shown different patterns. The temporal analysis across decades appears risky in that the present study was conducted in the central Red Sea, with a different protocol and focus, the earlier studies in the Gulf of Aqaba. I think it would be better to resist and not try to construct a long-term trend, but rather point out the lack of comparable data? Also, when data are plotted respecting the data hierarchy I can only see significant crosshelf trends for the parrots.

SOME PARTS WERE CONFUSING. I found it at times difficult to keep track of the data and terms. Is it correct that the bioerosion (net) value was called Gnet, net accretion/erosion values and Gnetbenthos in different parts of the MS? Gbenthos and Gnetbenthos are too similar, wouldn't it be better to clearly separate those terms? What does G stand for? Growth? Why then is E marked separately, it is negative growth? Wouldn't it be easier for the reader to skip the Gs and Es and use more intuitive words? To abbreviate in situ accretion, in situ bioeroder biomass = approximation of bioerosion, assay net accretion/bioerosion? I am not sure what you mean with "cumulative" data in the block assay over time. You did not repeatedly measure the same blocks, right? And if you added data from the second and third block set to the first that would be inappropriate, as the successional stages change over time. You need to evaluate the block sets separately and can only display a 30 mo net accretion/erosion situation from

blocks that were exactly that long in the water. Please clarify what you did. For urchins and parrots all 6 subsites were assessed for biomass and abundances – why are only 4 subsites presented for the respective bioerosion data?

THE TITLE COULD BE TWEAKED TO BETTER REPRESENT THE CONTENTS OF THE PAPER. I feel that putting the budget first is misleading, because these data are only a small part of the paper, and the huge error value makes the overall budget estimate a highly unreliable value. Would it be OK to change the title to something more along the line of: Ecological drivers of coral reef carbonate cycling in the central Red Sea – a high temperature, high total alkalinity environment

Berumen ML, Hoey AS, Bass WH, Bouwmeester J, Catania D, Cochran JE, Khalil MT, Miyake S, Mughal MR, Spät JL, Saenz-Agudelo P. The status of coral reef ecology research in the Red Sea. Coral Reefs. 2013 Sep 1;32(3):737-48. Schönberg CH, Fang JK, Carballo JL. Bioeroding sponges and the future of coral reefs. InClimate Change, Ocean Acidification and Sponges 2017 (pp. 179-372). Springer, Cham.

TECHNICAL COMMENTS: Are included as attached.

Please also note the supplement to this comment:
https://www.biogeosciences-discuss.net/bg-2018-57/bg-2018-57-RC2-supplement.pdf
* * *
[Figure]

**Fig. 1.** Demonstration as figures

**Supplement:**

Technical comments:

l. 25: Complicated sentence. Can you simplyfy by dropping sub-clauses?

l. 36: … AT correlated well and positiveLY with reef growth …

l. 46: Incorrect statements. There are highly functional and very important cold-water reefs. Even warm water coral reefs can exist and thrive under oligotrophic conditions. And to some degree reefs can occur in marginal or even hostile environments. Suggestion for rewording: Positively accreting warm-water coral reefs usually occur in aragonite-saturated and oligotrophic  oceans, where pivotal ecosystem functions can be best maintained…

l. 54: Too simple a view? OK, maybe you have to keep it short… Still, maybe have a look at

Perry CT, Harborne AR. Bioerosion on modern reefs: impacts and responses under changing ecological and environmental conditions. InCoral Reefs at the Crossroads 2016 (pp. 69-101). Springer, Dordrecht.

Schönberg CH, Fang JK, Carreiro-Silva M, Tribollet A, Wisshak M. Bioerosion: the other ocean acidification problem. ICES Journal of Marine Science. 2017 May 1;74(4):895-925.

Glynn 1997 and Glynn and Manzello 2015 are the same thing. Please choose one.

l. 63: Should "slowing down" not be replaced with something like "interrupting"? Corals often die, after all.

l. 64: Convoluted sentence. How about: As ocean acidification decreases the ocean's pH and $\Omega$a at the same time, calcification becomes energetically more costly (…).

l. 67: There are so many examples for OA-enhanced bioerosion by now that you need to use "e.g." or cite an overview. Please also add: "frequently" a hallmark.

l. 82: Replace "is" widely attributed with "was" to refer to earlier results, not a fact.

l. 83-84: Again, use the past tense to indicate that you refer to oether people's results.

l. 89: Replace "Aside" with "Apart"

l. 90: What about e.g.

Lazar B, Loya Y. Bioerosion of coral reefs-A chemical approach. Limnology and Oceanography. 1991 Mar 1;36(2):377-83.

Mokady O, Lazar B, Loya Y. Echinoid bioerosion as a major structuring force of Red Sea coral reefs. The Biological Bulletin. 1996 Jun 1;190(3):367-72.

Alwany MA, Thaler E, Stachowitsch M. Parrotfish bioerosion on Egyptian red sea reefs. Journal of experimental marine biology and ecology. 2009 Apr 15;371(2):170-6.

Bertram GC. 60. Some Aspects of the Breakdown of Coral at Ghardaqa, Red Sea. Journal of Zoology. 1936 Dec 1;106(4):1011-26.

Erez J, Reynaud S, Silverman J, Schneider K, Allemand D. Coral calcification under ocean acidification and global change. InCoral reefs: an ecosystem in transition 2011 (pp. 151-176). Springer, Dordrecht.

Hassan M. Modification of carbonate substrata by bioerosion and bioaccretion on coral reefs of the Red Sea. Shaker Verlag; 1998.

Zundelevich A, Lazar B, Ilan M. Chemical versus mechanical bioerosion of coral reefs by boring sponges-lessons from Pione cf. vastifica. Journal of experimental biology. 2007 Jan 1;210(1):91-6.

Mokady O, Graur SR. Coral-host specificity of Red Sea Lithophaga bivalves: interspecific and intraspecific variation in 12S mitochondrial. Molecular Marine Biology and Biotechnology. 1994;3(3):158-64.

Cornelia Maier 1997. Distribution and abundance of internal bioeroders in coral reefs. A field survey in the northern Red Sea. MSc (diploma) thesis, ZMT & Bremen University, Germany, 94 pp.

And there is more on calcification as well:

Braithwaite CJ. Patterns of accretion of reefs in the Sudanese Red Sea. Marine Geology. 1982 Feb 1;45(3-4):297-325.

Etc…

There was a bit of an overview re bioerosion and lack of data from the Red Sea in

Schönberg CH, Fang JK, Carballo JL. Bioeroding sponges and the future of coral reefs. InClimate Change, Ocean Acidification and Sponges 2017 (pp. 179-372). Springer, Cham.

l. 91: OR bioerosion.

l. 100: Settlement blocks capture ambient endolithic bioerosion rates only after several years, see research by e.g. William Kiene. Blocks can underestimate bioerosion by magnitudes. It is thus good that you had a set of 30 mo ones, which may still be a bit early, but better than in other studies. Consider that you may have captured an early stage endolith community, which would likely be dominated by different organisms than later. Please replace "on" potential drivers with "of".

E.g. Kiene WE. A model of bioerosion on the Great Barrier Reef. InProc 6th int coral Reef Symp 1988 Aug (Vol. 3, pp. 449-454).

Hutchings PA, Kiene WE, Cunningham RB, Donnelly C. Spatial and temporal patterns of non-colonial boring organisms (polychaetes, sipunculans and bivalve molluscs) in Porites at Lizard Island, Great Barrier Reef. Coral Reefs. 1992 Apr 1;11(1):23-31.

Kiene WE, Hutchings PA. Bioerosion experiments at Lizard Island, Great Barrier Reef. Coral reefs. 1994 May 1;13(2):91-8.

l. 101: provides "a" broad insight

l. 106: As the other 2 papers are in a slightly different context it would be good to provide the basic data on the cross shelf differences?

l. 111: Nearshore-fore, midshore-fore, midshore lagoon and offshore-fore?  Why is the lagoon in there? The midshore data are not independent of each other.

l. 114: This sentence can again be made easier to read by dropping commas: At each station additional seawater samples were collected on SCUBA for 5 - 6 consecutive weeks during each of the seasons for the determination of inorganic nutrients and carbonate chemistry: nitrate…

l. 119: Use small characters in the title, also in l. 120 for the loggers

l. 122: Insert "the" in front of "pH probes"

l. 127: "cubitainer" is a brand name. Either use "container" or add a bracket with the producer info.

l. 128:  Replace "over" with "via" or "through" or something

l. 129: Which were the discrete samples, the 4L or the syringe samples?

l. 140: salinity with a small letter

l. 142: use small letter in "free scale" and provide reference for free the free scale being a good equivalent, e.g.

Dickson AG. pH scales and proton-transfer reactions in saline media such as sea water. Geochimica Et Cosmochimica Acta. 1984 Nov 1;48(11):2299-308.

Waters JF, Millero FJ. The free proton concentration scale for seawater pH. Marine Chemistry. 2013 Feb 20;149:8-22.

l. 149: How many blocks were there in total? 64?

l. 151: How were the blocks deployed? It would make a huge difference whether they were fastened directly on the bottom (allowing direct lateral borer invasion and urchin grazing), as opposed to placing them on some sort of rack (where only larval settlement occurs and grazer access would be reduced).

l. 154: Presumably after bleaching the blocks were thoroughly rinsed to remove the bleaching salts? How were the sites distributed over the exposure times? Can you provide a schematic figure? Wouln't it be much easier to understand if you gave the lagoon the same status as the backreefs? Then you would have 3 fore-reef, exposed sites and 3 sheltered sites? It does not make much sense that the 30 mo approach included the lagoon, and you should not include it giving it the same status as the fore-reef sites.  In your case the factor "exposure" would be nested within reef transect, it would be like a subsample per reef.

l. 160: How was this information matched with biota in the blocks? Were any of the species identified, either those seen on the reef or in the blocks? The Red Sea is not well represented in the borer taxonomies, except for the bivalves.

l. 163: WERE assessed

l. 164: What do you mean with non-calcifyers? General reef biota or only bioeroders?

l. 166: Sponges are a major group of bioeroders, yet you grouped them with algae? And didn't you have area that was not covered by living organisms, e.g. patches of sand? It would have been good to have a value for bioerodable calcium carbonate as opposed to area covered by calcifyers or non-bioeroders. Borer abundances strongly vary with substrate type and dead surface areas and have to be normalised to the available substrate in order to make their occurrences comparable between sites (mainly shown for sponges):

Carballo JL, Bautista-Guerrero E, Leyte-Morales GE. Boring sponges and the modeling of coral reefs in the east Pacific Ocean. Marine Ecology Progress Series. 2008 Mar 18;356:113-22.
Schönberg CH. Monitoring bioeroding sponges: using rubble, quadrat, or intercept surveys?. The Biological Bulletin. 2015 Apr;228(2):137-55.

That doesn't apply to your block data (unless you fastened them in areas without abundant dead substrate = lower of larval supply), but will have an effect on your distributions.

l. 170: You did not assess general borer densities, but only the two dominant epilithic bioeroder groups or grazers. Please reword, be more specific (also 2.5). It would also be important to know whether you only included parrots that really have an impact on bioerosion or also others, like scapers and spp. that mainly eat erect sea weeds (see references below). I guess you included all, because you had fairly small size classes in there? That would dilute the data. Pity that you did not look the borers.

Bellwood DR, Choat JH. A functional analysis of grazing in parrotfishes (family Scaridae): the ecological implications. InAlternative life-history styles of fishes 1990 (pp. 189-214). Springer, Dordrecht.
McAfee ST, Morgan SG. Resource use by five sympatric parrotfishes in the San Blas Archipelago, Panama. Marine Biology. 1996 May 1;125(3):427-37.
Lokrantz J, Nyström M, Thyresson M, Johansson C. The non-linear relationship between body size and function in parrotfishes. Coral Reefs. 2008 Dec 1;27(4):967-74.

l. 182: How did you assess accretion and bioerosion? You can provide the details in the supplement, but the main text still needs to be understandable by itself, so why not say that the local accretion/erosion data were assessed by you and published earlier? Gnetbenthos was derived from the blocks? Then you can only rely on 30 mo data for the forereefs (the other periods being even shorter)? And you only have net values? Considering the extreme variation you found this may be a problem.

l. 184: I assume that the "site-specific" net data refer to the blocks? How did you assess calcification by itself? Please specify

l. 202: "showed" = in the past, and delete "a"

l. 212: Gbudget DATA were tested…

I am not quite sure I understand your data design? You had the factors "time" and "reef = distance to shore". You only had very few replicates (blocks) at the lowest level with N=4. This is very small sample size for something as variable as net calcification/bioerodion. Where does your factor "reef area = fore/back/lagoon" go? I think you are quite wrong to use simple ANOVAs, because "reef area" is not independent and should be nested within "distance from shore". So your factors "time" and "distance from shore" are independent between factors and fully crossed for 6 and 12 mo, but "reef area" is a within factor and presently unbalanced. Thus you don't have a 1-factorial simple ANOVA, but a 2-factorial mixed model with the additional factor "reef area" nested within "distance from shore". It would make your life easier if you would call "reef area" "exposure to water movement" and give backreef and lagoon the same status. Having only 4 blocks at the lowest level means you will have to be very careful with your data design, and your present analysis seems misguided to me. I think you should evaluate your data like this:

| DISTANCE FROM SHORE | | | x |
|---|---|---|---|
| NEARSHORE | MIDSHELF | OFFSHORE | TIME OF |
| Exposed: forereef | Exposed: forereef | Exposed: forereef | EXPOSURE: |
| Sheltered: backreef | Sheltered: lagoon | Sheltered: backreef | 6 and 12 mo
30 mo |

Distance from shore
Time of exposure

Time of exposure x distance from shore
Exposure to water movement (nested in distance from shore)

Why don't you evaluate everything in PERMANOVA, which saves the need for transformation and gives you more test power than nonparametrics, as well as allowing testing for factor interaction etc.

l. 221: You have 3 reef sites (near, mid, off) and 2 or 3 reef areas (fore, back, lagoon), which are nested within reef site. You should not use 4 levels in this approach.
l. 225: Why did you exclude urchins for the data analysis of the blocks? Wouldn't urchins have grazed on your blocks? Same for coral cover. Your data are net values, i.e. they include coral settlement. Please increase the range of the tested parameters for the block values. Did you cull out covariables?
l. 240: The difference across the shelf? I assume the water became cooler with distance from shore? This needs to be clear without access to the supplement. The next sentence seems to be wrong? Do you mean the "OFFshore and the midshore"?
l. 248: All 6 sites? Are these hierarchical means, i.e. per reef area and then seasonal means across the distances? Please note that in nested designs the means need to respect the same data hierarchy and need to be calculated stepwise. This part is also not quite clear in the earlier parts of the Methods, please specify and consider for all displayed means. "Nearshore reef" is that back or fore or a mean of back and fore?
l. 253: replace "in" with "at the sites"
l. 254: delet "was"
l. 258: "while the midshelf lagoon", replace "was" with "were". Did you include the backreef sites? Looking at your figures shows me that you didn't, which is a shame. In this case you can explain differences between sheltered vs. exposed only for the midshelf site and should not generalise.
l. 257: please reword to "small at the exposed offshore and midshore siteS"
l. 261: delete "a"
l. 264: replace "contents" with "levels"
l. 268: All this and some of the following is horrible to read with all the parameter abbreviations and long brackets. It is difficult to find the bits of sentences in between. Having all the data in clean overview in a table, is it necessary to clutter the text with so many brackets or can you delete a few of them? The last you can probably do is reducing the number of repetitive units, e.g. "(2422 µmol $A_T$ kg$^{-1}$, 2076 µmol $C_T$ kg$^{-1}$, and 1821 µmol $HCO_3^-$ kg$^{-1}$)" could become "(2422, 2076, and 1821 µmol kg$^{-1}$, respectively)"?
l. 271: the "p" in pCO2 is usually written in italics, which makes it clear that it is a defined unit (correct throughout). Could you also put definitions in the text for the main parameters, not just in the table legend? "Ranged" infers the use of "from... to"
l. 272: "increaseD" (past tense). Replace "of note, ..." with "it should be onted that..."? What do you mean with "propagates to uncertainty"? Do you mean "was in part near the resolution level"?
l. 279: hyphen between ocean and facing
l, 284: Accretion is half of your budget. You need to put some of this in your main text, I think. The results need to be clear in the end, e.g. what they are based on.
l. 286: They are epilithich macrobioeroders. Please make that clear, you did not assess any of the endolithic bioeroders.

l. 288: Urchin and fish means are often in the same range of magnitude as the error values. The large errors make these values highly unreliable.

l. 290: I disagree with the statements of this para. I used the raw data from the supplement to get a better visual impression of your data. I will send an Excel file in attachment to show what I mean. You cannot really say that the urchins were most abundant nearshore and then decreased, because all the error bars overlapped. There is no evidence for a trend. I would strongly recommend to include figures like the ones I tried out, because then you can easily see that the urchins cannot be matched to any particular straightforward pattern, with the possible exception of a much reduced urchin biomass at the offshore site. This would be due to much smaller urchins, because the abundances themselves do not look significatly different.

For the parrotfish, however you will be able to find differences, and this looks quite interesting: Abundances increased across the shelf at the sheltered sites, but decreased at the exposed sites. Bioerosion was strongest inshore exposed sites, but with huge error bars. Overall means: no difference for anything.

So it would be much more interesting to display the effects for both factors cleanly separated, not sure whether it is so interesting to know the range – e.g. especially the bigger parrotfish would have an impact for bioerosion. The overal plots suggest that the parrotfish will have the larger influence (should better be plotted with a secondary axis).

l. 301: Cumulative? Please explain. My understanding was that you had 4 replicates per situation, and no repetitive measurements? You can't add results from different block dets together, you need to respect successional stages of settlement. Replace "in" with "at".

Maybe it would be good to have Figure S1 in the MS. The respective analyses again need to be nested (exposure within distace from shore). Pity that you did not fully replicate all settings. The block data for 6 and 12 mo do not mean much and do not separate out in your supplementary figure. Only after 30 mo the mid- and offshore blocks start to show different patterns. But you can only judge the "exposure to water movement" effect at the midshelf site.

l. 313: Please recalculate respecting the nesting level. It is quite bad that your error is larger than the mean, maybe it would be better to express that same value only for the forereef data? Or separate for exposed and sheltered?

l. 318: What is "net-accretion/erosion of bare substrate Gnetbenthos"? Is that different from Gnet? Maybe 2.5 could be a bit more detailed?

l. 322&323: showED, negatively correlated with what?

l. 329: Can you be more specific about the percentages? If you have 81% of the variation explained, then the difference to 74% would be 7, not 78? Obviously, you haded co-varying factors in your model (e.g. T means and SDs), why didn't you test and cull some of that in PERMANOVA?

l. 334: Replace "so far" with "to date"

l. 337: I would not call this "comprehensive", but "detailed' would be OK.

l. 338: linkED

l. 341: integrateD

l. 353: "weather goal"? replace with "standard", delete "as"?

l. 360: Convoluted sentence, difficult to follow. How about:

To ASSESS  whether Red Sea reefs LIKE OTHER MARINE HABITATS will IN FUTURE reach a critically low $\Omega a$  OR maintain a LONG-TERM calcification-friendly sea water chemistry ,

 HIGH PRECISION experimentS and  high resolution monitoring  are needed.

l. 367: decreaseD, (comma)… increaseD

l. 370: replace "on local" with "at", "WERE similar", add "patterns' after "seasonality"

l. 377: Don't understand sentence: Amongst the range maxima of pH units was ~1.40. Please reword.

l. 378: Replace "in the" with "at the"; siteS, (comma)…

l. 382: What was not considered?

l. 386: delete "s" from "averages"; the wording is not quite clear, better remove "average" and "for the crosshelf gradient"? Use "gradient" instead of "differences" in next sentence.

l. 391: Replace "supplied" with something like "modified", "affected" or "replenished". Please provide a reaference for this statement. DepleteS.

l. 394: Please simplify this fragmented sentence. It should be "correlated WITH"

l. 397: reflecteD (please carefully check the MS and keep the tenses uniform); might read better as: "Nearshore habitats with low reef growth capacity were associated with higher mean temperatures and strong biotic feedbacks that caused comparatively intensive pH fluctuations."

l. 399: Silbiger et al. net accretion responded to pH anomalies: "Previously, SMALL-SCALE pH ANOMALIES HAVE been shown to have a significant impact on LOCAL accretion and erosion dynamics" (probably too small-scale to generalise, see Schönberg CH, Tribollet A, Fang JK, Carreiro-Silva M, Wisshak M. Viewpoints in bioerosion research—are we really disagreeing? A reply to the comment by Silbiger and DeCarlo (2017). ICES Journal of Marine Science. 2017 Oct 3;74(9):2494-500.).

l. 400: Use past tense if referring to your own results, insert "can" in front of "exert" if meant as a more general statement.

l. 405: identifieD, add "as a positive factor" after "concentration" and delete the subclause after the bracket

l. 409: replace "shown" with "demonstrated" (avoid repetition)

l. 410: better to use "coral-algal symbioses" throughout, without "the"?

l. 411: Remove "and", start new sentence with "Conversely"

l. 412: remove "the", make it "in light of"

l. 413: nutrient ratios IN THE CENTRAL RED SEA to understand their effects on LOCAL large-scale and long-term trends of reef growth

l. 418: replace "on" with "at"

l. 420: WAS reported, replace "negative production" with "erosional"

l. 427: showeD. This part is a bit confusing. The methods implied that parrots were counted as epilithic bioeroders? But here it seems the effect of macroalgal cropping was more important? Given the context, this needs to be made clearer here and in the methods: Which groups were included for what purpose of assessment? In the following the two contrasting effects (macroalga cropping and possible bioeroder control vs direct bioerosion) are mixed together. When more parrots correlates to reef growth, I guess you mostly counted the croppers, not the bioeroders? Potentially there is also a lot of among-biota feedback that can be important, but difficult to separate out, see e.g.

Carreiro-Silva M, McClanahan TR. Echinoid bioerosion and herbivory on Kenyan coral reefs: the role of protection from fishing. Journal of Experimental Marine Biology and Ecology. 2001 Jul 30;262(2):133-53.

Brown-Saracino J, Peckol P, Curran HA, Robbart ML. Spatial variation in sea urchins, fish predators, and bioerosion rates on coral reefs of Belize. Coral Reefs. 2007 Mar 1;26(1):71-8.

Mapstone BD, Andrew NL, Chancerelle Y, Salvat B. Mediating effects of sea urchins on interactions among corals, algae and herbivorous fish in the Moorea lagoon, French Polynesia. Marine Ecology Progress Series. 2007 Mar 5;332:143-53.

McClanahan TR. Response of the coral reef benthos and herbivory to fishery closure management and the 1998 ENSO disturbance. Oecologia. 2008 Feb 1;155(1):169-77.

Kennedy EV, Perry CT, Halloran PR, Iglesias-Prieto R, Schönberg CH, Wisshak M, Form AU, Carricart-Ganivet JP, Fine M, Eakin CM, Mumby PJ. Avoiding coral reef functional collapse requires local and global action. Current Biology. 2013 May 20;23(10):912-8.

l. 438: Use "biomass was" (singular)

l. 439: insert "sites" after "offshore"

l. 441: replace "is" with "has been"

l. 443: Bit lost here. You cite evidence for fishing pressure, but the parrots were one of your main structuring force? How then do your sites compare to the less fished sites?

l. 448: This works only when using the 30 mo blocks, right?

l. 450: ...I don't understand this. If your bioerosion data were all assessed as net values, how can you separate out the two processes to give separate values? No, you are still doing this as net values, right? I think you need to reword this. How about: Net reef accretion increased across reef sites from nearshore to offshore, with offshore accretion being twice as strong as inshore erosion. Our Gbudget estimates can thus be interpreted as evidence for the formation of an offshore barrier reef in the central Red Sea

l. 452: showeD; delete "states" and change to singular

l. 453: replace "lowest' with 'most intense"

l. 457: remove "s" from "in part", put bracket at the end of the sentence or at least after "complex"

l. 459: Confused again... You counted the fish, but you didn't count bite marks? If your parrots included a significant amount of herbivores rather than excavators, how would your biomasses allow a good estimate of fish-generated bioerosion? How can you sound so sure?

l. 461: replace "in" with "on"

l. 469: insert "data" after Gbudget, and "with conditions at the majority" This sentence contradicts the last sentence of the paragraph above it. -0.8 to 4.5 is a large range, caused by widely different conditions. How is this "comparable"? I am not sure whether your data are representative enough to be set into a global context. Your overall budget was 0.65 ± 1.73 kg m-2 y-1, i.e. your error was almost 3x as high as your value, so what is the value really? You used net values that were not controlled for calcification, i.e. you could not separate accretion and erosion from your block data, which only had 4 replicates and likely still represented early stages of colonisation, maybe not so typical. And I am not sure how accurately your epilithic biota counts reflect bioerosion. I felt more comfortable with the last sentence of the paragraph before.

I am even less happy with the historical comparison, where people used a different approach than you, which would again have an impact on the outcome, apart from having been conducted at different sites. You cannot say whether there was a change over time, and you lose credibility by doing so, especially by putting enough weight on this to mention it in the abstract. It would be better to delete 4.4 or to use this part to highlight the lack of historical data in a context that can suitably compared. The last part is OK, but I would not call your data as "valuable baseline" with this extreme error. Please re-assess your data, maybe after an improved data assessment the error value will be lower?

l. 479: provideD

l. 510: Please replace "geographic" with "spatial", you cannot generalise enough

to claim you can explain things at geographic level.

l. 510: Delete "not higher than but"? Specify "other regions". Across the shelf? Of the world?

l. 510: I am not sure you captured bioerosion very well. I assume that the blocks did not yet reflect the big borers adequately, and did they allow access to grazer-nioeroders? The small sample size and the missing calcification control also made it difficult to ariive at more reliable values. Given your large error values, bioerosion might have been much higher than your means.

l. 512: I would suggest that you delete any statements re historical trends and rather stress the lack of comparable data, urging for more research.

References:

Please choose whether you want to use any of the extra literature I provided. Can you please use uniform formats in the references and carefully check through the list for correct formats and writing? E.g. some titles use capitals in each word, others not. Latin names are not always in italics, the species name sometimes starts with a capital letter. L. 368: should be New York. The 2 in $CO_2$ should be subscript. Etc... Also in the supplement.

Fig. 2: Daytime is one word.

Fig. 3: What is the difference between measured and estimated? Giving the two midshelf sites the same status as the near- and offshore sites is misleading. Maybe it would be best to use only the 3 sites, or to leave gaps for the other 2 sheltered sites, or to display the lagoon site in a separate panel series.

Fig. 4: Your blocks do not yet look very affected by bioeroders. There is no clear evidence of grazer-bioerosion, yet you gave the parrots a big importance. Did you find any bioeroding molluscs in the blocks (which have an important role in the RS).

Fig. 5: Same problem as for Fig. 3 re site hierarchy. You have two data pairs each reef, sheltered and exposed. These are not independent, which needs to be obvious in the figures as much as it needs to be respected in the analyses. Your in- and offshore error bars are huge. The netbenthos and echino bars need a different scale, this way the data cannot be appreciated. What does the "c" mean over the grey offshore bar?

Table 1:

Please note that your maeans need to be calculated in analogy to your model, and I assume that they are presently incorrect. Within a given season, you need to first calculate means across 4 replicates, then across per reef area (exposed, sheltered – this results in an unbalanced situation, because you only have sheltered values midshelf), then across the reef sites (near, mid, off). Do you mean "calculate" when you write "estimate"?

Tab. 2: This is finally a data display in full analogy. What do you mean with "cumulative"? Please explain in Methods.

Tab. 3: Again problem with non-independence of midshelf sites and large error values in- and offshore, making the means very unreliable.

Tab. 4: I think you need to reduce the number of variable and remove the co-variables.

Tab. 5: You need to make clear that your R2 you mentioned in the text were accumulative. It may be better to use the un-cumulative ones in the text – more intuitive.

Supplement: OK, so this makes it clearer how you assessed epilithic bioerosion. I still think you should mention in the methods that you concentrated on species that are known bioeroders, providing the references. Hoey et al. is printed and a 2016 paper.
Tab S6: Why is fish bioerosion expressed as negative data, but not urchin bioerosion?
Tab S7: Is the first error value 0.6 a printing error or is it really so huge?
Tab S8: That suggests, however, that you included non-eroding parrots after all. Why did you assume these eroded?
Tab S6 and S9: Why do you have 4 data sets here if you have 6 in the tables these are based on? You have the data to display all 6 don't you, why don't you show them? Otherwise you have the same problem re your midshore sites here that need to be addressed.
Fig. S1: Maybe better in the main text?

---

## Referee Comment (RC3) · Anonymous Referee #3 · 19 Apr 2018

This is a comprehensive study that provides valuable baseline data for reef growth in the central Red Sea. The study covers the timely and relevant topics of climate change and OA and assesses drivers of reef growth in a unique area (the Red Sea) that is of growing interest owing to the health of corals in an extreme environment. The study uses an extensive dataset to assess the relationships between reef growth and various abiotic (carbonate chemistry parameters, temp, phosphate) and biotic (bioeroders, coral cover, etc.) drivers. The study shows that reef growth is positively correlated to parameters indicative of carbonate ion availability (e.g., At, omega, carbonate ion concentration) and to phosphate concentrations; while growth is negatively correlated to temperature, $pCO_2$, and pH variability (biotic feedbacks). Reef growth was also

highly correlated to parrotfish abundance. Overall, this is a well-designed, comprehensive study. The manuscript is clear and well-written (though beware of numerous minor grammatical errors). I do have some concerns that should be addressed before publication (below).

Detailed Comments: Starting at Line 69: suggest including 1-2 sentences explaining what a census-based calcification budget approach is and explaining how Perry's Caribbean-based methods were modified for the Red Sea (supplemental).

Line 371: what's the cause of the $PO_4^{3-}$ enrichment in winter?

Line 386: average, not averages

Line 402: "…manipulations on reef communities in situ…" –add in recent manipulative field experiments (e.g., Albright Nature 2016 and 2018).

Suggest a glossary to help readers keep track of various terms (e.g., Gnet, Gbenthos, Gnetbenthos, etc.)

It's somewhat surprising that relatively small changes in carbonate chemistry explain such a large variation in Gnet (even from net accretion to net erosion). According to Table 1, the delta At is < 50 umol in summer, and even less in winter.

Is there a plot of discrete pH versus, or plotted on top of, continuous pH – to see how these two methods compared? CTD pH sensors typically aren't very reliable. Further, two different pH scales are used?

---

## Author Comment (AC1) · 13 May 2018

bg-2018-57

Author response "Coral reef carbonate budgets and ecological drivers in the naturally high temperature and high total alkalinity environment of the Red Sea"

We thank the editor and reviewers for their overall positive evaluation. We appreciate their detailed and constructive feedback and their recognition of the relevance of the presented data. We are happy to consider their recommendations in a revised manuscript.

[Figure]

Overall, the reviewers suggest a similar set of improvements of the current manuscript. We propose to address these as outlined below:

*pH monitoring

-> Reviewers #1 and #3 point out that the approach for pH monitoring does not comply with best practices for carbonate chemistry analysis of seawater, which requires the use of Tris buffers (Total scale measurements). In our study, NBS scale standard buffers were used. In addition, CTD sensors, as employed in our study, are not recommended for precise seawater pH monitoring.

=> We agree with the reviewers that the collected pH data and carbonate chemistry calculations based on these are suboptimal. In a revised version, we suggest to address this by removing these data from the manuscript.

*meta-analysis

-> Our meta-analysis (distance based linear models) shows that total alkalinity is the best explanatory variable of the reef growth parameters (Gnet and Gbudget). Reviewers #1 and #3 have pointed out that the analysis may be correct, but biologically not necessarily relevant, as the changes in total alkalinity were minor and in this order of magnitude are unlikely to result in significant effects on reef calcification processes.

=> We can relate to the reviewers' reasoning. The "precise" results obtained from linear models may not be reasonably justified from a biological point of view. In a revised version, we suggest to address this by removing the results of the distance based linear models. We propose to keep the correlation analysis, as these statistics offer an insightful basic overview on the numerical relationships in our dataset. Overall, we will tone down the conclusions drawn from the results of our meta-analysis.

*sampling design and statistical approach

-> All reviewers have requested a glossary of variables included in the census-based approach. -> Reviewers #1 and #2 ask for a comprehensive scheme of the study

design for Gnet (net accretion/ erosion measured by limestone block assay) and Gbudget (census-based reef carbonate budgets) to make the statistical procedures more comprehensive. -> Reviewer #2 asks to remove Gbudget data for the lagoonal site (a midshore backreef) from the manuscript, as it is the only data point from a backreef, thus not acceptable to be analysed at the same factor level as the exposed reefs. -> Reviewer #2 suggests a statistical approach considering repeated measurements for the Gnet data calculations.

=> We thank the reviewers for the remarks. In a revised version, we will provide a glossary and a scheme of study design to clarify the data acquisition for the two reef growth parameters assessed. Indeed, we agree that this additional information will help to clarify our results, data collection, and design of experiments. We will further revise parts of the methods section to better detail sampling procedures and design, as well as statistical testing. => We will remove Gbudget data from the lagoonal site from the manuscript as suggested. This will increase statistical power and provide clearer cross-shelf patterns of reef growth for exposed reefs. => We think the suggestion of using a repeated-measures statistical approach for Gnet data analysis is not necessarily adequate for our dataset. We assume that this suggestion is based on a misunderstanding of our description of sampling method and design. We are thankful that the reviewer brought this up. In fact, each limestone block in the assay was deployed and measured only once, which does not call for a repeated-measures analysis. To improve clarity, we suggest to provide a more comprehensive description of the sampling scheme for Gnet to justify the statistical tests that were chosen.

*bioeroding taxa

-> Reviewer #2 requests more information on the bioeroding taxa monitored in this study

=> We agree that providing detailed composition of micro-bioeroder communities (epi- and endolithic organisms) would be beneficial. Unfortunately, we did not investigate

this aspect, as the focus of the study was to measure the mass loss/accretion of the limestone blocks. We will highlight central aspects of the limestone block assay in the revised version of the manuscript. In addition, we will include further details on bio-eroder taxa surveyed in the census-based approach (fish, sea urchins) as requested.

*historical comparison

-> Reviewer #2 recommends to tone down on the historical comparisons

=> We will shorten and tone down our discussion on historical comparison and reef growth trajectories. In addition, we will emphasize the need of comparative data for a better understanding of future projections.

*summary

Overall, we will shorten the revised manuscript and will change the focus on reef growth data of exposed reef sites along a cross-shelf gradient rather than on the physico-chemical environment. Figures and tables will be adjusted accordingly and we will consider the inclusion of the additional references suggested by the reviewers.

We would like to thank the reviewers again for their time and their offer to re-evaluate the revised manuscript. We are looking forward to compiling a revised version, once encouraged by the editor to re-submit our work.

On behalf of all authors, Christian R Voolstra

———————————————————

---

## Author Response (AR1)

**bg-2018-57**

**"Coral reef carbonate budgets and ecological drivers in the central Red Sea - a naturally high temperature and high total alkalinity environment"**

**Response to Editor & Reviewers**

**Associate Editor Decision: Reconsider after major revisions**

13 May 2018 by Jean-Pierre Gattuso

Comments to the Author:

Dear Authors,

I refer to your manuscript bg-2018-57 submitted for publication to Biogeosciences. While the three referees agree that you have collected a valuable data set and that the scientific significance is good to excellent, they identified critical issues. Scientific quality was found to be fair by all referees and two of them also found the presentation quality fair. Hence, there is a need for a major revision.

Please revise the manuscript along the line of your reply to the referees. Note that you have not addressed all the issues which were raised (flow, light, title). I strongly recommend that you thoroughly consider all comments and provide convincing replies because a revised manuscript would be sent again to the two most critical referees for a second round of review.

Best regards,

Jean-Pierre Gattuso
BG Editor

PS: I disagree with referee #2 about the comment s/he made on line 46 of your manuscript. To me, cold water coral communities are not coral reefs and I mostly agree with your statement. To account for reefs which grow in subtropical, marginal environments, you could perhaps revise the sentence slightly as "Coral reef growth is *mostly* limited to warm, aragonite-saturated, and oligotrophic tropical oceans".

We thank the editor and reviewers for their overall positive evaluation. We appreciate their detailed and constructive feedback and their recognition of the relevance of the presented data.

Overall, the reviewers suggest a similar set of improvements of the current manuscript. In the revised manuscript, we have removed all pH monitoring data, the meta-analysis, clarified on sampling design & statistical approach and removed lagoon reef sites to account for a balanced

experimental design. We also provide more detail on bioeroding taxa and have toned down on the historical comparison.

Major changes implemented in the revised manuscript

Removal of pH data
⇨ Following the reviewers' request, all absolute measurements of pH and all related carbonate chemistry data (i.e. $HCO_3^-$, $CO_3^{2-}$, $\Omega_a$, $C_T$, $pCO_2$) have been removed.
⇨ We decided to retain the diurnal fluctuation of pH (expressed as the diurnal standard deviation of pH recorded by the CTDs, and site-specific means+/- SD thereof). These values are independent of the calibration standards used and represent a valuable environmental parameter of biotic feedback in the reef.
⇨ As requested by one of the reviewers, we have included another abiotic variable, salinity, recorded by the CTDs that were not presented in the original manuscript.
⇨ Due to removing a great deal of abiotic data, we have reworked the manuscript to focus more on reef growth. Hence, the new order of parameters and analyses included in the revised manuscript is:
  o $G_{net}$ data
  o $G_{budget}$ data (including related biotic reef census variables)
  o Abiotic parameters (CTDs and seawater samples)
  o Correlations
⇨ Corresponding display figures removed or reworked:
  o Figure 2 (Temperature and pH regimes) was removed
  o Figure 3 (Inorganic nutrients and carbonate system conditions) is now the main figure for abiotic data (Figure 4, revised manuscript). Carbonate system parameters (except measured $A_T$) were removed. CTD-recorded parameters (temperature, salinity, and diurnal pH SD) are presented as boxplots.
  o Table 1 (abiotic data) was updated accordingly and is now Table 4 in the revised manuscript.
  o Table 6 (Global comparison of the carbonate system) was altogether removed.

Removal of distance-based linear models
⇨ Since some datasets partially featured a rather large error and low replication (e.g., part of census data used as input for $G_{budgets}$), we agree that this translates into low statistical power and is not suitable for running a confirmative model. We hence excluded results from the distance-based linear models in the revised manuscript to focus on results from spearman rank correlations (Table 5 in revised manuscript, which is the updated version of old Table 4).

Study design and statistical analyses
⇨ As suggested by the reviewers, we now present the sampling design in greater detail. Figure 1 in the revised manuscdript now now a map of the reef sites including schematic diagrams of the sampling designs for $G_{net}$, $G_{budget}$, and the monitoring of abiotic parameters.
⇨ We added a glossary of the measured and calculated reef growth metrics and related biotic variables (Table 1 in the revised manuscript).

⇨ We removed data for the sheltered sites (backreefs and backreef lagoon), to account for a balanced study design. Consequently, we repeated statistical analyses on the new datasets under a simplified study design. Specifically, $G_{net}$ under the 2-factorial design of the fixed factor 'reef' of three levels (offshore, midshore, nearshore) and the random factor deployment time (6, 12, 30 months); $G_{budget}$ under the 1-factorial design of the fixed factor 'reef' of three levels (offshore, midshore, nearshore)
  - Table 2 ($G_{net}$) and Table 3 ($G_{budget}$) were updated accordingly.
  - Also, we show $G_{net}$ in a new Figure 2 and $G_{budget}$ as a new Figure 3 as bar plots

Bioeroding taxa
⇨ The goal of the study was to focus on reef growth and the budget aspect of carbonate accretion/erosion rather than on the nature of bioeroder communities. Therefore, we did not account for community composition settled on the limestone blocks. However, we added information about calcifier and bioeroder groups observed over time based on visual inspection of blocks. The manuscript therefore expands the current knowledge on succession of endolithic communities.
⇨ As suggested by reviewer #2, the manuscript now more clearly addresses the bioeroders that were quantified (i.e., parrotfish and seas urchins) as "grazing and epilithic bioeroders" (as opposed to endolithic microeroders).
⇨ We now differentiate all parrotfish into scrapers and excavators in reference to Alwany et al (2009) that previously studied parrotfish bioerosion rates in the Red Sea (Table S7). Unidentified parrotfish were put together in the category 'other' scaridae, which exclusively contains specimens belonging to the genus *Scarus*, predominantly comprised of scrapers (Green and Bellwood, 2009. These revisions provide a more comprehensive effort in the characterization of parrotfish count data and their integration in the $G_{budgets}$.

Historical comparison
⇨ We toned down our discussion of the historical comparison according to the reviewer's suggestions. Accordingly, we have changed the title of this section from "Historical perspective" to "Reef growth trajectories in the Red Sea". In the revised discussion, the emphasis is now on the importance of future studies to understand the trajectory of the wider region. We still discuss previous historical Red Sea reef growth studies in brief, which we consider an essential reference.

Manuscript title
⇨ Due to the substantial revision which shifted the focus of the manuscript to the carbonate budgets, we rephrased the title as follows: "Coral reef carbonate budgets and ecological drivers in the central Red Sea - a naturally high temperature and high total alkalinity environment"

As the editor and reviewers will see, the above implemented changes significantly shortened the manuscript and provide stronger focus on reef growth and budget data. Figures and tables were adjusted accordingly and we considered all comments by all reviewers. We provide a detailed response below.

We hope we have satisfactorily addressed all concerns and suggestions.

Thank you on behalf of all authors,

Christian R Voolstra

**Anonymous Referee #1 Decision: Reconsider after major revisions**

This paper presents an interesting study that aimed to characterize the effects of biotic and abiotic factors on the carbonate budget of different sites of a reef in the Red Sea. This study is based on a pretty extensive dataset that combines benthic community monitoring, determinations of organisms' calcification, monitoring of chemical and physical parameters, etc. The paper could be pleasant to read but the discussion drags into discussing obvious facts and do not really discussed the results and the significance of this study.

⇨ We thank the reviewer for the overall positive evaluation and constructive feedback.

Aside of the specific comments listed below I have 2 major comments:

- First, the determination of pH and then the carbonate chemistry was incorrect. pH on the NBS scale is not the appropriate scale to determine seawater pH. Furthermore, the authors used the R package seacarb and the pH free-scale (while measurements were done on the NBS scale) for all the calculations of the carbonate chemistry, which is incorrect. A quick test on CO2sys shows that for a pH of 8.1 (NBS scale), a TA of 2300 and a temperature of 28, the corresponding pCO2 is 507 and the saturation state 3.19. Using the same pH value on the free –scale yields to a pCO2 of 472 and a saturation state of 3.34.

⇨ We agree with the reviewer that the collected pH data and carbonate chemistry calculations based on these are suboptimal. As a consequence, we removed all pH data from the manuscript. We decided to retain the diurnal pH fluctuation as a parameter, as they are not dependent on absolute pH values and informative for readers.

- Second, one of the main conclusions of the authors is that TA is the main driver of reef calcification. While I do not question the mathematical approach of the authors, I do not believe that the relative minor differences in TA between sites can explain the large differences in Gnet and Gbudget between sites. From a biological and biogeochemical point of view, TA differences of 15-50 umol are not highly significant and are not sufficient to explain the large differences in precipitation and erosion between sites. A positive effect of TA on coral calcification has indeed been shown in previous studies but the TA enhancement was of several hundreds of mol.

⇨ We removed the distance-based linear model from the manuscript. The discussion of TA was accordingly shortened and adjusted in the revised manuscript.

It now reads:

"The increase in TA is often associated with increased carbonate ion concentration and aragonite saturation state which facilitate the precipitation of carbonates supporting the performance of reef-builders (Albright et al., 2016, 2018; Langdon et al., 2000; Schneider and Erez, 2006; Silbiger et al., 2014). We identified a positive correlation of TA with reef growth in our dataset. The difference in TA across our study sites was small, but in the range of a natural cross-shelf

difference reported from other reefs (e.g. reefs in Bermuda, 20 - 40 μmol TA kg$^{-1}$, Bates et al., 2010), and as high as 50 μmol kg$^{-1}$, the TA enrichment that enhanced net community calcification in a reef-enclosed lagoon (Albright et al., 2016). On the other hand, high calcification rates can deplete TA, whereas dissolution of carbonates can enrich TA measurably, specifically in (semi) enclosed systems (Bates et al., 2010), which we do not observe along the cross shelf gradient. It remains to be further investigated how TA dynamics across the shelf relate to reef growth processes."

Referee #1 specific/technical comments

L-25-26: "Beneficial and detrimental factors" is not clear, what does that refer to?

⇨ We adjusted the wording to make a better connection between the sentences.

Now reads:

"The structural framework provided by corals is crucial for reef ecosystem function and services, but high seawater temperatures can be detrimental to the calcification capacity of reef-building organisms. The Red Sea is very warm, but total alkalinity is naturally high and beneficial for reef accretion. To date, we know little about how such beneficial and detrimental abiotic factors affect each other and the balance between calcification and erosion on Red Sea coral reefs, that is overall reef growth, in this unique ocean basin."

L-49-50: Sediment export is also very important in a reef budget.
L-52-53: What about flow and light? Those two parameters are critical but not discussed at all in the manuscript.

⇨ We added these two aspects as potentially contributing factors to the Introduction and later briefly refer to these aspects in some parts of the Discussion.

Introduction now reads:

"The balance of carbonate loss and accretion is influenced by biotic and abiotic factors. On a reef scale, the main antagonists are calcifying benthic communities, e.g., scleractinian corals and coralline algal crusts, opposed by grazing and endolithic bioeroders, e.g., parrotfish, sea urchins, microbioeroding chlorophytes, boring sponges, and other macroborers (Glynn, 1997; Hutchings, 1986; Perry et al., 2008; Tribollet and Golubic, 2011). The export or loss of carbonate as sediments is considered an essential part, in particular in the wider geomorphic perspective of reef carbonate production states (Cyronak et al., 2013; Perry et al., 2008, 2017). Temperature and carbonate chemistry parameters (e.g., pH, total alkalinity TA, and aragonite saturation state Ω$_a$, pCO$_2$) have been identified as important players in regulating these carbonate accretion and erosion processes (Albright et al., 2018; Schönberg et al., 2017). Furthermore, different light regimes across depths, water flow, and wave exposure can alter the rates of reef-formation processes (Dullo et al., 1995; Glynn and Manzello, 2015; Kleypas et al., 2001)."

L-57: It would be good for the reader to start by clearly defining what Gbudget and Gnet are referring to in this manuscript.

⇨ We introduce key-concepts of reef growth in the beginning of the Introduction and in the last part outlining the motivations and goals of our study, where we detail the two metrics that we present (final section of the Introduction). Additionally, we now provide a glossary (Table 1 in the revised manuscript).

Now reads:

"To comparatively assess the persistence of reef frameworks at regional and global scales, a census-based reef carbonate budget (*ReefBudget*) approach that integrates reef site-specific ecological data into the calculation of the erosion-accretion balance was introduced recently (Kennedy et al., 2013; Perry et al., 2012, 2015). The *ReefBudget* approach shows that 37 % of all current reefs studied are in a net-erosive state (Perry et al., 2013). For the Caribbean, it revealed a 50 % decrease of reef growth compared to historical mid- to late-Holocene reef growth (Perry et al., 2013). Indeed, the use of carbonate budgets provided valuable insight into the reef growth trajectories in the Seychelles, where surveys conducted since the 1990ies provide important ecological baseline data that were employed in the reef growth calculations (Januchowski-Hartley et al., 2017). Other studies of carbonate budgets highlight the susceptibility of marginal coral reefs to ocean warming and acidification (Couce et al., 2012). Such marginal reefs are found in the Eastern Pacific or in the Middle East in the Persian/Arabian Gulf, where reefs exist at their environmental limits, e.g., low pH or high temperatures, respectively (Bates et al., 2010; Manzello, 2010; Riegl, 2003; Sheppard and Loughland, 2002). […]

We present net-accretion/erosion rates ($G_{net}$) measured *in situ* using limestone blocks deployed in the reefs, which simultaneously capture the rates of epilithic accretion and epilithic and endolithic bioerosion. We also apply a census-based approach adapted from the *ReefBudget* protocol (Perry et al., 2012) to estimate reef growth on an ecosystem scale, as the net carbonate production state or carbonate budget ($G_{budget}$). To achieve this, we assessed the abundances and calcification rates of major reef-building coral taxa (*Porites*, *Pocillopora*, and *Acropora*) and calcifying crusts (e.g., coralline algae), together with the abundances of epilithic bioeroders (parrotfish and sea urchins) at our study sites. These ecological data were integrated with the Red Sea site- and species-specific calcification or erosion rates to calculate $G_{budget}$. We then present data from an environmental monitoring of reefs along an environmental cross-shelf gradient during winter and summer. Finally, we correlate potential abiotic and biotic drivers with the two reef growth metrics, $G_{net}$ and $G_{budget}$. Our study provides a broad and first insight into reef growth dynamics and a comparative baseline to further assess the effects of ongoing environmental change on reef growth in the central Red Sea."

L-59: Not clear, calcification rate of the reef or corals?

⇨ We clarified this by adding "corals and coralline algae".

L61-63: This is not entirely true. Calcification rates of some organisms are actually enhanced by warming. Also, the temperature threshold between decrease in

calcification and death is very thin for corals.

⇨  Reworded accordingly:

"TA and Ωa, positively correlate with calcification rates (Marubini et al., 2008; Schneider and Erez, 2006), and while calcification rates of corals and coralline algae increase with higher temperature, they have upper thermal limits (Jokiel and Coles, 1990; Marshall and Clode, 2004; Vásquez-Elizondo and Enríquez, 2016)."

L-64-65: This is still an hypothesis that is highly debated.

⇨  Thank you for pointing this out. We agree and adjusted the sentence:

"Arguably, calcification under these conditions could become energetically costlier (Cai et al., 2016; Cohen and Holcomb, 2009; Strahl et al., 2015; Waldbusser et al., 2016)."

L-67-69: What about a crown-of-thorn starfish outbreaks?

⇨  We added this information. Additionally, we added grazing sea urchin outbreaks:

"Locally impaired reef growth due to an increased intensity or frequency of extreme climate events (Eakin, 2001; Schuhmacher et al., 2005), human impacts including pollution and eutrophication (Chazottes et al., 2002; Edinger et al., 2000), and other ecological events such as population outbreaks of grazing sea urchins or crown-of-thorn starfish that feed on coral can induce reef framework degradation (Bak, 1994; Pisapia et al., 2016; Uthicke et al., 2015)"

L-86-87:  Tambutté et al. 2011 is not the best reference here.

Tambutté et al. 2011 focuses on calcification at the cellular level, but also provides a summary of environmental influences. To complement this, two more references describing community scale responses and environmental variables were added:

Albright, R., Caldeira, L., Hosfelt, J., Kwiatkowski, L., Maclaren, J. K., Mason, B. M., Nebuchina, Y., Ninokawa, A., Pongratz, J., Ricke, K. L., Rivlin, T., Schneider, K., Sesboüé, M., Shamberger, K., Silverman, J., Wolfe, K., Zhu, K. and Caldeira, K.: Reversal of ocean acidification enhances net coral reef calcification, Nature, advance online publication, doi:10.1038/nature17155, 2016.

Langdon, C., Takahashi, T., Sweeney, C., Chipman, D., Goddard, J., Marubini, F., Aceves, H., Barnett, H. and Atkinson, M. J.: Effect of calcium carbonate saturation state on the calcification rate of an experimental coral reef, Global Biogeochemical Cycles, 14(2), 639–654, doi:10.1029/1999GB001195, 2000.

L120-121: The pH probes from CTD are unreliable and not recommended to characterize pH in seawater.
L127-144: See my major comment#1

⇨ We have removed all absolute measurements of pH and all related carbonate chemistry data in the revised manuscript

⇨ We decided to keep the pH data of diurnal variability, expressed as the diurnal standard deviations of pH recorded by the CTDs. These values are independent of the calibration standards used and represent a valuable environmental parameter regarding implications of biotic feedbacks in the reef, which have recently received much attention from the scientific community (see e.g. Camp et al. 2018 and references therein). As such, they might provide useful baseline info for the reader.

Camp, E. F., Schoepf, V., Mumby, P. J., Hardtke, L. A., Rodolfo-Metalpa, R., Smith, D. J. and Suggett, D. J.: The Future of Coral Reefs Subject to Rapid Climate Change: Lessons from Natural Extreme Environments, Front. Mar. Sci., 5, doi:10.3389/fmars.2018.00004, 2018.

L-146-156: It is not clear how many blocks were deployed and retrieved at each sites/time points.

⇨ We revised the text accordingly and now provide the sampling design scheme for the limestone block assay in Fig. 1 (a).

"[… ] Four replicate blocks were deployed at the reef sites for each exposure timeframe (Fig. 1 a), where they were exposed to the natural processes of calcification and erosion, for 6 months (September 2012 - March 2013), for 12 months (June 2013 - June 2014), and for 30 months each (January 2013-June 2015)."

L-182-188: I would recommend adding the supplementary material text here.

⇨ The principle of the $G_{budget}$ calculations is defined by the reference to Perry et al., but additionally we extended the text and provide a more extended summary of the procedure of $G_{budget}$ calculations. We also refer to Fig.1 (b), and Fig.3 (a) which illustrate how $G_{budgets}$ were derived. Nevertheless, the detailed description of the $G_{budget}$ calculation stayed in the supplement to keep the Materials and Methods concise.

Text now reads:

"Ecosystem scale reef carbonate budgets, $G_{budget}$ [kg m$^{-1}$ y$^{-1}$], were determined following the census-based *ReefBudget* approach by Perry et al. (2012) (Table 1). $G_{budget}$ incorporates local census data, site-specific net-accretion/erosion data ($G_{net}$ over 30 months) and calcification data (buoyant weight measurements) collected for the present and from a previous study (Roik et al., 2015). Importantly, the approach incorporates epilithic bioerosion, which is based on abundance rather than assessment of actual bite or erosion rates; therefore, parrotfish and sea urchin census data collected in this study are readily employed in the *ReefBudget* calculations using bite and erosion rates from the literature (Alwany et al., 2009; Perry et al., 2012). In summary, site-specific benthic calcification rates ($G_{benthos}$, kg m$^{-1}$ y$^{-1}$), net-accretion/erosion rates of reef "rock" surface area ($G_{netbenthos}$, kg m$^{-1}$ y$^{-1}$), and epilithic erosion rates by sea urchins ($E_{echino}$, kg m$^{-1}$ y$^{-1}$) and parrotfishes ($E_{parrot}$, kg m$^{-1}$ y$^{-1}$), were determined for the $G_{budget}$ calculations (Fig. 1 (b) and

Fig. 3 (a)). A detailed account of Red Sea specific calculations and modifications of the *ReefBudget* approach employed in this study are outlined in the supplementary materials (Text S1, Equation box S1-3, and Tables S2-7)."

Results: It is a bit strange to separate the pH from the other carbonate chemistry parameters. L270: Aside of the problem with the scales, there are some discrepancy between pH continuous and pH discrete. As a result, if the parameter of the carbo chem were calculated from the pH continuous the results for the saturation state etc would be completely different (especially for the inshore sites where continuous pH was much higher than the discrete one). Were any cross-calibrations made between the pH continuous and the pH discrete?

⇨ The pH values and carbonate chemistry were removed from the revised manuscript.

L-304: n= ?

⇨ Sample size information was added to the text in the methods chapter (chapter 2.2. in the revised manuscript). Additionally, sample size information is provided in the study design scheme (Fig.1 a).

"Net-accretion/erosion rates ($G_{net}$) were assessed using a "limestone block assay". Blocks cut from "coral stone" limestone were purchased from a local building material supplier in Jeddah, KSA, and each block was fixed with one stainless steel bolt to aluminum racks permanently deployed at the monitoring station of each reef site (a total of 36 blocks, n = 4, Fig. S1). The blocks were oriented in parallel to the reef slope with one side facing up while the other side was facing down towards the reef. Block dimensions were 100 x 100 x 21 mm with an average density of $\rho = 2.3$ kg L$^{-1}$."

L323: Why is pH continuous used in the model while the carbonate chemistry parameters are calculated from pHdiscrete?

⇨ The pH values and carbonate chemistry were removed from the revised manuscript.

L-326-327: As explained before it is hard to explain biologically how such small changes in At can explain 65% of Gnet.

⇨ We agree. Since a relatively large error is associated with some of the measurements, e.g. abundance data from the reef census that is used as input for $G_{budgets}$ estimations, the within-group variation is likely too high for running a confirmative model. We excluded the results from the distance-based linear models from the revised manuscript and focus our exploration of environmental drivers on the results from spearman rank correlations (now in Table 5, which is the updated version of Table 4)
⇨ Given that the difference in TA values across the shelf-gradient still correlate with the patterns of $G_{net}$ and $G_{budget}$ as, the relationship is now discussed as follows:

"The increase in TA is often associated with increased carbonate ion concentration and aragonite saturation state which facilitate the precipitation of carbonates supporting the performance of

reef-builders (Albright et al., 2016, 2018; Langdon et al., 2000; Schneider and Erez, 2006; Silbiger et al., 2014). We identified a positive correlation of TA with reef growth in our dataset. The difference in TA across our study sites was small, but in the range of a natural cross-shelf difference reported from other reefs (e.g. reefs in Bermuda, 20 - 40 µmol TA kg$^{-1}$, Bates et al., 2010), and as high as 50 µmol kg$^{-1}$, the TA enrichment that enhanced net community calcification in a reef-enclosed lagoon (Albright et al., 2016). On the other hand, high calcification rates can deplete TA, whereas dissolution of carbonates can enrich TA measurably, specifically in (semi) enclosed systems (Bates et al., 2010), which we do not observe along the cross shelf gradient. It remains to be further investigated how TA dynamics across the shelf relate to reef growth processes"

L-347-348: O2 could indeed be an important driver, why was O2 not determined?

⇨ We agree and thank the reviewer for pointing this out. In fact, O$_2$ was measured via CTDs throughout the monitoring period and is now included in the revised manuscript.

L-351-354: and the wrong scale was used.
L-358-359: The high temperature of the Red Sea also explains the high omega.

⇨ The pH values and carbonate chemistry were removed from the revised manuscript.

L-365-366: This is already well known:

⇨ This is an introductory sentence to the section to provide a basis and a background for the reader.

L-376: pH goes down to 7.3 in Camp et al. 2017 (Sci Rep)

⇨ The pH values and carbonate chemistry data and discussion were removed from the MS, which renders this point irrelevant.

L-381: pH variations are not an indication of fluctuation in AT.

⇨ We agree and we rephrased accordingly. Initially, we intended to say that the variation of pH is indicative of the dynamics in other carbonate species, but also in other seawater chemistry variables, which all are influenced by the biotic feedbacks.

Text now reads:

"The strong variability of diurnal pH on the other hand is supposedly influencing performance of calcifiers and bioeroders. Previously, small scale pH anomalies were demonstrated to correlate with net accretion dynamics by showing higher net accretion prevailing in sites of less variable pH conditions locally (Price et al., 2012; Silbiger et al., 2014), which reflects the pattern that we observe in our study. pH fluctuation is a biotic feedback signature in reef habitats, which entails changes in sea water chemistry caused by dominant biotic processes, i.e., calcification, carbonate dissolution, and respiration/photosynthesis (Bates et al., 2010; Silverman et al., 2007a;

Zundelevich et al., 2007). This fluctuation accompanies changes in other factors such as carbonate system variables, e.g. $pCO_2$, aragonite saturation state (Shaw et al., 2012; Silbiger et al., 2014), which again can modify the antagonistic processes of calcification and bioerosion, i.e., dissolution (e.g., Andersson, 2015; Langdon et al., 2000; Tribollet et al., 2009)."

L-405: PO43- is not "a source of energy".

⇨ Rephrased accordingly.

Now reads:

"Interestingly, our study also identified $PO_4^{3-}$ concentration as an abiotic correlate of reef growth. In the Red Sea high N:P ratios indicate that P is a limiting micronutrient, e.g. for phytoplankton (Fahmy, 2003). $PO_4^{3-}$ is not only essential for pelagic primary producers, but also for reef calcifiers and their photosymbionts, such as the stony corals and their micro-algal *Symbiodinium* endosymbionts (Ferrier-Pagès et al., 2016)."

L-446-452: Isn't that the case in a lot of reefs?

⇨ We removed this sentence.

L489-490: Was this decrease in calcification linked to bleaching events, etc?

⇨ The referenced study based on coral skeleton core analyses, and describes a decadal trend from 1980 to 2010 for one coral species in the central Red Sea. In this regard, the study points out to "investigate the potential for subliminal (nonvisible) effects of recently rising SSTs" and their data show that "skeletal growth histories provide evidence for prolonged thermal stress that has inhibited the recovery of *D. heliopora* skeletal growth since 1998."

**Anonymous Referee #2 Decision: Reconsider after major revisions**

This is a mostly well-written MS, and the language and style is above average, the figures and tables are of high quality. The study concerns a highly relevant topic from an interesting marine area that may be less well studied in this context than other coral reefs (Berumen et al. 2013; Schonberg et al. 2017). The MS is based on a large work effort and produced a large amount of valuable data. I compliment the authors for tackling such a timely and complex task.
However, I think the study has a number of shortcomings that need to be addressed before the MS can be published. In my opinion the data analyses were not correctly performed and will need to be redone. This will likely lead to the need to re-write a few parts of the MS. The Methods section may require some more detail for clarity, and maybe the terminology could be simplified or streamlined to make it easier for the reader.
In particular the assessment of the bioerosion needs to be clearer. I have further listed references that may be useful in the context. These cannot all be included and are subject to the choice of the authors. The title seems to be a bit misleading. I recommend publication after re-analysing the data and MS revision. It may thus be necessary to re-review the MS.

⇨ We thank the reviewer for the overall positive evaluation and constructive feedback. We respond to the specific comments in a point-by-point manner below.

DATA ANALYSIS: In my opinion the data evaluation is faulty, and the statistical models and means were not built respecting the existing data hierarchy. This matters even more as the sample size of 4 replicate blocks and 6 transects per subsite is very small for data that can be expected to be highly patchy and variable. To my understanding, there are two independent "between" factors, "distance from shore" (3 levels) and "time of exposure" (3 levels). Then there is the within factor "reef area" or "hydrodynamic exposure level" (backreef, forereef, lagoon or exposed, sheltered). The latter "within" levels are not independent within a given reef and would need to be nested in "distance from shore". Ignoring this data hierarchy resulted in pseudoreplication and a higher test power than actually justified (S12). This is the case for the 6 and 12 mo analyses, and the 30 mo analysis seems to include only 4 subsites, all at the same data level (near-fore, mid-fore, mid-lagoon and off-fore). I assume that the means are therefore also not correctly calculated (not stepwise). Sadly, this does not only apply to the block data. All figures and tables will need to be restructured accordingly, means will have to be calculated stepwise, following the same data hierarchy as the statistical models. It would be good to have some sort of schematic figure that visualises the data design and hierarchy. Maybe "lagoon" could be left out of the 30 mo analysis to make things clearer and possibly more powerful to pick up effects/trends? I think it would be acceptable to define "lagoon" and "backreef" as "sheltered" and give them the same data status, but it is not acceptable to include near-fore, mid-fore and off-fore at the same level as mid-lagoon. Unfortunately, the situation will cause significantly more work effort and potentially the need for rewriting parts of the MS.

⇨ We thank the reviewer for his time and thoughtful remarks. We consider it a valid suggestion to simplify the study design in order to make patterns of the cross-shelf gradient clearer. Accordingly, we have removed the data points for the sheltered sites (backreefs and backreef

lagoon), which contributed to  an unbalanced study design. Statistical tests are now performed on a simplified, but fully crossed study designs, i.e. $G_{net}$ data under the 2-factorial design of 'reef site' (offshore, midshore, nearshore) and 'deployment time' (6, 12, 30 months). In addition, it is now clarified in the Material and Methods section (ll. 131-143) that the block assay does not follow a repeated-measures design, but that each block was measured once. The $G_{budget}$ data is analyzed under a 1-factorial design with the factor 'reef' (offshore, midshore, nearshore).

⇨ Further, we now make a clearer distinction between the two reef growth metrics, $G_{net}$ and $G_{budget}$, by including a new Figure 1, which provides a map of the reef sites including schematic diagrams of the sampling designs for $G_{net}$, $G_{budget}$, and the abiotic monitoring. Additionally, we provide a glossary of all acronyms used for the various metrics (new Table 1).

THE CHOICE OF THE BIOERODER TAXA AND THE ASSESSMENT IS NOT ENTIRELY CLEAR. The bioeroders were assessed in two groups – dominant epilithic bioeroders via biomass estimates, and endoliths in a block assay. It is not quite clear whether the counted scarids only represented "excavators", i.e. fishes that bite and break calcium carbonate or whether the "others" contained fishes that mainly eat fleshy algae and thus dilute the overall value? Please clarify in the Methods.

⇨ Thank you for pointing this out. All scarids counted in the present study could be assigned to either of two herbivore functional groups: excavators, or scrapers, which both remove calcium carbonate, albeit at different rates (Green and Bellwood 2009). The 'other' category puts together scarids of the genus *Scaru'* for which a species identification was not possible. As the vast majority of *Scarus* are categorized as scrapers (Green and Bellwood 2009), we herewith categorized unidentified *Scarus* ("Other *Scarus*") as such. Further, in order to make bioeroder assessment more comprehensive, we added more detail on the parrotfish survey:

Text reads:

"We focused on the most abundant bioeroding parrotfish species in the Red Sea (Table S7), which encompasses two herbivorous functional groups, excavators and scrapers (Green and Bellwood 2009). Most abundant across study sites were the excavators *Chlorurus gibbus*, *Scarus ghobban*, and *Cetoscarus bicolor*, and the scrapers *Scarus frenatus*, *Chlorurus sordidus*, *Scarus niger*, and *Scarus ferrugenius*, following Alwany et al. (2009). Additionally, we counted *Hipposcarus harid*, which occurred frequently at the study sites, along with members of the genus *Scarus* that could not be identified to species level and accounted for in the category 'Other *Scarus*'. *H. harid* and *Scarus spp.* were broadly categorized as scrapers (Green and Bellwood, 2009)."

⇨ For 'Other *Scarus*', bioerosion rates were based on averages from *Scarus* species provided by Alwany et al (2009), assuming that all *Scarus* sp. are scrapers and potentially contribute to bioerosion.
⇨ In the text, we refer to the supplements, where surveys, categories, erosion rates are detailed:

"A detailed account of Red Sea specific calculations and modifications of the *ReefBudget* approach employed in this study are outlined in the supplementary materials (Text S1, Equation box S1-3, and Tables S2-7)."

The blocks were weighed but not otherwise assessed? 30 months exposure to settlement is not that long, and unless you consistently found larger borers in the blocks you probably have to assume that you are still capturing earlier successional communities. Can provide more details in the results what borers were present in the 6, 12 and 30 mo blocks?

⇨ We did not quantify the endolithic bioeroder community composition in detail in the limestone blocks as our study motivation was to assess reef growth and budget aspects of carbonate loss/accretion rather than on the nature of bioeroders.
⇨ However, we now provide accounts from a visual inspection of calcifier and bioeroder groups present or absent from block surfaces. We have added this information to the Material and Methods:

"The measurements represent the result of calcification and bioerosion processes impacting the deployed blocks. Visible traces of boring endolithic fauna were only found on the surfaces of blocks recovered after 12 and 30 months as presented in Fig. 2 (c)-(f). A brief visual inspection of the block surfaces after retrieval showed colonization by coralline algae, bryozoans, boring sponges, small size boring worms and clams, as well as parrotfish bite-marks. No coral recruits were identified."

⇨ We added a new section to the Discussion that refers to studies of the succession patterns of endolithic communities, see section 4.1.2
⇨ We agree that a 30-month exposure of blocks does not capture the full succession of bioeroders. We point this out in the Discussion:

"[…] In our study, the increase of $G_{net}$ between the 12- and 30-month deployment (~0.30 kg m$^{-2}$ yr$^{-1}$ on average in the nearshore and offshore site) indicates that calcifying and eroding communities were still in a state of succession. We cannot rule out that the blocks deployed for 30 months still represent an immature community and underestimate maximal calcification and erosion rates."

THE BIOERODER DATA MAY PERHAPS NOT BE REPRESENTATIVE, AND THERE MAY BE A RISK OF COMPARING APPLES WITH PEARS. E.g. William Kiene has assessed temporal successions of coral reef biota in a comparatively pristine environment settling onto experimental blocks. According to his data, the present 30 mo blocks may still be reflecting a developing phase of that settlement. The block assay was not controlled by implementing non-erodable blocks so that the net value could be corrected with data that matched the experimental situation. Calcification and epilithic bioerosion was assessed in situ and thus in all likelihood concerned a mature community, the blocks assessed borers and may not be as representative. In any case, it would be good if the authors could provide more detail on how the blocks were deployed (fixed to the reef and allowing lateral invasion and grazer access, or on a rack only allowing larval settlement and probably excluding urchins) and what were their observations after retrieval. This would enable a better understanding what the data represent.

The comparison of the present data with past data from the Gulf of Aqaba may also be a case of apples and pears (see below). Overall the interpretation needs to be cautious. It is a pity that dominant borer distributions were not assessed in situ, which is a significant data lack for the Red Sea and could have been done while assessing the benthos. The epilithic bioeroders have always received more interest. In any case, this needs to be reflected in the wording, you did not assess bioeroders on the reef, but only the dominant epilithic bioeroder-grazers (fish, urchins).

⇨ In the Results we now clarify that the measurements taken do not allow us to differentiate between mere erosion and calcification, but rather represents the result of net-accretion/-erosion.

" Net-accretion/-erosion rates $G_{net}$ were measured in assays over periods of 6, 12, and 30 months in the reef sites along the cross-shelf gradient. These measurements represent the result of calcification and bioerosion processes impacting the deployed blocks."

⇨ In order to visualize the deployment technique of the limestone blocks, we now show an image of the rack in the monitoring site in Figure S1 of the revised manuscript and provide further detail in the Material and Methods :

"Net-accretion/-erosion rates ($G_{net}$) were assessed using a "limestone block assay". Blocks cut from "coral stone" limestone were purchased from a local building material supplier in Jeddah, KSA, and each block was fixed with one stainless steel bolt to aluminum racks permanently deployed at the monitoring station of each reef site (a total of 36 blocks, n = 4, Fig. S1). The blocks were oriented in parallel to the reef slope with one side facing up while the other side was facing down towards the reef. Block dimensions were 100 x 100 x 21 mm with an average density of $\rho = 2.3$ kg $L^{-1}$. Blocks were dry-weighed before and after deployment on the reefs."

THE DATA NEED TO BE USED AND INTERPRETED WITH CAUTION. The data are still valuable and good to have. However, in view of some of the above comments, and the extremely large error values, the interpretation cannot be generalised too much and needs to make allowances – in a larger dataset with smaller error the data may have shown different patterns. The temporal analysis across decades appears risky in that the present study was conducted in the central Red Sea, with a different protocol and focus, the earlier studies in the Gulf of Aqaba. I think it would be better to resist and not try to construct a long-term trend, but rather point out the lack of comparable data? Also, when data are plotted respecting the data hierarchy I can only see significant cross shelf trends for the parrots.

⇨ Agreed. We now are more outspoken with regard to the potential considerable error associated with the census-based data. Furthermore, we have toned down with regard to the historical comparison, while at the same time emphasizing on the importance of targeting trajectories in a more extensive future study, see chapter 4.4.

SOME PARTS WERE CONFUSING. I found it at times difficult to keep track of the data and terms. Is it correct that the bioerosion (net) value was called Gnet, net accretion/erosion values and Gnetbenthos in different parts of the MS? Gbenthos and Gnetbenthos are too similar, wouldn't it be better to clearly separate those terms? What does G stand for? Growth? Why then

is E marked separately, it is negative growth? Wouldn't it be easier for the reader to skip the Gs and Es and use more intuitive words? To abbreviate in situ accretion, in situ bioeroder biomass = approximation of bioerosion, assay net accretion/bioerosion? I am not sure what you mean with "cumulative" data in the block assay over time. You did not repeatedly measure the same blocks, right? And if you added data from the second and third block set to the first that would be inappropriate, as the successional stages change over time. You need to evaluate the block sets separately and can only display a 30 mo net accretion/erosion situation from blocks that were exactly that long in the water. Please clarify what you did. For urchins and parrots all 6 subsites were assessed for biomass and abundances – why are only 4 subsites presented for the respective bioerosion data?

⇨ Apologies. To address this, we clarified terminology of reef growth metrics and various biotic variables by adding a glossary in Table 1 of the revised manuscript. Furthermore, data collection designs for $G_{net}$, $G_{budget}$, and the abiotic monitoring are now presented in a schematic figure (Figure 1). Additionally, we have simplified the study design as per reviewer suggestion by removing data related to the backreefs ('lagoon'), which gave rise to an imbalanced study design in the original manuscript. The manuscript now only contains data related to forereef habitats.

THE TITLE COULD BE TWEAKED TO BETTER REPRESENT THE CONTENTS OF THE PAPER. I feel that putting the budget first is misleading, because these data are only a small part of the paper, and the huge error value makes the overall budget estimate a highly unreliable value. Would it be OK to change the title to something more along the line of: Ecological drivers of coral reef carbonate cycling in the central Red Sea – a high temperature, high total alkalinity environment

Berumen ML, Hoey AS, Bass WH, Bouwmeester J, Catania D, Cochran JE, Khalil MT, Miyake S, Mughal MR, Spät JL, Saenz-Agudelo P. The status of coral reef ecology research in the Red Sea. Coral Reefs. 2013 Sep 1;32(3):737-48.

Schönberg CH, Fang JK, Carballo JL. Bioeroding sponges and the future of coral reefs. In Climate Change, Ocean Acidification and Sponges 2017 (pp. 179-372). Springer, Cham.

⇨ Since carbonate budgets now dominate the content of the revised manuscript, we would like to kindly retain them in the title. The term essentially conveys the actual content of the study to the reader. However, we have chosen to adapt the word structure as suggested by the reviewer in this comment to "Coral reef carbonate budgets and ecological drivers in the central Red Sea - a naturally high temperature and high total alkalinity environment".

Referee #2 specific/technical comments

l. 25: Complicated sentence. Can you simplify by dropping sub-clauses?

Now reads:

"The structural framework provided by corals is crucial for reef ecosystem function and services, but high seawater temperatures can be detrimental to the calcification capacity of reef-building organisms. The Red Sea is very warm, but total alkalinity is naturally high and beneficial for reef accretion. To date, we know little about how such beneficial and detrimental abiotic factors affect each other and the balance between calcification and erosion on Red Sea coral reefs, that is overall reef growth, in this unique ocean basin."

l. 36: … AT correlated well and positiveLY with reef growth …

⇨ Adjusted accordingly.

l. 46: Incorrect statements. There are highly functional and very important cold-water reefs. Even warm water coral reefs can exist and thrive under oligotrophic conditions. And to some degree reefs can occur in marginal or even hostile environments. Suggestion for rewording: Positively accreting warm-water coral reefs usually occur in aragonite-saturated and oligotrophic tropical oceans, where pivotal ecosystem functions can be best maintained…

⇨ Since this is an introductory sentence, we aimed for a short and concise statement. Coral reefs were originally described to be shallow-water structures reaching the ocean's surface, therefore acting as physical structures breaking waves. Cold-water reefs commonly form banks rather than reefs. We would like to follow the editor's suggestion and phrase the sentence as follows:

"Coral reef growth is mostly limited to warm, aragonite-saturated, and oligotrophic tropical oceans and is pivotal for reef ecosystem functioning (Buddemeier, 1997; Kleypas et al., 1999). The coral reef framework not only maintains a remarkable biodiversity, but also provides highly valuable ecosystem services that include food supply and coastal protection, among others (Moberg and Folke, 1999; Reaka-Kudla, 1997)."

l. 54: Too simple a view? OK, maybe you have to keep it short… Still, maybe have a look at

Perry CT, Harborne AR. Bioerosion on modern reefs: impacts and responses under changing ecological and environmental conditions. InCoral Reefs at the Crossroads 2016 (pp. 69-101). Springer, Dordrecht

Schönberg CH, Fang JK, Carreiro-Silva M, Tribollet A, Wisshak M. Bioerosion: the other ocean acidification problem. ICES
Journal of Marine Science. 2017 May 1;74(4):895-925.

Glynn 1997 and Glynn and Manzello 2015 are the same thing. Please choose one.

⇨ As suggested we have added some more detail with regard to carbonate accretion and loss. We also incorporated some of the suggested literature:

"Coral reef growth is mostly limited to warm, aragonite-saturated, and oligotrophic tropical oceans and is pivotal for reef ecosystem functioning (Buddemeier, 1997; Kleypas et al., 1999).

The coral reef framework not only maintains a remarkable biodiversity, but also provides highly valuable ecosystem services that include food supply and coastal protection, among others (Moberg and Folke, 1999; Reaka-Kudla, 1997). Biogenic calcification, erosion, and dissolution contribute to the formation of the reef framework constructed of calcium carbonate ($CaCO_3$, mainly aragonite). The balance of carbonate loss and accretion is influenced by biotic and abiotic factors. On reef scale, the main antagonists are calcifying benthic communities, e.g., scleractinian corals and coralline algal crusts, opposed by grazing and endolithic bioeroders, e.g., parrotfish, sea urchins, microbioeroding chlorophytes, boring sponges, and other macroborers (Glynn, 1997; Hutchings, 1986; Perry et al., 2008; Tribollet and Golubic, 2011). The export or loss of carbonate as sediments is considered an essential part, in particular in the wider geomorphic perspective of reef carbonate production states (Cyronak et al., 2013; Perry et al., 2008, 2017). Temperature and carbonate chemistry parameters (e.g., pH, total alkalinity TA, and aragonite saturation state $\Omega_a$, $pCO_2$) have been identified as important players in regulating these carbonate accretion and erosion processes (Albright et al., 2018; Schönberg et al., 2017). Furthermore, different light regimes across depths, water flow, and wave exposure can alter the rates of reef-formation processes (Dullo et al., 1995; Glynn and Manzello, 2015; Kleypas et al., 2001). "

l. 63: Should "slowing down" not be replaced with something like "interrupting"? Corals often die, after all.

⇨ We rephrased the sentences as suggested. The sentence now reads:

"TA and $\Omega_a$, positively correlate with calcification rates (Marubini et al., 2008; Schneider and Erez, 2006), and while calcification rates of corals and coralline algae increase with higher temperature, they have upper thermal limits (Jokiel and Coles, 1990; Marshall and Clode, 2004; Vásquez-Elizondo and Enríquez, 2016)."

l. 64: Convoluted sentence. How about: As ocean acidification decreases the ocean's pH and $\Omega a$ at the same time, calcification becomes energetically more costly (…).

⇨ We rewrote this sentence following the suggestions of two reviewers:

"Arguably, calcification under these conditions could become energetically costlier (Cai et al., 2016; Cohen and Holcomb, 2009; Strahl et al., 2015; Waldbusser et al., 2016)"

l. 67: There are so many examples for OA-enhanced bioerosion by now that you need to use "e.g." or cite an overview. Please also add: "frequently" a hallmark.

⇨ Revised accordingly.

l. 82: Replace "is" widely attributed with "was" to refer to earlier results, not a fact.
l. 83-84: Again, use the past tense to indicate that you refer to oether people's results.
l. 89: Replace "Aside" with "Apart"

⇨ We included these suggestions.

l. 90: What about e.g.

Lazar B, Loya Y. Bioerosion of coral reefs-A chemical approach. Limnology and Oceanography. 1991 Mar 1;36(2):377-83.

Mokady O, Lazar B, Loya Y. Echinoid bioerosion as a major structuring force of Red Sea coral reefs. The Biological Bulletin. 1996 Jun 1;190(3):367-72.

Alwany MA, Thaler E, Stachowitsch M. Parrotfish bioerosion on Egyptian red sea reefs. Journal of experimental marine biology and ecology. 2009 Apr 15;371(2):170-6.

Bertram GC. 60. Some Aspects of the Breakdown of Coral at Ghardaqa, Red Sea. Journal of Zoology. 1936 Dec 1;106(4):1011-26.

Erez J, Reynaud S, Silverman J, Schneider K, Allemand D. Coral calcification under ocean acidification and global change. In Coral reefs: an ecosystem in transition 2011 (pp. 151-176). Springer, Dordrecht.

Hassan M. Modification of carbonate substrata by bioerosion and bioaccretion on coral reefs of the Red Sea. Shaker Verlag; 1998.

Zundelevich A, Lazar B, Ilan M. Chemical versus mechanical bioerosion of coral reefs by boring sponges-lessons from Pione cf. vastifica. Journal of experimental biology. 2007 Jan 1;210(1):91-6.

Mokady O, Graur SR. Coral-host specificity of Red Sea Lithophaga bivalves: interspecific and intraspecific variation in 12S mitochondrial. Molecular Marine Biology and Biotechnology. 1994;3(3):158-64.

Cornelia Maier 1997. Distribution and abundance of internal bioeroders in coral reefs. A field survey in the northern Red Sea. MSc (diploma) thesis, ZMT & Bremen University, Germany, 94 pp.

And there is more on calcification as well:

Braithwaite CJ. Patterns of accretion of reefs in the Sudanese Red Sea. Marine Geology. 1982 Feb 1;45(3-4):297-325.

Etc…

There was a bit of an overview re bioerosion and lack of data from the Red Sea in

Schönberg CH, Fang JK, Carballo JL. Bioeroding sponges and the future of coral reefs. In Climate Change, Ocean Acidification and Sponges 2017 (pp. 179-372). Springer, Cham.

⇨ In the corresponding section we refer to reef-scale carbonate budget studies "which considered both calcification and erosion/dissolution" (as stated in line 100 of the original manuscript). The studies suggested here focus only on one of the two aspects and do not integrate calcification and erosion into a full carbonate budget picture. Since we already provided examples of Red Sea studies investigating the calcification aspect, we added two more references focusing on the bioerosion aspect.

Text now reads:

"Little is known about the reef-scale carbonate budgets of Red Sea coral reefs (Jones et al., 2015). Apart from one early assessment of reef growth capacity for a high-latitude reef in the Gulf of Aqaba (northern Red Sea) that considered both calcification and bioerosion/dissolution rates (Dullo et al., 1996), studies only report calcification rates (e.g., Cantin et al., 2010; Heiss,

1995; Roik et al., 2015; Sawall and Al-Sofyani, 2015) or focus on bioerosion generally caused by one group of bioeroders (Alwany et al., 2009; Kleemann, 2001; Mokady et al., 1996)."

l. 91: OR bioerosion.

⇨ Revised accordingly (see above).

l. 100: Settlement blocks capture ambient endolithic bioerosion rates only after several years, see research by e.g. William Kiene. Blocks can underestimate bioerosion by magnitudes. It is thus good that you had a set of 30 mo ones, which may still be a bit early, but better than in other studies. Consider that you may have captured an early stage endolith community, which would likely be dominated by different organisms than later. Please replace "on" potential drivers with "of".

E.g. Kiene WE. A model of bioerosion on the Great Barrier Reef. InProc 6th int coral Reef Symp 1988 Aug (Vol. 3, pp. 449-454).
Hutchings PA, Kiene WE, Cunningham RB, Donnelly C. Spatial and temporal patterns of non-colonial boring organisms (polychaetes, sipunculans and bivalve molluscs) in Porites at Lizard Island, Great Barrier Reef. Coral Reefs. 1992 Apr 1;11(1):23-31.
Kiene WE, Hutchings PA. Bioerosion experiments at Lizard Island, Great Barrier Reef. Coral reefs. 1994 May 1;13(2):91-8.

Settlement blocks capture ambient endolithic bioerosion rates only after several years, see research by e.g. William Kiene

⇨ We agree and discuss $G_{net}$ results in more detail considering the results on succession of bioeroding taxa, see specifically section 4.1.1.

l. 101: provides "a" broad insight

⇨ Corrected accordingly.

l. 106: As the other 2 papers are in a slightly different context it would be good to provide the basic data on the cross shelf differences?

⇨ This is presented as one part of the manuscript: now chapter 3.4 'Abiotic parameters relevant for reef growth'

l. 111: Nearshore-fore, midshore-fore, midshore lagoon and offshore-fore? Why is the lagoon in there? The midshore data are not independent of each other.

⇨ All backreef data (and lagoon data) have been removed from the revised manuscript. We now refer to the three exposed reef sites along the shelf gradient, i.e, offshore, midshore, nearshore reef.

l. 114: This sentence can again be made easier to read by dropping commas: At each

station additional seawater samples were collected on SCUBA for 5 - 6 consecutive weeks during each of the seasons for the determination of inorganic nutrients and carbonate chemistry: nitrate…

⇨   The sentence now reads as follows:

"At each station seawater samples were collected on SCUBA for 5 - 6 consecutive weeks during each of the seasons to determine inorganic nutrients, i.e, nitrate and nitrite ($NO_3^-$ & $NO_2^-$), ammonia ($NH_4^+$), phosphate ($PO_4^{3-}$), and total alkalinity (TA) (Table S1)."

l. 119: Use small characters in the title, also in l. 120 for the loggers

⇨  Amended accordingly.

l. 122: Insert "the" in front of "pH probes"

⇨  This passage was removed along with pH data.

l. 127: "cubitainer" is a brand name. Either use "container" or add a bracket with the producer info.

⇨  We now write: "…using 4 L collection containers".

l. 128: Replace "over" with "via" or "through" or something

⇨  Amended accordingly.

l. 129: Which were the discrete samples, the 4L or the syringe samples?

⇨  "Seawater samples were collected on SCUBA at each of the stations using 4 L collection containers (Table S1). Simultaneously, 60 mL seawater samples were taken through a 0.45 μm syringe filter for TA measurements."

l. 140: salinity with a small letter

⇨  This passage was removed along with carbonate chemistry data, following reviewer's suggestions.

l. 142: use small letter in "free scale" and provide reference for free the free scale being a good equivalent, e.g.
Dickson AG. pH scales and proton-transfer reactions in saline media such as sea water. Geochimica Et Cosmochimica Acta. 1984 Nov 1;48(11):2299-308.
Waters JF, Millero FJ. The free proton concentration scale for seawater pH. Marine Chemistry. 2013 Feb 20;149:8-22.

⇨ This passage was removed along with carbonate chemistry data, following reviewer's suggestions.

l. 149: How many blocks were there in total? 64?

l. 151: How were the blocks deployed? It would make a huge difference whether they were fastened directly on the bottom (allowing direct lateral borer invasion and urchin grazing), as opposed to placing them on some sort of rack (where only larval settlement occurs and grazer access would be reduced).

⇨ We included this information (3 reef sites x 3 deployment times x 4 replicates = 36 blocks). Also, we added the orientation of blocks in the reef and an image of the deployment rack. We expand on the implication of this deployment in the Discussion.

The section now reads:

"Net-accretion/erosion rates ($G_{net}$) were assessed using a "limestone block assay". Blocks cut from "coral stone" limestone were purchased from a local building material supplier in Jeddah, KSA, and each block was fixed with one stainless steel bolt to aluminum racks permanently deployed at the monitoring station of each reef site (a total of 36 blocks, n = 4, Fig. S1). The blocks were oriented in parallel to the reef slope with one side facing up while the other side was facing down towards the reef. Block dimensions were 100 x 100 x 21 mm with an average density of $\rho = 2.3$ kg $L^{-1}$. Blocks were dry-weighed before and after deployment on the reefs. "

l. 154: Presumably after bleaching the blocks were thoroughly rinsed to remove the bleaching salts?

⇨ We have added this information:

"Upon recovery, the blocks were treated with 10 % bleach for 24 - 36 h and rinsed with deionized water to remove organic material and any residual salts."

How were the sites distributed over the exposure times? Can you provide a schematic figure? Wouln't it be much easier to understand if you gave the lagoon the same status as the backreefs? Then you would have 3 fore-reef, exposed sites and 3 sheltered sites? It does not make much sense that the 30 mo approach included the lagoon, and you should not include it giving it the same status as the fore-reef sites. In your case the factor "exposure" would be nested within reef transect, it would be like a subsample per reef.

⇨ We now provide a schematic overview of the sites. The backreef sites and backreef lagoon site were removed from the revised manuscript.

l. 160: How was this information matched with biota in the blocks? Were any of the species identified, either those seen on the reef or in the blocks? The Red Sea is not well represented in the borer taxonomies, except for the bivalves.

⇨ We agree that providing detailed composition of bioeroding fauna would be a valuable contribution the MS. Unfortunately, we did not investigate this aspect, as the focus of the study was to measure the mass loss/accretion of the limestone blocks.

l. 163: WERE assessed

⇨ Amended accordingly.

l. 164: What do you mean with non-calcifyers? General reef biota or only bioeroders?

l. 166: Sponges are a major group of bioeroders, yet you grouped them with algae? And didn't you have area that was not covered by living organisms, e.g. patches of sand? It would have been good to have a value for bioerodable calcium carbonate as opposed to area covered by calcifyers or non-bioeroders. Borer abundances strongly vary with substrate type and dead surface areas and have to be normalised to the available substrate in order to make their occurrences comparable between sites (mainly shown for sponges):

Carballo JL, Bautista-Guerrero E, Leyte-Morales GE. Boring sponges and the modeling of coral reefs in the east Pacific Ocean. Marine Ecology Progress Series. 2008 Mar 18;356:113-22.

Schönberg CH. Monitoring bioeroding sponges: using rubble, quadrat, or intercept surveys?. The Biological Bulletin. 2015 Apr;228(2):137-55.

That doesn't apply to your block data (unless you fastened them in areas without abundant dead substrate = lower of larval supply), but will have an effect on your distributions.

⇨ We restructured the paragraph and refer to new Table S2 to provide more detail on the procedure.

The text now reads:

"Community composition and coverage of coral reef calcifying groups were assessed in six replicate transects per site using the belt-transect rugosity method (Perry et al., 2012) as detailed in Roik et al. (2015). From these surveys we extracted data on benthic calcifiers (% cover total hard coral, % hard coral morphs (branching, encrusting, massive, and platy/foliose), % major reef-building coral families (Acroporidae, Pocilloporidae, and Poritidae), % cover calcareous crusts, % recently dead coral, and % rock surface area for carbonate budget calculations (Table S2). In addition, benthic rugosity was assessed in the same transects following the *Chain and Tape Method* (n = 6, Perry et al., 2012)."

l. 170: You did not assess general borer densities, but only the two dominant epilithic bioeroder groups or grazers. Please reword, be more specific (also 2.5). It would also be important to know whether you only included parrots that really have an impact on bioerosion or also others, like scapers and spp. that mainly eat erect sea weeds (see references below). I guess you included all, because you had fairly small size classes in there? That would dilute the data. Pity that you did not look the borers.

Bellwood DR, Choat JH. A functional analysis of grazing in parrotfishes (family Scaridae): the ecological implications. In Alternative life-history styles of fishes 1990 (pp. 189-214). Springer, Dordrecht.

McAfee ST, Morgan SG. Resource use by five sympatric parrotfishes in the San Blas Archipelago, Panama. Marine Biology. 1996 May 1;125(3):427-37.

Lokrantz J, Nyström M, Thyresson M, Johansson C. The non-linear relationship between body size and function in parrotfishes. Coral Reefs. 2008 Dec 1;27(4):967-74.

⇨ We are now more specific when referring to parrotfishes and sea urchins. The corresponding section headers were changed to: "Epilithic bioeroder/grazer populations along the cross-shelf gradient" and "Abundances and biomasses of epilithic bioeroder/grazer".

l. 182: How did you assess accretion and bioerosion? You can provide the details in the supplement, but the main text still needs to be understandable by itself, so why not say that the local accretion/erosion data were assessed by you and published earlier? Gnetbenthos was derived from the blocks? Then you can only rely on 30 mo data for the forereefs (the other periods being even shorter)? And you only have net values? Considering the extreme variation you found this may be a problem.

l. 184: I assume that the "site-specific" net data refer to the blocks? How did you assess calcification by itself? Please specify

⇨ To clarify this aspect, we now provide the method for calcification and erosion assessment in the text. The new extended $G_{budget}$ methods text, should contain all requested details:

"Ecosystem scale reef carbonate budgets, $G_{budget}$ [kg m$^{-1}$ y$^{-1}$], were determined following the census-based *ReefBudget* approach by Perry et al. (2012) (Table 1). $G_{budget}$ incorporates local census data, site-specific net-accretion/erosion data ($G_{net}$ over 30 months) and calcification data (buoyant weight measurements) collected for the present and from a previous study (Roik et al., 2015). Importantly, the approach incorporates epilithic bioerosion, which is based on abundance rather than assessment of actual bite or erosion rates; therefore, parrotfish and sea urchin census data collected in this study are readily employed in the *ReefBudget* calculations using bite and erosion rates from the literature (Alwany et al., 2009; Perry et al., 2012). In summary, site-specific benthic calcification rates ($G_{benthos}$, kg m$^{-1}$ y$^{-1}$), net-accretion/erosion rates of reef "rock" surface area ($G_{netbenthos}$, kg m$^{-1}$ y$^{-1}$), and epilithic erosion rates by sea urchins ($E_{echino}$, kg m$^{-1}$ y$^{-1}$) and parrotfishes ($E_{parrot}$, kg m$^{-1}$ y$^{-1}$), were determined for the $G_{budget}$ calculations (Fig. 1 (b) and Fig. 3 (a)). A detailed account of Red Sea specific calculations and modifications of the *ReefBudget* approach employed in this study are outlined in the supplementary materials (Text S1, Equation box S1-3, and Tables S2-7)."

l. 202: "showed" = in the past, and delete "a"

⇨ This passage was removed along with carbonate chemistry.

l. 212: Gbudget DATA were tested…

⇨ Amended accordingly.

I am not quite sure I understand your data design? You had the factors "time" and "reef = distance to shore". You only had very few replicates (blocks) at the lowest level with N=4. This is very small sample size for something as variable as net calcification/bioerodion. Where does your factor "reef area = fore/back/lagoon" go? I think you are quite wrong to use simple ANOVAs, because "reef area" is not independent and should be nested within "distance from shore". So your factors "time" and "distance from shore" are independent between factors and fully crossed for 6 and 12 mo, but "reef area" is a within factor and presently unbalanced. Thus you don't have a 1-factorial simple ANOVA, but a 2-factorial mixed model with the additional factor "reef area" nested within "distance from shore". It would make your life easier if you would call "reef area" "exposure to water movement" and give backreef and lagoon the same status. Having only 4 blocks at the lowest level means you will have to be very careful with your data design, and your present analysis seems misguided to me. I think you should evaluate your data like this:
DISTANCE FROM SHORE x NEARSHORE MIDSHELF OFFSHORE TIME OF EXPOSURE:
6 and 12 mo 30 mo
Exposed: forereef Exposed: forereef Exposed: forereef
Sheltered: backreef Sheltered: lagoon Sheltered: backreef
Distance from shore
Time of exposure
Time of exposure x distance from shore
Exposure to water movement (nested in distance from shore)
Why don't you evaluate everything in PERMANOVA, which saves the need for transformation and gives you more test power than nonparametrics, as well as allowing testing for factor interaction etc.
l. 221: You have 3 reef sites (near, mid, off) and 2 or 3 reef areas (fore, back, lagoon), which are nested within reef site. You should not use 4 levels in this approach.

⇨ As suggested, we simplified the study design in order to focus on cross-shelf patterns. We removed the data points for the sheltered sites (backreefs and lagoon), which contributed to an unbalanced study design. Statistical tests are now performed under simplified and fully crossed study designs, i.e. $G_{net}$ data under the 2-factorial design of 'deployment time' (6, 12, 30 months) and 'reef site' (offshore, midshore, nearshore). In the Material and Methods section, we clarify the block assay design (no repeated-measures design, each block was measured only once). The $G_{budget}$ data are analyzed under a 1-factorial design with the factor 'reef' (offshore, midshore, nearshore).

⇨ To clarify the sampling design we now make a clearer distinction between the two reef growth metrics, $G_{net}$ and $G_{budget}$, by including a new Figure 1, which provides a map of the reef sites including schematic diagrams of the sampling designs for $G_{net}$, $G_{budget}$, and the abiotic monitoring. Additionally, we provide a glossary of all acronyms used for the various metrics (new Table 1).

l. 225: Why did you exclude urchins for the data analysis of the blocks? Wouldn't urchins have grazed on your blocks? Same for coral cover. Your data are net values, i.e. they include coral settlement. Please increase the range of the tested parameters for the block values. Did you cull out covariables?

⇨ Sea urchin abundance were included in the analyses. However, since no coral recruit were observed on the blocks by visual inspection upon recovery, coral abundance is not included in the analysis.

Text now read as follows:

"Biotic predictors were variables that likely impacted the limestone blocks, i.e. parrotfish abundances, sea urchin abundances, calcareous crusts cover, and algal and sponge cover. Since we did not observe any coral recruits on the blocks, we did not include % coral cover and related variables in the correlations."

⇨ Since we removed the model and present only single correlation analyses, we do not specify covariables.

l. 240: The difference across the shelf? I assume the water became cooler with distance from shore? This needs to be clear without access to the supplement. The next sentence seems to be wrong? Do you mean the "OFFshore and the midshore"?

⇨ The table from l.240 was removed and all abiotic data is now presented in Table 4 of the revised manuscript. The text has been revised accordingly and substantially.

⇨ The nearshore reef in the central Red Sea is the coolest reef over the winter (due to the cooling effect of the winds and the shallow water), and the hottest in the summer, as it is lacking the cooling effect of the open ocean water masses (see e.g., Roik et al. 2015 and Roik et al. 2016). This effect however goes beyond the scope of this study.

l. 248: All 6 sites? Are these hierarchical means, i.e. per reef area and then seasonal means across the distances? Please noe that in nested designs the means need to respect the same data hierarchy and need to be calculated stepwise. This part is also not quite clear in the earlier parts of the Methods, please specify and consider for all displayed means. "Nearshore reef" is that back or fore or a mean of back and fore?
l. 253: replace "in" with "at the sites"
l. 254: delet "was"

⇨ pH data and consequentially this paragraph were removed from the revised manuscript.

l. 258: "while the midshelf lagoon", replace "was" with "were". Did you include the backreef sites? Looking at your figures shows me that you didn't, which is a shame. In this case you can explain differences between sheltered vs. exposed only for the midshelf site and should not generalise.

⇨ Data from the backreef sites including the midshore lagoon reef site were removed from the revised manuscript.

l. 257: please reword to "small at the exposed offshore and midshore sites"

⇨ Data from the backreef sites including the midshore lagoon reef site were removed from the revised manuscript.

l. 261: delete "a"

⇨ Corrected accordingly.

l. 264: replace "contents" with "levels"

⇨ Corrected accordingly.

l. 268: All this and some of the following is horrible to read with all the parameter abbreviations and long brackets. It is difficult to find the bits of sentences in between. Having all the data in clean overview in a table, is it necessary to clutter the text with so many brackets or can you delete a few of them? The last you can probably do is reducing the number of repetitive units, e.g. "(2422 μmol AT kg-1, 2076 μmol CT kg-1,and 1821 μmol HCO3- kg-1)" could become "(2422, 2076, and 1821 μmol kg-1,respectively)"?
l. 271: the "p" in pCO2 is usually written in italics, which makes it clear that it is a defined unit (correct throughout). Could you also put definitions in the text for the main parameters, not just in the table legend? "Ranged" infers the use of "from… to"
l. 272: "increaseD" (past tense). Replace "of note, …" with "it should be onted that…"?
What do you mean with "propagates to uncertainty"? Do you mean "was in part near the resolution level"?

⇨ Carbonate system data and subsequently this paragraph were removed from the revised manuscript.

l. 279: hyphen between ocean and facing

⇨ As we removed data from the backreefs and refer only to the exposed/fore reef sites, terms are removed from the text.

l, 284: Accretion is half of your budget. You need to put some of this in your main text, I think. The results need to be clear in the end, e.g. what they are based on.

⇨ This is reported at a later point in the revised manuscript "Results of Carbonate budgets 3.3".

l. 286: They are epilithich macrobioeroders. Please make that clear, you did not assess any of the endolithic bioeroders.

⇨ We now refer to parrotfishes and sea urchins as "Epilithic bioeroders/grazers throughout the revised manuscript.

l. 288: Urchin and fish means are often in the same range of magnitude as the error values. The large errors make these values highly unreliable.

⇨ We are aware of this and we now relate to this fact in the Discussion:

"In the correlation analyses, both grazers, i.e., sea urchins and parrotfish, negatively correlated with $G_{budget}$, however these correlations were not very strong ($\rho \sim -0.5$) and non-significant. The weak correlation may be influenced by a considerable variability in the reef census dataset, specifically regarding parrotfish abundances. Observer bias (parrotfish keep minimum distance from surveyors during dives and may therefore not enter survey plots; pers. obs.), natural (e.g., species distribution, habitat preferences, reef rugosity, and mobility or large roving excavating species, such as *Bolbometopon muricatum*), and/or anthropogenically-driven factors (e.g., differential fishing pressure) may also contribute to the observed data heterogeneity (McClanahan, 1994; McClanahan et al., 1994). Indeed, the Saudi Arabian central Red Sea has been subject to decade-long unregulated fishing pressure, which has significantly altered reef fish community structures and reduced overall fish biomass compared to less impacted Red Sea regions (Kattan et al., 2017). Unregulated fishing could at least in part explain the differences of fish abundance dynamics between the present study and reefs on the GBR and the Caribbean. The heterogeneity of grazer populations further propagates into the $G_{budgets}$, resulting in a considerable within-site variability that reduces power of statistical tests and correlations."

l. 290: I disagree with the statements of this para. I used the raw data from the supplement to get a better visual impression of your data. I will send an Excel file in attachment to show what I mean. You cannot really say that the urchins were most abundant nearshore and then decreased, because all the error bars overlapped. There is no evidence for a trend. I would strongly recommend to include figures like the ones I tried out, because then you can easily see that the urchins cannot be matched to any particular straightforward pattern, with the possible exception of a much reduced urchin biomass at the offshore site. This would be due to much smaller urchins, because the abundances themselves do not look significantly different.
For the parrotfish, however you will be able to find differences, and this looks quite interesting: Abundances increased across the shelf at the sheltered sites, but decreased at the exposed sites. Bioerosion was strongest inshore exposed sites, but with huge error bars. Overall means: no difference for anything.
So it would be much more interesting to display the effects for both factors cleanly separated, not sure whether it is so interesting to know the range – e.g. especially the bigger parrotfish would have an impact for bioerosion. The overall plots suggest that the parrotfish will have the larger influence (should better be plotted with a secondary axis).

⇨ We thank the reviewer for their efforts and agree that more analyses could be done on the abundance and biomass data from the surveys. We deem such additional analyses to be beyond the scope of the current manuscript. Hence, we decided to report on the figures and used them as input data for the $G_{budget}$ calculations only. We therefore report the numbers in the Results and Supplement (Table S4 and S6 in the revised manuscript).

"Parrotfish mean abundances and biomass estimates ranged between $0.08 \pm 0.01$ and $0.17 \pm 0.60$ individuals m$^{-2}$, and $24.69 \pm 6.04$ and $82.18 \pm 46.67$ g m$^{-2}$, respectively (Table S6). The largest parrotfishes (category 5 parrotfish, i.e., $> 45 - 69$ cm fork length) were observed at the midshore site. With the exception of the midshore reef, category 1 ($5 - 14$ cm) parrotfish were commonly observed at all sites. Large parrotfishes (category 6 with $\geq 70$ cm fork length) were not observed during the surveys. For sea urchins, mean abundances of $0.002 \pm 0.004 - 0.014 \pm 0.006$ individuals m$^{-2}$ per site were observed and mean biomasses $0.05 \pm 0.04 - 1.43 \pm 0.98$ g m$^{-2}$ estimated per site, respectively (Table S4). The midshore site exhibited the largest range of sea urchin size classes (from categories 1 or 2 to the largest size class 5), while at the other two exposed sites, only the two smallest size classes of sea urchins were recorded."

⇨ As the reviewer makes a valid point considering the large SDs associated with data on epilithic bioeroders, we specifically point this out in the Results and in the Discussion (we gave examples from the revised text above).

l. 301: Cumulative? Please explain. My understanding was that you had 4 replicates per situation, and no repetitive measurements? You can't add results from different block dets together, you need to respect successional stages of settlement. Replace "in" with "at".

⇨ We revised this section to make the procedures clearer. This was also requested by another reviewer.

It now reads:

"Net-accretion/erosion rates G$_{net}$ were measured in assays over periods of 6, 12, and 30 months in the reef sites along the cross-shelf gradient. These measurements represent the result of calcification and bioerosion processes impacting the deployed blocks. Visible traces of boring endolithic fauna were only found on the surfaces of blocks recovered after 12 and 30 months as presented in Fig. 2 (c)-(f). A brief visual inspection of the block surfaces after retrieval showed colonization by coralline algae, bryozoans, boring sponges, small size boring worms and clams, as well as parrotfish bite-marks. No coral recruits were identified. Further analyses of the established calcifying and bioeroding communities were not within the scope of this study. G$_{net}$ based on the 30-months deployment of blocks ranged between -0.96 and 0.37 kg m$^{-2}$ y$^{-1}$ (Table 2). G$_{net}$ for 12 and 30-months blocks were negative on the nearshore reef (between -0.96 and -0.6 kg m$^{-2}$ y$^{-1}$, i.e., net erosion is apparent), slightly positive on the midshore reef ($0.01 - 0.06$ kg m$^{-2}$ y$^{-1}$, i.e., almost neutral carbonate production state), and positive on the offshore reef (up to 0.37 kg m$^{-2}$ y$^{-1}$, i.e., net accretion of reef framework)."

Maybe it would be good to have Figure S1 in the MS. The respective analyses again need to be nested (exposure within distace from shore). Pity that you did not fully replicate all settings. The block data for 6 and 12 mo do not mean much and do not separate out in your supplementary figure. Only after 30 mo the mid- and offshore blocks start to show different patterns. But you can only judge the "exposure to water movement" effect at the midshelf site.

l. 313: Please recalculate respecting the nesting level. It is quite bad that your error is larger than the mean, maybe it would be better to express that same value only for the forereef data? Or separate for exposed and sheltered?

⇨ We removed sheltered sites (backreefs and lagoon) from the revised manuscript, which were accountable for the unbalanced original study design (backreef were not all replicated in $G_{budget}$) and conducted the analyses on the remaining dataset.

⇨ As suggested we moved Figure S1 to the main manuscript (Figure 2 in revised manuscript). The figure has been adjusted to represent the revised dataset. Also, it is now presented as a bar graph that features the same structure as the presentation of the $G_{budget}$ data.

l. 318: What is "net-accretion/erosion of bare substrate Gnetbenthos"? Is that different from Gnet? Maybe 2.5 could be a bit ore detailed?

⇨ This is now outlined in the new Table 1, which provides a glossary for all terms/metrics used in the study.

l. 322&323: showED, negatively correlated with what?

Revised section reads:

"Strong and positive $G_{net}$-correlates were calcareous crust cover, $NO_3^-$&$NO_2^-$, $PO_4^{3-}$, and TA. Negative correlates were salinity, diurnal pH variation, and parrotfish abundance (strong correlates: $\rho > |0.75|$, $p < 0.001$). Abiotic $G_{budget}$-correlates were exactly the same set of variables as for $G_{net}$. Biotic correlates of $G_{budget}$ were only positive relationships, including calcareous crusts, hard corals, and rugosity. Parrotfish and sea urchin abundances had a negative effect on $G_{budget}$. However, the correlation was weak and not significant ($\rho \sim -0.5$). The non-calcifying benthos, which represents the coverage by algae, soft corals, and sponges, was not correlated with the dynamics of $G_{budget}$ and was correlated only weakly and not significantly with $G_{net}$ ($\rho \sim 0.5$) (Table 5, Table S13 and S14)."

l. 329: Can you be more specific about the percentages? If you have 81% of the variation explained, then the difference to 74% would be 7, not 78? Obviously, you haded co-varying factors in your model (e.g. T means and SDs), why didn't you test and cull some of that in PERMANOVA?

⇨ As detailed above in the general response, models were removed from the revised manuscript.

l. 334: Replace "so far" with "to date"

⇨ Revised accordingly.

l. 337: I would not call this "comprehensive", but "detailed' would be OK.

Revised sentence reads:

"Hence, the present study aimed for a detailed picture of reef growth in the central Red Sea by integrating the antagonistic processes of calcification and bioerosion."

l. 338: linked

⇨ We corrected this accordingly.

l. 341: integrated

⇨ We corrected this accordingly.

l. 353: "weather goal"? replace with "standard", delete "as"?
l. 360: Convoluted sentence, difficult to follow. How about:
To ASSESS test the hypothesis whether Red Sea reefs LIKE OTHER MARINE HABITATS
will IN FUTURE reach a critically low Ωa later, and OR maintain a LONG-TERM calcification-
friendly sea water chemistry on longer terms, compared to other tropical reef regions under OA,
an HIGH PRECISION experimentS and a high precision and high resolution monitoring of reef
carbonate chemistry are needed.
l. 367: decreaseD, (comma)… increased
l. 370: replace "on local" with "at", "WERE similar", add "patterns' after
"seasonality"
l. 377: Don't understand sentence: Amongst the range maxima of pH units was
~1.40. Please reword.
l. 378: Replace "in the" with "at the"; siteS, (comma)…
l. 382: What was not considered?
l. 386: delete "s" from "averages"; the wording is not quite clear, better remove
"average" and "for the crosshelf gradient"? Use "gradient" instead of "differences"
in next sentence.

⇨ pH data and carbonate system discussion are removed from the revised manuscript.

l. 391: Replace "supplied" with something like "modified", "affected" or "replenished".
Please provide a reaeference for this statement. DepleteS.
l. 394: Please simplify this fragmented sentence. It should be "correlated WITH"
l. 397: reflecteD (please carefully check the MS and keep the tenses uniform);
might read better as: "Nearshore habitats with low reef growth capacity were
associated with higher mean temperatures and strong biotic feedbacks that
caused comparatively intensive pH fluctuations."

⇨ pH data and carbonate system discussion were removed from revised manuscript, and this
section has been adjusted accordingly.

l. 399: Silbiger et al. net accretion responded to pH anomalies: "Previously, SMALL-SCALE pH
ANOMALIES HAVE been shown to have a significant impact on LOCAL accretion and erosion
dynamics" (probably too small-scale to generalise, see Schönberg CH, Tribollet A, Fang JK,

Carreiro-Silva M, Wisshak M. Viewpoints in bioerosion research—are we really disagreeing? A reply to the comment by Silbiger and DeCarlo (2017). ICES Journal of Marine Science. 2017 Oct 3;74(9):2494-500.).

⇨ pH data and carbonate system discussion were removed from revised manuscript, and this section has been adjusted accordingly.

l. 400: Use past tense if referring to your own results, insert "can" in front of "exert" if meant as a more general statement.

⇨ Following the reviewer's suggestions, we now use past tense when referring to our measurements and data, but employ present tense when making general statements based on our data or other studies.

l. 405: identifieD, add "as a positive factor" after "concentration" and delete the subclause after the bracket
l. 409: replace "shown" with "demonstrated" (avoid repetition)
l. 410: better to use "coral-algal symbioses" throughout, without "the"?
l. 411: Remove "and", start new sentence with "Conversely"
l. 412: remove "the", make it "in light of"
l. 413: nutrient ratios IN THE CENTRAL RED SEA to understand their effects on LOCAL large-scale and long-term trends of reef growth in the central Red Sea

⇨ We restructured the sentences incorporating the suggestions of the reviewer:

Now reads:

"Interestingly, our study also identified $PO_4^{3-}$ concentration as an abiotic correlate of reef growth. In the Red Sea high N:P ratios indicate that P is a limiting micronutrient, e.g. for phytoplankton (Fahmy, 2003). $PO_4^{3-}$ is not only essential for pelagic primary producers, but also for reef calcifiers and their photosymbionts, such as the stony corals and their micro-algal *Symbiodinium* endosymbionts (Ferrier-Pagès et al., 2016). Experimental studies have demonstrated that $PO_4^{3-}$ provision can maintain the coral-algae symbiosis in reef-building corals under heat stress (Ezzat et al., 2016). Conversely, P limitation can increase the stress susceptibility of this symbiosis (Pogoreutz et al., 2017; Rädecker et al., 2015; Wiedenmann et al., 2013). In light of our results, it will be of interest to link spatio-temporal variation of inorganic nutrient ratios with patterns of reef resilience in the central Red Sea to understand their effects on long-term trends of reef growth."

l. 418: replace "on" with "at"

⇨ We corrected this accordingly.

l. 420: WAS reported, replace "negative production" with "erosional"

⇨ We now write 'erosional' instead of 'negative production state', but we added $G_{budget}$ "data", and maintain "were", as we refer to data in plural.

l. 427: showeD. This part is a bit confusing. The methods implied that parrots were counted as epilithic bioeroders? But here it seems the effect of macroalgal cropping was more important? Given the context, this needs to be made clearer here and in the methods: Which groups were included for what purpose of assessment? In the following the two contrasting effects (macroalga cropping and possible bioeroder control vs direct bioerosion) are mixed together. When more parrots correlates to reef growth, I guess you mostly counted the croppers, not the bioeroders? Potentially there is also a lot of among-biota feedback that can be important, but difficult to separate out, see e.g.

Carreiro-Silva M, McClanahan TR. Echinoid bioerosion and herbivory on Kenyan coral reefs: the role of protection from fishing. Journal of Experimental Marine Biology and Ecology. 2001 Jul 30;262(2):133-53.

Brown-Saracino J, Peckol P, Curran HA, Robbart ML. Spatial variation in sea urchins, fish predators, and bioerosion rates on coral reefs of Belize. Coral Reefs. 2007 Mar 1;26(1):71-8.

Mapstone BD, Andrew NL, Chancerelle Y, Salvat B. Mediating effects of sea urchins on interactions among corals, algae and herbivorous fish in the Moorea lagoon, French Polynesia. Marine Ecology Progress Series. 2007 Mar 5;332:143-53.

McClanahan TR. Response of the coral reef benthos and herbivory to fishery closure management and the 1998 ENSO disturbance. Oecologia. 2008 Feb 1;155(1):169-77.

Kennedy EV, Perry CT, Halloran PR, Iglesias-Prieto R, Schönberg CH, Wisshak M, Form AU, Carricart-Ganivet JP, Fine M, Eakin CM, Mumby PJ. Avoiding coral reef functional collapse requires local and global action. Current Biology. 2013 May 20;23(10):912-8.

l. 438: Use "biomass was" (singular)
l. 439: insert "sites" after "offshore"
l. 443: Bit lost here. You cite evidence for fishing pressure, but the parrots were one of your main structuring force? How then do your sites compare to the less fished sites?

⇨ We removed the models and this part of the Discussion in the revised manuscript.

l. 441: replace "is" with "has been"

⇨ We corrected accordingly.

l. 448: This works only when using the 30 mo blocks, right?

⇨ This section has been substantially restructured in the revised manuscript. The question is now clarified in section 4.1, where we provide a detailed discussion of limestone blocks and deployment times.

l. 450: …I don't understand this. If your bioerosion data were all assessed as net values, how can you separate out the two processes to give separate values? No, you are still doing this as net values, right? I think you need to reword this. How about: Net reef accretion increased across reef sites from nearshore to offshore, with offshore accretion being twice as strong as inshore erosion. Our Gbudget estimates can thus be interpreted as evidence for the formation of an offshore barrier reef in the central Red Sea

⇨ We apologize for the misunderstanding and thank the reviewer for pointing this out. We do not refer to the total bioerosion values, but rather speak about the negative and positive G$_{budgets}$. In the text we often use "net erosion" or "net-erosive state" to refer to the negative budgets, and "net accretion" to refer to the positive budgets, as the reviewer correctly noted. These terms are now used consistently used throughout the manuscript. Since this particular text section of l. 450 has been restructured and reworded, the new version reads:

"Reef growth at the central Red Sea cross-shelf gradient averaged $0.66 \pm 2.01$ kg m$^{-2}$ y$^{-1}$, which was driven by the offshore reef, reflecting the location and habitat dependence for reef growth potential. A similar scenario has been observed on a reef platform in the Maldives, where heterogeneous reef accretion occurs and small reef areas can promote substantial net accretion and thereby greatly contribute to the maintenance of reef scale overall positive budgets (Perry et al., 2017)."

l. 452: showeD; delete "states" and change to singular
l. 453: replace "lowest' with 'most intense"

⇨ This section has been substantially restructured and does not exist in this original form in the revised manuscript. The respective contents are now provided in this new paragraph:

"Generally, most block assay studies conducted in various reef habitats and regions mostly delivered net-erosive rates. In the Red Sea net erosion was only observed nearshore (-0.96 kg m$^{-2}$ yr$^{-1}$, 30 months deployment), which is moderate compared to erosion rates measured in the GBR, French Polynesia, and Thailand (-4 or -8 kg m$^{-2}$ yr$^{-1}$) (Osorno et al., 2005; Pari et al., 1998; Schmidt and Richter, 2013; Tribollet and Golubic, 2005). "

l. 457: remove "s" from "in part", put bracket at the end of the sentence or at least after "complex"

⇨ This section has been substantially restructured and does not exist in this original form in the revised manuscript.

l. 459: Confused again… You counted the fish, but you didn't count bite marks? If your parrots included a significant amount of herbivores rather than excavators, how would your biomasses allow a good estimate of fish-generated bioerosion? How can you sound so sure?

⇨ We apologize for the confusion and would like to clarify this: Our calculations are based on abundances of bioeroding parrotfish exclusively, not bite marks or rates. This abundancebased approach to calculate parrotfish erosion rates is outlined in the *ReefBudget* protocol *sensu* Perry et al 2012. To tailor the approach for the Red Sea, we employ local data as presented in Alwany et al 2009 that provide data on activity, bite rates, and volumes (also see Table S7) of the most significant bioeroding parrotfish (please refer to Supplemental Text S1, detailing the approach, and Equation Box S3).

⇨ That being said, since herbivores rely on low energy food, parrotfish will spend most of their waking time feeding. Indeed, Alwany et al. 2009 found parrotfish bite rates to be relatively constant throughout the day, with peak rates in the early afternoon. We are therefore confident that even though we did not assess bite marks or rates in the present study, the approach by Perry being adjusted with Red Sea parrotfish activity data drawn from Alwany et al. 2009 adequately suits the purpose of our study.

⇨ To clarify the use of size and biomass as a proxy for bioerosion: both parameters will inevitably affect the bioerosional potential of an individual parrotfish, simply because bigger parrotfish take bigger bites. Of note, Alwany et al 2009 took parrotfish size into account, covering the entire adult size range of the studied parrotfish. We have utilized their valuable dataset to account for different parrotfish sizes in our survey and to integrate the variation into budget calculations.

⇨ 'herbivores' vs. 'excavators': Please kindly note that we follow the terminology *sensu* Green and Bellwood 2009. 'Herbivore' is not a single functional group but rather a trophic guild comprised of several functional groups: browsers, grazers/detritivores, scrapers/small excavators, and large excavators. While only the large excavators are referred to as bioeroders, scrapers and small excavators remove reef carbonate substrate too. Therefore, their inclusion in bioerosion budgets is warranted. According to Green and Bellwood 2009, the parrotfish taxa monitored in the present study can all be assigned to either the scrapers/small excavators (*Scarus* and *Hipposcarus*) or excavators (*Cetoscarus*). We assume the reviewer is referring to browsers when writing about 'herbivores'? Indeed, browsers do remove only fleshy macroalgae and potentially associated epilithic communities, but not the actual reef substrate. The only genera of parrotfish identified as true browsers, *Calotomus* and *Leptoscarus* (Green and Bellwood 2009) were observed on rare occasions at the studied sites only (pers. obs. C.P.) as they are commonly more abundant in vegetated habitats and were not taken into account in the budget calculations.

⇨ We are clarifying the parrotfish-related concerns raised by the reviewer in the following sections of the Methods, Supplement, and Discussion:

"We focused on the most abundant bioeroding parrotfish species in the Red Sea (Table S7), which encompassed two herbivorous functional groups: excavators and scrapers (Green and Bellwood 2009). Most abundant across study sites were the excavators *Chlorurus gibbus*, *Scarus ghobban*, and *Cetoscarus bicolor*, and the scrapers *Scarus frenatus*, *Chlorurus sordidus*, *Scarus niger* and *Scarus ferrugenius*, as described in Alwany et al., 2009. Additionally, we counted *Hipposcarus harid* which occurred frequently at the study sites, along with members of the genus *Scarus* that could not be identified to species level and were therefore pooled in the category of 'Other *Scarus*'. Both *H. harid* and *Scarus* spp. were broadly categorized as scrapers (Green and Bellwood, 2009)."

"Importantly, the approach incorporates epilithic bioerosion, which is based on abundance rather than assessment of actual bite or erosion rates; therefore, parrotfish and sea urchin census data collected in this study are readily employed in the *ReefBudget* calculations using bite and erosion rates from the literature (Alwany et al., 2009; Perry et al., 2012)."

l. 461: replace "in" with "on"

⇨  We corrected this accordingly.

l. 469: insert "data" after Gbudget, and "with conditions at the majority" This sentence contradicts the last sentence of the paragraph above it. -0.8 to 4.5 is a large range, caused by widely different conditions. How is this "comparable"? I am not sure whether your data are representative enough to be set into a global context.

⇨  We consider it important to compare our data to previous studies. Therefore, we believe that the references to these studies are essential for the manuscript. Most importantly, the $G_{budget}$ approach has been developed precisely for this purpose: to provide a widely standardized measure for inter-reef and inter-regional comparisons. Also, the block assay is a standardized approach and well suited for comparisons with other studies, given that deployment times and measurement methods were considered. This is we set our data in context to other studies.

⇨  The section of l. 469 has been substantially restructured. We think that in this new text the comparisons are better highlighted, as we mention all uncertainties that need to be considered when drawing conclusions from these data.

Now reads:

"Reef growth at the central Red Sea cross-shelf gradient averaged $0.66 \pm 2.01$ kg m$^{-2}$ y$^{-1}$, which was driven by the offshore reef, reflecting the location and habitat dependence for reef growth potential. A similar scenario has been observed on a reef platform in the Maldives, where heterogeneous reef accretion occurs and small reef areas can promote substantial net accretion and thereby greatly contribute to the maintenance of reef scale overall positive budgets (Perry et al., 2017).
The here presented central Red Sea $G_{budget}$ data are well within the range of contemporary reef carbonate budgets in the tropical western Atlantic ($2.55 \pm 3.83$ kg m$^{-2}$ y$^{-1}$) and the Indian Ocean ($1.41 \pm 3.02$ kg m$^{-2}$ y$^{-1}$) (Perry et al., 2018). However, our data along with most contemporary budgets are well below the suggested "optimal reef budget" of 5 - 10 kg m$^{-2}$ y$^{-1}$ observed in "healthy", high coral cover fore-reefs in both geographic regions (Perry et al., 2018; Vecsei, 2004). This coincides with the observed decrease in calcification rates of Red Sea corals due to the global warming trend (Cantin et al., 2010), which might have severe implications for contemporary and future reef formation in the region at large. While the present $G_{budget}$ data strongly suggest effective barrier reef formation in the central Red Sea (substantial accretion on the offshore reef), carbonate accretion rates and therefore reef formation in the central Red Sea may be hampered in the long run by the ongoing warming."

Your overall budget was 0.65 ± 1.73 kg m-2 y-1, i.e. your error was almost 3x as high as your value, so what is the value really? You used net values that were not controlled for calcification, i.e. you could not separate accretion and erosion from your block data, which only had 4 replicates and likely still represented early stages of colonisation, maybe not so typical. And I am not sure how accurately your epilithic biota counts reflect bioerosion. I felt more comfortable with the last sentence of the paragraph before.

⇨ The error of the cross-shelf is indeed larger than the value, which can be expected, as it encompasses both the negative and positive budgets inherent to the investigated cross-shelf gradient. Such a range of data is very common for $G_{budget}$ studies. As we provide the corresponding minima and maxima as well, providing the cross-shelf average and SD should be straightforward to comprehend for the reader.

For further reference please refer to:

Perry, C. T., Alvarez-Filip, L., Graham, N. A. J., Mumby, P. J., Wilson, S. K., Kench, P. S., Manzello, D. P., Morgan, K. M., Slangen, A. B. A., Thomson, D. P., Januchowski-Hartley, F., Smithers, S. G., Steneck, R. S., Carlton, R., Edinger, E. N., Enochs, I. C., Estrada-Saldívar, N., Haywood, M. D. E., Kolodziej, G., Murphy, G. N., Pérez-Cervantes, E., Suchley, A., Valentino, L., Boenish, R., Wilson, M. and Macdonald, C.: Loss of coral reef growth capacity to track future increases in sea level, Nature, 1, doi:10.1038/s41586-018-0194-z, 2018.

Perry, C. T., Morgan, K. M. and Yarlett, R. T.: Reef Habitat Type and Spatial Extent as Interacting Controls on Platform-Scale Carbonate Budgets, Front. Mar. Sci., 4, doi:10.3389/fmars.2017.00185, 2017.

⇨ To avoid further confusion regarding $G_{net}$ and $G_{budget}$, the discussion of the limestone block assays and the reef scale data are now in the two dedicated sections 4.1 and 4.2.

I am even less happy with the historical comparison, where people used a different approach than you, which would again have an impact on the outcome, apart from having been conducted at different sites. You cannot say whether there was a change over time, and you lose credibility by doing so, especially by putting enough weight on this to mention it in the abstract. It would be better to delete 4.4 or to use this part to highlight the lack of historical data in a context that can suitably compared. The last part is OK, but I would not call your data as "valuable baseline" with this extreme error. Please re-assess your data, maybe after an improved data assessment the error value will be lower?

⇨ We adjusted Discussion according to the reviewer's suggestions, in particular by toning down on the conclusions from the comparison with past data. The header of section 4.4 was changed from "Historical perspective" to "Reef growth trajectories in the Red Sea".

l. 479: provided

⇨ Amended accordingly.

l. 510: Please replace "geographic" with "spatial", you cannot generalise enough to claim you can explain things at geographic level.
l. 510: Delete "not higher than but"? Specify "other regions". Across the shelf? Of the world?

⇨ This section has been substantially restructured and does not exist in its original form in the revised manuscript. The above concerns raised by the reviewer should have become obsolete.

l. 510: I am not sure you captured bioerosion very well. I assume that the blocks did not yet reflect the big borers adequately, and did they allow access to grazernioeroders? The small sample size and the missing calcification control also made it difficult to ariive at more reliable values. Given your large error values, bioerosion might have been much higher than your means.

⇨ We added discussion of the effect of deployment time on the blocks and information about how representative erosion rates are at which succession stage, see section 4.1.2.

⇨ We consider the reviewer's remarks on error values as important, and therefore have devoted new parts in the Discussion to address the within-site variability and the implications for the statistical tests and the interpretation of the data, see e.g. sections 4.2.1 and 4.2.2.

l. 512: I would suggest that you delete any statements re historical trends and rather stress the lack of comparable data, urging for more research.

⇨ We consider it essential to reference past studies from the Red Sea for the scope of the manuscript. As suggested by the reviewer, we have toned down our conclusions on the trajectories of Red Sea reef growth and instead emphasize more on the importance of future studies.

References:
Please choose whether you want to use any of the extra literature I provided. Can you please use uniform formats in the references and carefully check through the list for correct formats and writing? E.g. some titles use capitals in each word, others not. Latin names are not always in italics, the species name sometimes starts with a capital letter.
L. 368: should be New York. The 2 in CO2 should be subscript. Etc… Also in the supplement.

⇨ We thank the reviewer for pointing out inconsistencies and suggesting literature, the majority of which was incorporated into the revised manuscript

Fig. 2: Daytime is one word.

⇨ pH data were removed; consequently, this figure was taken out.

Fig. 3: What is the difference between measured and estimated? Giving the two midshelf sites the same status as the near- and offshore sites is misleading. Maybe it would be best to use only the 3 sites, or to leave gaps for the other 2 sheltered sites, or to display the lagoon site in a separate panel series.

⇨ "estimated" (= indirectly assessed) refers to the calculated carbonate chemistry (from the measured variables, pH and TA). This is obsolete now, as these parameters have been removed from the revised manuscript.
⇨ We removed the backreefs resulting in a more straightforward study design.

Fig. 4: Your blocks do not yet look very affected by bioeroders. There is no clear evidence of grazer-bioerosion, yet you gave the parrots a big importance. Did you find any bioeroding molluscs in the blocks (which have an important role in the RS).

⇨ We have indeed seen clams and parrotfish traces on the block surfaces, but have never seen sea urchins on the blocks. We added the information from the visual inspection of the blocks after retrieval:

"Visible traces of boring endolithic fauna were only found on the surfaces of blocks recovered after 12 and 30 months as presented in Fig. 2 (c)-(f). A brief visual inspection of the block surfaces after retrieval showed colonization by coralline algae, bryozoans, boring sponges, small size boring worms and clams, as well as parrotfish bite-marks. No coral recruits were identified."

⇨ Initially we did not include sea urchin abundance in the correlation with $G_{net}$, as they were never observed on the blocks. Despite the absence of quantitative evidence that either sea urchins or parrotfish engaged in bioerosional activity with the blocks, it cannot be ruled out at this stage. Therefore, we included them in the analysis. As we do not have any quantitative data for macroborers such as clams, we cannot investigate further in this direction. The revised Material and Methods reads:

"Biotic predictors were variables that likely impacted the limestone blocks, i.e. parrotfish abundances, sea urchin abundances, calcareous crusts cover, and algal and sponge cover. Since we did not observe any coral recruits on the blocks, we did not include % coral cover and related variables in the correlations.
$G_{budget}$ correlations included all the above-mentioned abiotic variables and 13 biotic transect variables (i.e., parrot fish abundances, sea urchin abundances, % branching coral, % encrusting coral, % massive coral, % platy/foliose coral, % of Acroporidae, % Pocilloporidae, % Poritidae, % total hard coral cover, calcareous crusts cover, algal and sponge cover, and rugosity). Prior to analysis, some of the predictors (i.e., % platy/foliose corals and % Poritidae) were $\log_{10}(x+1)$ transformed to improve the symmetry in their distributions (Table 5 and Table S14)."

Fig. 5: Same problem as for Fig. 3 re site hierarchy. You have two data pairs each reef, sheltered and exposed. These are not independent, which needs to be obvious in the figures as much as it needs to be respected in the analyses. Your in- and offshore error bars are huge. The netbenthos and echino bars need a different scale, this way the data cannot be appreciated. What does the "c" mean over the grey offshore bar?

⇨ "estimated" refers to the calculated carbonate chemistry (from pH and TA). This is obsolete now, as it has been removed from the revised manuscript.
⇨ We removed the backreefs resulting in a more straightforward study design.

Table 1: Please note that your maeans need to be calculated in analogy to your model, and I assume that they are presently incorrect. Within a given season, you need to first calculate means across 4 replicates, then across per reef area (exposed, sheltered –this results in an unbalanced situation, because you only have sheltered values midshelf), then across the reef sites (near, mid, off). Do you mean "calculate" when you write "estimate"?

⇨ We removed the backreefs resulting in a more straightforward study design.

Tab. 2: This is finally a data display in full analogy. What do you mean with "cumulative"? Please explain in Methods.

⇨ We revised the legend, now reads:

"**Table 2. Net-accretion/erosion rates G$_{net}$ in coral reefs along a cross-shelf gradient in the central Red Sea.** G$_{net}$ [kg m$^{-2}$ y$^{-1}$] was calculated using weight gain/loss of limestone blocks that were deployed in the reefs. For each deployment duration, 6, 12, and 30 months, a set of 4 replicate blocks was used. Each block was measured once. Means per reef site and standard deviations in brackets; y = year."

Tab. 3: Again problem with non-independence of midshelf sites and large error values in- and offshore, making the means very unreliable.

⇨ We removed the backreefs resulting in a simplified study design.

Tab. 4: I think you need to reduce the number of variable and remove the covariables.

⇨ We removed the models and now show the single correlation tests only, which makes controlling for covariables obsolete.

Tab. 5: You need to make clear that your R2 you mentioned in the text were accumulative. It may be better to use the un-cumulative ones in the text – more intuitive.

⇨ Obsolete; we removed the models from the revised manuscript.

Supplement: OK, so this makes it clearer how you assessed epilithic bioerosion. I still think you should mention in the methods that you concentrated on species that are known bioeroders, providing the references.

Hoey et al. is printed and a 2016 paper.

⇨ Thank you. We updated this.

Tab S6: Why is fish bioerosion expressed as negative data, but not urchin bioerosion?

⇨ Thank you for pointing this out. Of course, both fish and urchin erosion are supposed to be negative data. We have corrected this accordingly.

Tab S7: Is the first error value 0.6 a printing error or is it really so huge?

⇨ Yes, this is the data variability, which is also obvious in the $G_{budget}$ data (see error bars and standard deviations). We mention this in the Discussion, e.g.

"The weak correlation may be influenced by a considerable variability in the reef census dataset, specifically regarding parrotfish abundances. Observer bias (parrotfish keep minimum distance from surveyors during dives and may therefore not enter survey plots; pers. obs.), natural (e.g., species distribution, habitat preferences, reef rugosity, and mobility or large roving excavating species, such as *Bolbometopon muricatum*), and/or anthropogenically-driven factors (e.g., differential fishing pressure) may also contribute to the observed data heterogeneity (McClanahan, 1994; McClanahan et al., 1994). Indeed, the Saudi Arabian central Red Sea has been subject to decade-long unregulated fishing pressure, which has significantly altered reef fish community structures and reduced overall fish biomass compared to less impacted Red Sea regions (Kattan et al., 2017). Unregulated fishing could at least in part explain the differences of fish abundance dynamics between the present study and reefs on the GBR and the Caribbean. The heterogeneity of grazer populations further propagates into $G_{budgets}$ estimates, resulting in a considerable within-site variability that reduces power of statistical tests and correlations."

Tab S8: That suggests, however, that you included non-eroding parrots after all. Why did you assume these eroded?

⇨ Please kindly refer to our detailed previous response on parrotfish functional groups. We have not included non-eroding parrotfish in our analysis, only excavators and scrapers (*Cetoscarus*, *Chlorourus*, *Scarus*, *Hipposcarus*). All "other Scaridae" (in the original manuscript) were members of the genus *Scarus* that could not be determined to the species level. Most *Scarus* are considered scrapers, which do remove small, albeit significant, amounts of reef framework (Green and Bellwood 2009). Non-eroding parrotfish include the genera *Calotomus* and *Leptoscarus*, which were not monitored in the present study. This is detailed now in the Material and Methods section to avoid confusion.

Tab S6 and S9: Why do you have 4 data sets here if you have 6 in the tables these are based on? You have the data to display all 6 don't you, why don't you show them? Otherwise you have the same problem re your midshore sites here that need to be addressed.

⇨ We removed data from all back reefs now and use three datasets from the exposed sites of the reefs offshore, midshore, nearshore.

Fig. S1: Maybe better in the main text?

⇨ The figure was updated and included in the revised manuscript as Figure 2.

**Anonymous Referee #3 Decision: Accepted subject to minor revisions**

This is a comprehensive study that provides valuable baseline data for reef growth in the central Red Sea. The study covers the timely and relevant topics of climate change and OA and assesses drivers of reef growth in a unique area (the Red Sea) that is of growing interest owing to the health of corals in an extreme environment. The study uses an extensive dataset to assess the relationships between reef growth and various abiotic (carbonate chemistry parameters, temp, phosphate) and biotic (bioeroders, coral cover, etc.) drivers. The study shows that reef growth is positively correlated to parameters indicative of carbonate ion availability (e.g., At, omega, carbonate ion concentration) and to phosphate concentrations; while growth is negatively correlated to temperature, pCO2, and pH variability (biotic feedbacks). Reef growth was also C1 highly correlated to parrotfish abundance. Overall, this is a well-designed, comprehensive study. The manuscript is clear and well-written (though beware of numerous minor grammatical errors). I do have some concerns that should be addressed before publication (below).

⇨ We thank the reviewer for the overall positive evaluation and constructive feedback. We reply to all of his comments in the following.

Referee #3 specific/technical comments

Starting at Line 69: suggest including 1-2 sentences explaining what a census-based calcification budget approach is and explaining how Perry's Caribbean-based methods were modified for the Red Sea (supplemental).

⇨ We thank the reviewer for this suggestion. We added the following sentence:

"To comparatively assess the persistence of reef framework at regional and global scales, a census-based reef carbonate budget (*ReefBudget*) approach that integrates reef site-specific ecological data into the calculation of the erosion-accretion balance was introduced recently (Kennedy et al., 2013; Perry et al., 2012, 2015). […]"

⇨ We now point out the modifications to the method in the supplemental text more clearly:

"This approach corresponds to the methodology described in (Perry et al., 2012) which can be accessed under *ReefBudget* (http://geography.exeter.ac.uk/reefbudget/). Adjustments were made according to the availability of data from the Red Sea reef sites:
- All census data used in our study has been collected from a discrete depth (7.5 and 9 m), while *ReefBudget* considers two depth ranges (0 - 5 m and 5 - 10 m)
- In place of estimating microbioerosion and boring sponge erosion from census data and site-specific or literature-reported erosion rates, we employ site-specific $G_{net}$ data (i.e, net-accretion/erosion rates measured in a limestone block assay) which includes the rates of endolithic bioerosion
We use genus- and site-specific calcification rates (Roik et al. 2015) and species- and size-specific parrotfish erosion rates from the northern Red Sea (Alwany et al. 2009)."

Line 371: what's the cause of the PO43- enrichment in winter?

⇨ We now do not discuss the seasonal dynamics in detail, but rather focus on the abiotic-biotic correlations only. To name or discuss the possible cause for the P enrichment in the central Red Sea, more literature research would be required. Since we shifted the focus away from the abiotic monitoring, we decided that the seasonal dynamics go beyond the scope of the manuscript.

Line 386: average, not averages

⇨ We corrected this accordingly.

Line 402: ": : :manipulations on reef communities in situ: : :" –add in recent manipulative field experiments (e.g., Albright Nature 2016 and 2018).

⇨ These references were added to the MS.

Suggest a glossary to help readers keep track of various terms (e.g., Gnet, Gbenthos, Gnetbenthos, etc.)

⇨ A glossary was added in the revised manuscript as Table 1.

It's somewhat surprising that relatively small changes in carbonate chemistry explain such a large variation in Gnet (even from net accretion to net erosion). According to Table 1, the delta At is < 50 umol in summer, and even less in winter.

⇨ We removed the model and present and discuss correlations only. Despite the small cross-shelf difference, TA is among the stronger positive correlates, which we discuss as follows:

"The increase in TA is often associated with increased carbonate ion concentration and aragonite saturation state, which facilitate the precipitation of carbonates supporting the performance of reef-builders (Albright et al., 2016, 2018; Langdon et al., 2000; Schneider and Erez, 2006; Silbiger et al., 2014). We identified a positive correlation of TA with reef growth in our dataset. The difference in TA across our study sites was small, but in the range of a natural cross-shelf difference reported from other reefs (e.g. reefs in Bermuda, 20 - 40 µmol TA kg$^{-1}$, Bates et al., 2010), and as high as 50 µmol kg$^{-1}$, the TA enrichment that enhanced net community calcification in a reef-enclosed lagoon (Albright et al., 2016). On the other hand, high calcification rates can deplete TA, whereas dissolution of carbonates can enrich TA measurably, specifically in (semi) enclosed systems (Bates et al., 2010), which we do not observe along the cross shelf gradient. It remains to be further investigated how TA dynamics across the shelf relate to reef growth processes."

Is there a plot of discrete pH versus, or plotted on top of, continuous pH – to see how these two methods compared? CTD pH sensors typically aren't very reliable. Further, two different pH scales are used?

⇨ We removed all absolute NBS scale measured pH values and the calculated carbonate chemistry following reviewer's concerns.

---

## Author Response (AR2)

**bg-2018-57: R2**

**"Coral reef carbonate budgets and ecological drivers in the central Red Sea - a naturally high temperature and high total alkalinity environment"**

**Associate Editor Decision: Publish subject to minor revisions**

01 Sep 2018 by Jean-Pierre Gattuso

Comments to the Author:

Dear Author,

Thank you for submitting a revised version of your manuscript submitted to Biogeosciences, which can be accepted for publication after minor revision. Please address the suggestions provided by the reviewers. When submitting the revised version, please let me know which of the changes were not implemented, if any, and why. This will speed up final acceptance.

I look forward to seeing this paper published and thank you for considering Biogeosciences to publish these very interesting results.

Best regards,
Jean-Pierre Gattuso

We thank the editor and reviewers for their positive assessment regarding our manuscript revision. In the following, we attach the two reviewers' reports and respond to their questions and comments in a point-by-point manner. We have addressed and implemented all suggestions. Changed sections or additions to the manuscript are highlighted in the provided revised manuscript.

In brief,

- we now use the more precise unit of kg $CaCO_3$ m$^{-2}$ y$^{-1}$ when referring to carbonate accretion or removal throughout the manuscript.
- we refer to and added a most recent publication of a carbonate budget analysis by Perry et al. 2018 to the Introduction.
- we include more detail about the methodology of the limestone block assay to the Introduction.
- we added more detailed information about the frequency of CTD calibrations to the Materials & Methods.
- we added/reworded parts of the Discussion regarding 1) the observations of the high physicochemical variability in the nearshore reef, 2) coral recruitment on the limestone blocks, and 3) the relationship between coral cover and carbonate budgets.
- we carefully revisited the manuscript to streamline the text and improve reading flow.

**Referee #1: Steeve Comeau, comeau@obs-vlfr.fr: accepted subject to minor revisions**

Submitted on 30 Aug 2018

The authors did an impressive job to take into account the numerous comments from the reviewers. The manuscript is much improved and clearer. It reads well and the data presented represent an impressive data set. I only have few minor comments:

We thank reviewer 1 for his time and positive feedback. We have addressed the remaining minor comments as outlined below:

-line 33: specify kg CaCO3 m-2 y-1

We agree that the suggested unit "$\mathbf{kg\ CaCO_3\ m^{-2}\ y^{-1}}$" is more precise for defining mass weight of accreted or removed $CaCO_3$. We are now using this unit accordingly throughout the manuscript and the supplement.

-line 80: The recently published paper by Perry in Science could be discussed here (I acknowledge that it was not yet published when the authors were revising this manuscript).

We thank the reviewer for pointing out Perry et al 2018. Indeed, we have referred to the new outcomes from the meta-analyses by Perry et al. 2018 in the revised discussion of our data (l. 458-462). We agree it should be included in the introduction as well, as it highlights the application the of carbonate budget data and how it can provide insight into coral reef trajectories. Text reads:

> "Most recently, carbonate budget data were used to explore the relation of vertical reef growth potential and trends in sea level rise suggesting that reef submergence poses a threat as long as climate-driven and human-made perturbations persist (Perry et al., 2018)."

-line 104: There is a lot of introductory material on the census-based approach but nothing on the limestone blocks. It would be good to add few lines on their previous use.

We have added a brief description of the limestone block method to the Introduction. We inserted this in l. 71 – 81 before mentioning of the census-based approach. This order was chosen to reflect the order of reporting in the Methods and Results (limestone block assay first, census-based approach second). Text reads:

> "A number of studies have employed experimental limestone blocks cut from coral skeletons to study reef growth processes (Chazottes et al., 1995; Kiene and Hutchings, 1994; Silbiger et al., 2014; Tribollet and Golubic, 2005). Deployment of such blocks in a reef captures the endolithic and epilithic accretion and erosion agents and forces, simultaneously allowing for the measurement of net-accretion and net-erosion rates. In

particular, these studies have provided insight into the colonization progression and activity of endolithic micro- and macroorganisms."

line 195: How often was the calibration done?

The factory calibration of the CTDs was performed one time, and was in-house verified twice (once before each deployment period). We have included this information and the exact deployment times to the respective paragraph. Text reads:

> "Factory-calibrated conductivity-temperature-depth loggers (CTDs, SBE 16plusV2 SEACAT, RS-232, Sea-Bird Electronics, Bellevue, WA, USA) were deployed at the monitoring stations on tripods at ~0.5 m above the reef to collect time series data of temperature, salinity, and pHNBS at hourly intervals. The pH probes (SBE 18/27, Sea-bird Electronics) were factory calibrated before the winter deployment (9th February - 7th April 2014). Calibrations were verified using NBS scale standard buffers (pH 7 and 10, Fixanal, Fluka Analytics, Sigma Aldrich, Germany) before the winter and the summer deployment (19th June - 23th October 2014)."

line 324: it's interesting to see such a high variability in pH despite a low abundance of living organisms.

Noted. We are mentioning the observation of high variability of pH in the nearshore reefs, which is likely due to biotic feedbacks and discuss this now in more detail. We propose that the low water exchange rates in the nearshore site may enhance the biotic feedbacks from photosynthesis, respiration, and dissolution. In addition, not only the benthic community, but also biological activity in the carbonate reef sediments may be a factor affecting diel biochemical cycles at the nearshore site. We have added discussing the biological activity of sediments. Accordingly, we adjusted the paragraph in the Discussion. Now reads:

> "The nearshore reef is located on the shelf, surrounded by shallow waters of extended residency time and has a lower water exchange rate compared to the other two reef sites (Roik et al., 2016). Evaporation and limited flow, particularly during summer, may increase salinity, which was overall higher at this reef site. However, the difference to the other sites was minuscule and unlikely to have affected calcifying (Röthig et al., 2016) and bioeroding biota. The variability of diurnal pH on the other hand presumably has stronger impacts on the performance of calcifiers and bioeroders. Previously, pH variability across a reef flat and slope were demonstrated to correlate with net accretion dynamics by showing higher net accretion prevailing in sites of less variable pH conditions (Price et al., 2012; Silbiger et al., 2014), which reflects the pattern observed here. The fluctuation in pH may (in part) represent a biotic feedback signature in reef habitats, which entails changes in sea water chemistry caused by dominant biotic processes, i.e., calcification, carbonate dissolution, and respiration/photosynthesis (Bates et al., 2010; Silverman et al., 2007a; Zundelevich et al., 2007). Commonly, such pH fluctuations are influenced by changes in carbonate system variables, e.g. DIC and TA

(Shaw et al., 2012; Silbiger et al., 2014), which can modify the antagonistic processes of calcification and bioerosion/dissolution (e.g., Andersson, 2015; Langdon et al., 2000; Tribollet et al., 2009). In particular, at our nearshore study site, where benthic macro community abundance was low, biological activity in the sandy bottom (e.g., permeable carbonate sands) might be a crucial factor contributing to the biotic feedback (Andersson, 2015; Cyronak et al., 2013; Eyre et al., 2018).”

line 383: I am not sure to understand here, if bioerosion is very high but net calcification still positive that actually demonstrates pretty high calcification rates from coralline.

We thank the reviewer for pointing this out. Yes, the rates from the limestone block assays in the Red Sea offshore reef clearly showed high calcification from coralline algae, which is a stark contrast to the observation from other block assay studies. It was our intention to emphasize this message. We have revised this section by changing the order of the sentences in order to improve logic and flow. Now read as follows:

“Generally, most block assay studies conducted in various reef habitats and regions found net-erosive rates. For instance, studies from reefs in the Thai Andaman Sea and Indonesian Java Sea note that the accretion by calcifying crusts, such as coralline algae, were negligible compared to the high degree of bioerosion measured in the limestone blocks (Edinger et al., 2000; Schmidt and Richter, 2013). In contrast, our limestone block assays captured a substantial net accretion rate, in particular for the offshore reef site in the central Red Sea (0.37 kg $CaCO_3$ $m^{-2}$ $y^{-1}$ net accretion), indicating that accretion was substantial, while erosion was negligible. The midshore reef was characterized by a near-neutral or minor net accretion (0.06 kg $CaCO_3$ $m^{-2}$ $y^{-1}$) on the order of net accretion rates recorded in French Polynesia in reef sites of uninhabited, oceanic atolls (0.08 and 0.62 kg $CaCO_3$ $m^{-2}$ $y^{-1}$; Pari et al., 1998). Notably, our study recorded a net-erosive state only in the Red Sea nearshore site (-0.96 kg $CaCO_3$ $m^{-2}$ $y^{-1}$, 30 months deployment). This is a moderate rate compared to the larger net erosion observed in the GBR, French Polynesia, and Thailand (-4 or -8 kg $CaCO_3$ $m^{-2}$ $y^{-1}$) (Osorno et al., 2005; Pari et al., 1998; Schmidt and Richter, 2013; Tribollet and Golubic, 2005).”

line 407: Any hypotheses to explain the absence of coral recruits?

Not really. Since we did not quantify coral recruitment on the limestone blocks (via e.g. microscopy), we cannot rule out the possibility that newly settled coral polyps might have been missed, although we did not see any during visual inspection. Based on this, we think it is unlikely that corals might have substantially contributed to the accretion on the limestone blocks.

We adjusted the respective text:

Results addition: “No coral recruits were noticed by the unaided eye.”
Discussion addition: “Given that we could not identify coral recruits on any limestone block, we assume that contribution of corals to the measured accretion was minor.

However, we acknowledge that we might have missed some that could be detected by more sophisticated methods (e.g. such as microscopic examination)."

line 445: How do the coral covers compare between those "healthy reef" and the present study?

The reviewer makes a valid point here. We now compare our results with previous reports of "high accretion heathy reefs" (Vecsei 2001) by including the percentages of hard coral cover. We added the following lines to the Discussion:

"The here presented central Red Sea $G_{budget}$ data are within the range of contemporary reef carbonate budgets from the Atlantic ($2.55 \pm 3.83$ kg $CaCO_3$ $m^{-2}$ $y^{-1}$) and Indian Ocean ($1.41 \pm 3.02$ kg $CaCO_3$ $m^{-2}$ $y^{-1}$) (Perry et al., 2018). Notably, these data are below the suggested "optimal reef budget" of 5 - 10 kg $CaCO_3$ $m^{-2}$ $y^{-1}$ observed in "healthy", high coral cover fore-reefs (see data in Perry et al., 2018 and comparisons therein; Vecsei, 2001, 2004). The decline in coral cover is likely central to the reduced carbonate budgets in contemporary reefs. For instance, the reef sites investigated in the present study do not exceed a coral cover of 40 % (as observed for the offshore study site). In comparison, the dataset compiled by Vecsei 2001 encompasses hard coral cover of up to 80% for the Indo-Pacific and up to 95 % for Pacific Islands. Further, the reduced contemporary carbonate budgets coincide with the observed decrease in calcification rates of Red Sea corals at large (Cantin et al., 2010; Steiner et al., 2018). As such, the effect of climate change and the corresponding increase in seawater temperature may have severe consequences via overall decrease in coral reef cover as well as via reduced calcification of the resident corals. Hence, although the present $G_{budget}$ data still suggest effective barrier reef formation in the central Red Sea (substantial accretion on the offshore reef), carbonate accretion rates and therefore reef formation in the central Red Sea may be hampered in the long run by the ongoing warming."

**Anonymous Referee #2 Decision: accepted as is**

Submitted on 01 Sep 2018

The authors did an impressive job to take into account the numerous comments from the reviewers. The manuscript is much improved and clearer. It reads well and the data presented represent an impressive data set. I only have few minor comments:

We thank the reviewer for their time and endorsement of our revised manuscript.